# Umbrella review of meta-analyses on the risk factors, protective factors, consequences and interventions of cyberbullying victimization

K. T. A. Sandeeshwara Kasturiratna [1] ✉, Andree Hartanto [1] ✉,
Crystal H. Y. Chen[1], Eddie M. W. Tong[2,3] & Nadyanna M. Majeed[1,2]

The increasing prevalence of cyberbullying victimization has become a commonplace issue globally. Although research has explored various predictors and consequences of cyberbullying victimization, most focus on a narrow range of variables or contexts, highlighting the need to comprehensively review and synthesize the wealth of empirical findings. We conducted a systematic review of meta-analyses on cyberbullying victimization, incorporating 56 meta-analyses and 296 effect sizes (sample size range 421–1,136,080, sample size median 53,183; searched via EBSCOhost ERIC, EBSCOhost PsycInfo, PubMed, Scopus, Web of Science, 13 cyberbullying-related journals, Google Scholar and ProQuest Dissertations and Theses) to address the following critical questions: (1) What are the crucial sociodemographic and psychological profiles of cyberbullying victims? (2) What critical contextual and environmental factors are associated with cyberbullying victimization? (3) What are the key psychological and behavioural consequences of cyberbullying victimization? (4) How effective are existing interventions in mitigating impacts of cyberbullying? Included meta-analyses had to focus on cyberbullying victimization and report at least one predictor or consequence. A quality assessment was conducted using the Joanna Briggs Institute Critical Appraisal Instrument for Systematic Reviews and Research Syntheses. Findings suggest that females, school-aged populations, traditional bullying victims and frequent internet users were more likely to be cyberbullied. Unregulated school environments and unsupportive parental relationships were also associated with increased cyberbullying victimization. Cyberbullying victimization was consistently associated with negative psychological outcomes, lower school performance and maladaptive coping behaviours. More importantly, the current review found that cyberbullying intervention programmes show promising results. The current review underscores the importance of devoting adequate resources to mitigating cyberbullying victimization.

Amid rapid technological advancements, the internet has become a prevalent platform for social interaction, particularly among youth and adolescents[1–3]. These digital environments, while fostering connections and personal expression[4–8], present new challenges[9], including cyberbullying—an important and growing concern[10]. Referring to intentional acts of aggression carried out via electronic media[11], cyberbullying has become a commonplace issue in recent times[12,13]. Globally, around four in ten adults who use the internet have experienced cyberbullying[14]. In the United States, nearly half of adolescents have experienced at least one instance of cyberbullying[15]. Within Asia, countries such as Singapore, China, Malaysia and South Korea all report high prevalence rates close to 50% (refs. 16,17).

This increasing prevalence of cyberbullying with greater digital media use, however, does not uniformly indicate that more internet usage directly leads to more instances of cyberbullying[18,19]. Indeed, the relationship between increased digital activity and cyberbullying is influenced by various factors, such as digital literacy[20,21], the availability of social support networks[22–24] and the effectiveness of preventative measures[25,26]. These factors vary widely across different social and cultural contexts, highlighting the complexity of cyberbullying as a phenomenon. Given this complexity, defining cyberbullying precisely is essential for the effective dissection of these contributing factors.

## Defining cyberbullying

While there is currently no consensus within research on the precise definition of cyberbullying[27,28], there are some universally accepted elements. First, it is widely recognized that cyberbullying involves electronic media[28]. The term 'electronic media' itself is broad; some definitions restrict it to internet and mobile phones[29–32], while others apply a more detailed taxonomy of technology[33,34]. Given the rapid evolution of technology, it is pragmatic to adopt a broader definition that encompasses both current and forthcoming technologies by using a definition such as 'actions carried out via any electronic means' rather than specifying devices through which cyberbullying occur.

Second, it is also generally agreed that cyberbullying involves a form of aggression towards an individual or a group[28,35]. However, different studies operationalize aggression differently. For instance, most research identifies cyberbullying through behaviours such as sending aggressive messages online[28,35–37], whereas Mills et al.[38] operationalized cyberbullying as online social exclusion. Willard[39] developed a comprehensive taxonomy of cyberbullying that includes flaming, online harassment, outing and trickery, sexting, exclusion, impersonation and cyberstalking.

In consideration, the current work defined cyberbullying as any aggressive or bullying behaviour aimed towards an individual or a group using any electronic means. This definition encompasses aspects such as sextortion (threatening to use an explicit photo or video of someone to make demands/pressure them[40]), online social exclusion (excluding an individual via blocking or distancing over online means[39]) and cyberdating abuse (a form of control and harassment by the dating partner using electronic media[41]), as these behaviours involve aggressive acts via electronic media.

## Cyberbullying versus traditional bullying

It is widely accepted that cyberbullying is an extension of traditional bullying[42], with many researchers modelling their definitions of cyberbullying on the main characteristics of traditionally bullying[43]: intention, repetition and power imbalance[44]. While there is a high correlation between traditional bullying and cyberbullying[28,45,46], marked differences exist between the two. First, the intention behind cyberbullying can often be ambiguous to the victim owing to the lack of non-verbal cues; actions perceived as humorous by the perpetrator might be interpreted as hurtful by the recipient[27,47]. Second, the concept of repetition differs within the online realm; perpetrators may only commit a single aggressive act to victims, but that one post, comment, or image does not need to be reposted by the original perpetrator to be considered

repetitive[48,49]. It can be shared or forwarded by others, continually harming the victim without further direct action from the perpetrator. Third, power imbalances are not always a prerequisite for cyberbullying[42]. Owing to the anonymity of digital platforms and the lack of physical confrontation, individuals who may not typically engage in face-to-face bullying can easily perpetrate online harassment[47]. While research on cyberbullying has attempted to define power imbalance in terms of digital literacy[50], this may not necessarily confer notable advantages in the current environment as the proliferation of various platforms and their ease of use has simplified the act of bullying online[27].

Most crucially, by eliminating the need for face-to-face interaction and allowing anonymity[51,52], cyberbullying allows for online disinhibition[53]. According to the Online Disinhibition Effect Thoery[53], the internet, which offers anonymity by allowing users to adopt usernames, allows individuals to separate their online actions from their offline identity. This reduces the sense of responsibility for their online actions and motivates perpetrators to engage in cyberbullying, which increases the possible incidence of victimization[53] and allows victims themselves to become future cyberbullies[54,55]. Thus, it is imperative to better understand cyberbullying as a phenomenon distinct from traditional bullying to prevent creating a vicious cycle of internet-based aggressive behaviour that perpetuates negative consequences.

## Measurement issues in cyberbullying research

Another focal point within cyberbullying research is the challenge of measurement. The absence of a unified definition complicates measurement as studies often adopt divergent definitions and employ various scales that may not fully capture the phenomenon. For instance, some studies limit cyberbullying to online peer victimization[34,56], while others do not[29,33]. Additionally, older studies frequently omit definitions of cyberbullying[28,57]. While newer studies tend to provide one, they vary substantially in word choice, using terms such as 'cyberaggression', 'cyberstalking' or 'cyberbullying', which can confuse respondents and hinder comparability across studies[58]. The development of cyberbullying scales also shows inconsistencies, with many not adhering to recommended guidelines for item development and only about half reporting validity statistics[58]. Moreover, the rapid evolution of digital platforms continually outdates older cyberbullying scales that may not account for newer methods of cyberbullying[59].

These measurement challenges are intensified by the need to consider developmental stages. Children, adolescents and adults can experience and interpret cyberbullying in fundamentally different ways due to their developmental cognitive and social capacities[60]. For example, younger children may lack emotional maturity to accurately identify cyberbullying incidents[61], whereas adolescents, as they become more integrated with society, may both experience it more and also be able to identify it[60]. Adults, on the other hand, might interpret interactions differently based on life experiences and maturity, influencing their responses to potential cyberbullying scenarios[13,62]. This variability across age groups necessitates the synthesis of unique and common factors of cyberbullying to develop more robust cyberbullying measures and identify universally applicable predictors and consequences.

These complex issues within defining and measuring cyberbullying, combined with its potentially severe effects on victims, emphasize the importance of holistically synthesizing existing research. Thus, it is essential to better understand four major areas of research within cyberbullying: (1) sociodemographic and psychological profiles of victims, (2) various contextual and environmental predictors of cyberbullying victimization, (3) the consequences of cyberbullying victimization and (4) the efficacy of existing intervention programmes aimed at preventing cyberbullying.

## Sociodemographic and psychological predictors

One primary question in cyberbullying research revolves around identifying sociodemographic and psychological profiles of cyberbullying

victims. In terms of sociodemographic factors, research has shown that females and minorities (that is, racial and sexual) were more likely to be subjected to cyberbullying victimization[63–66]. Furthermore, personality traits, such as neuroticism and low agreeableness, can contribute to cyberbullying by affecting how individuals interact online and perceive hostile interactions[67]. Individuals with higher levels of anxiety, depression and anger are also more likely to become victims of cyberbullying[28,55,68], as they tend to be distanced from social groups and resort more to online media[69,70].

## Contextual and environmental predictors

Another important research question discussed within cyberbullying research concerns the contextual and environmental factors associated with cyberbullying victimization. Unregulated family and school climates, as well as unrestricted internet use, were prominent contextual risk factors associated with cyberbullying victimization[64,71–73]. Unregulated environments provide vulnerable targets and allow the unrestrained perpetration of cyberbullying in the absence of parental guardians or teachers, consistent with the Routine Activity Theory—deviant behaviours such as cyberbullying occur in the presence of motivated offenders, suitable targets and an absence of capable guardians[74–76]. The Routine Activity Theory suggests that the lack of effective supervision increases the opportunity for cyberbullying, emphasizing the importance of considering environmental factors as a predictor of cyberbullying victimization.

## Psychological and behavioural consequences

Third, another essential question within cyberbullying literature pertains to understanding the consequences of online victimization[46,77–80]. Mental health problems such as depression, anxiety and suicidal ideation are commonly identified as psychological consequences of cyberbullying victimization[36,81–83]. Research indicates that this relationship between psychological problems and cyberbullying victimization is bidirectional, as individuals with pre-existing conditions are more vulnerable to cyberbullying, which in turn exacerbates their symptoms[84]. Furthermore, these psychological consequences can snowball into behavioural consequences as well. Cyberbullying victims show lower school attendance, academic achievement[28,85] and worse peer relationships[82] and tend to engage more in both traditional and cyberbullying perpetration[36,86]. This is in line with the General Strain Theory, as the negative emotional strain caused by being cyberbullied may lead individuals to engage in deviant acts such as bullying, especially in the anonymized cyberspace[18,87,88].

## Effectiveness of interventions

Another key question frequently explored in cyberbullying research concerns the effectiveness of interventions specifically designed to prevent cyberbullying. Presently, many intervention programmes focus on educating individuals about cyberbullying and equipping them with coping strategies to handle its risk factors[89,90]. Additionally, some studies highlight various programme types that incorporate digital interventions[89] and emphasize the involvement of specific social groups, such as families[91]. However, the effectiveness of these anti-cyberbullying programmes remains uncertain, as indicated by previous reviews that report mixed results[26,92].

## The current review

Despite the extensive investigations into factors linked with cyberbullying victimization and the consolidation of predictors and outcomes through meta-analytic studies, there is a lack of comprehensive synthesis of these meta-analyses. While many predictors and consequences are associated with cybervictimization, most of the existing meta-analyses focus on assessing a single factor's relationship with cyberbullying[93,94]. For example, Barlett and Coyne[95] solely examined age as a risk factor associated with cybervictimization, while Sun and Fan[66] solely focused on the association between gender and cyberbullying victimization. Considering that being a victim of cyberbullying is usually the result of a combination of risk factors rather than one individual factor and that cyberbullying can lead to a diverse range of effects[80], it is pertinent to combine the various meta-analyses and gain a holistic understanding of the interconnectedness between the risks and outcomes of cyberbullying victimization.

Thus, the current work aims to conduct a systematic review of meta-analyses on potential predictors and consequences associated with cyberbullying victimization. Using a systematic review methodology will offer the opportunity to examine a broad scope of factors investigated by scholars and consider whether there is consensus in the field[96,97]. Specifically, this review will address the following critical questions: (1) What are the crucial sociodemographic and psychological profiles of cyberbullying victims? (2) What critical contextual and environmental factors are associated with cyberbullying victimization? (3) What are the key psychological and behavioural consequences of cyberbullying for victims? (4) How effective are existing interventions in mitigating the impacts of cyberbullying? By summarizing the associations reported in meta-analyses, this review aims to provide a clearer picture regarding the phenomena of cyberbullying victimization.

## Results

### Search outcome and eligibility

As illustrated in the Preferred Reporting Items for Systematic Reviews and Meta-Analyse (PRISMA) flowchart (Fig. 8), the initial search returned 1,583 records, of which 1,149 remained after the removal of duplicates. Title and abstract screening resulted in the removal of a further 818 records. Full text-screening resulted in the removal of 331 records, leaving a final total of 56 records[16,25,26,28,35–38,64–67,73,82,85,86,89–93,94,98–132], covering all regions (see Fig. 6 for full details). The characteristics of the 56 included meta-analyses are presented in Table 1 (see Fig. 1 for more descriptive statistics).

### Quality of included records

Based on the Joanna Briggs Institute (JBI) Critical Appraisal Instrument for Systematic Reviews and Research Syntheses tool, methodological quality scores for included records ranged from 6 to 11 (median 9; see Table 2 for a breakdown of the quality appraisal scores by record and Fig. 2 for a breakdown of the quality appraisal scores by criteria). As all 56 records had at least six 'yes' responses, it was concluded that there was no discernible methodological bias within any of the included meta-analyses. Of note, 53.57% ($n = 30$) of the included meta-analyses did not conduct quality appraisal of their constituent empirical studies, and 46.43% of the included meta-analyses ($n = 26$) did not use an adequate breadth of sources within their search strategy (for example, did not search for unpublished literature).

### Overall results

**Predictors of cyberbullying victimization.** In total, 39 out of the 56 included records included effect sizes on the relationship between cyberbullying victimization and its predictors.

*Sociodemographic and personality predictors.* A total of 13 meta-analyses explored sociodemographic and personality factors associated with cyberbullying victimization (Fig. 3a). Within them, ten meta-analyses examined sociodemographic factors, including age, gender, minority status and socioeconomic background. Six out of the seven meta-analyses focusing on age—all of which focused on children, adolescents or college-aged samples—indicated that age denoted a higher risk of becoming a cyberbullying victim (median $r = 0.07$, range −0.02 to 0.40). Furthermore, 8 out of 11 meta-analyses indicated that females were more likely than males to be victims of cyberbullying victimization (median $r = 0.04$, range −0.14 to 0.27). With regard to marital status, no significant effect was observed across the

**Table 1 | Characteristics of the 56 included meta-analyses**

| Author(s) and citation | Document type | Country, region | Number of studies | Sample size | Participants | Number of sources searched | Date range of search | Date range of included studies | Research objective |
|---|---|---|---|---|---|---|---|---|---|
| Abregú-Crespo et al.[98] | Journal article | Europe and North America (73%) | 212 | 631,523 | Children and adolescents (aged 4–17 years) with neurodevelopmental or psychiatric conditions | 6 databases (ERIC, PsycArticles, PsycInfo, Psychology and Behavioural Sciences Collection, PubMed and Web of Science Core Collection) | Up to August 2023 | NA | Assess the odds of bullying involvement and its association with mental health measures in these populations |
| Barlett et al.[99] | Journal article | 62 countries (Africa, Asia–Pacific, Australia, Europe, Middle East, North America and South/Latin America) | 211 | 1,136,080 | Youth and adult populations (mean age 9.00–37.04 years) | 13 Databases (Academic Search Complete, Business Source Complete, Communication and Mass Media Complete, Criminal Justice Abstracts, Education Research Complete, Family Studies Abstracts, HealthSource: Nursing/Academic Edition, Human Resources Abstracts, MEDLINE, PsycArticles, PsycInfo, SocINDEX and Social Sciences Full Text) and 15 journals (Aggressive Behavior; British Journal of Developmental Psychology; Computers in Human Behavior; Cyberpsychology, Behavior, and Social Networking; Developmental Psychology; European Journal of Developmental Psychology; Journal of Adolescence; Journal of School of Adolescent Health; Journal of School Psychology; Journal of School Violence; Journal of Youth and Adolescence; New Media and Society; School Psychology International; School Psychology Quarterly; School Psychology Review) | NA | 2007–2022 | Examine the correlations between cyberbullying and other variables while statistically controlling for traditional bullying |
| Caridade and Braga[64] | Journal article | 4 countries (Europe and North America) | 16 | 12,760 | Adolescents and young adults (mean age 10–26 years) | 9 databases (Academic Research Complete, Business Source Complete, Complementary Index, EBSCOhost, ERIC, Psychology and Behavioral Sciences Collection, PubMed, Science Direct, Scopus and Social Sciences Citation), reference lists of review articles and direct referrals by authors | Up to February 2019 | 2014–2018 | Identify risk and protective factors associated with youth cyber-dating abuse |
| Chen et al.[73] | Journal article | Asia–Pacific, Europe and North America | 81 | 99,741 | Children, adolescents and adults (age range not included) | 8 databases (Communication and Mass Media Complete, EBSCO, ERIC, MEDLINE, Nursing and Allied Health Source, PsycInfo, PubMed and Web of Science Direct) and reference lists of reviews | NA | 2004–2015 | Systematically examine the predictors of cyberbullying from a social cognitive and media effects approach |
| Chen et al.[89] | Journal article | 12 countries (Asia–Pacific, Australia, Europe and North America) | 18 | 55,473 | Primary school to college students (age range not included) | 6 databases (EMBASE, ERIC, MEDLINE, PsycInfo, Social Service Abstracts, Sociological Abstracts) | Up to January 2021 | 2000–2018 | Examine and compare the effectiveness of digital health interventions in reducing bullying and cyberbullying |
| Chen et al.[100] | Journal article | 25 countries (Africa, Asia–Pacific, Australia, Europe, Middle East, North America and South/Latin America) | 39 | 128,097 | Adolescents (aged 10–18 years) without special attributes (for example, clinical populations) | 9 databases (ERIC, PsycArticles, PsycInfo, Psychology and Behavioral Sciences Collection, PubMed, ScienceDirect, Scopus, Web of Science and Wiley Online Library) | Up to April 2022 | 1993–2021 | Examine the relation between experiences of bullying and victimization and life satisfaction among adolescents |

**Table 1 (continued) | Characteristics of the 56 included meta-analyses**

| Author(s) and citation | Document type | Country, region | Number of studies | Sample size | Participants | Number of sources searched | Date range of search | Date range of included studies | Research objective |
|---|---|---|---|---|---|---|---|---|---|
| Christina et al.[101] | Journal article | 13 countries (Africa, Asia–Pacific, Europe and North America) | 85 | 117,520 | School-aged participants (younger children and young adults excluded) (mean age 6.3–16 years) | 5 databases (Academic Search Premier, MEDLINE, PsychInfo, Scopus and Web of Science) | Up to May 2020 | 1995–2020 | Examine the bidirectional effects between internalizing problems and peer victimization within a meta-analytic framework |
| Doty et al.[102] | Journal article | 11 countries (Australia, Europe, Middle East and North America) | 30 | 48,548 | Youth (aged up to 24 years) | 7 databases (Academic Search Premier, Compendex, ERIC, PsychInfo, Psychology and Behavioural Sciences Collection, PubMed and Web of Science) | 2010–2019 | 2011–2019 | Describe cyberbullying preventative interventions in relation to intervention characteristics and risk of bias; qualitatively illustrate the dosage, modalities and contexts of existing cyberbullying preventative interventions using harvest plots; and quantitatively examine the effect of cyberbullying preventative interventions on perpetration and victimization by dosage, modalities and contexts |
| Eberle[103] | Dissertation/ thesis | United States (North America) | 8 | 10,509 | Minors (aged below 18 years) and adults (age range not included) | 7 databases (ERIC, PAIS Index, ProQuest One Academic, PsycArticles, PsycInfo, PTSDpubs and Sociological Abstracts) | Up to August 2023 | 2007–2022 | Conduct a meta-analysis to identify the risk and protective markers of sextortion to identify points of prevention and intervention |
| Erbiçer et al.[104] | Journal article | Turkey (Europe) | 59 | 34,068 | Children and young people (aged 10–16 years) | 8 databases (DergiPark, ERIC, National Thesis Center of Türkiye, ProQuest, PubMed, Scopus, Turkish Psychiatry Index and Web of Science) | Up to December 2020 | 2012–2020 | Conduct a systematic review and meta-analysis of the studies conducted in Turkey to clarify risk and protective factors and outcomes of cyberbullying perpetration and victimization |
| Fisher et al.[36] | Journal article | NA | 55 | 257,678 | Adolescents (aged 12–18 years) | 9 databases (ERIC, International Bibliography of the Social Sciences, ProQuest Central, ProQuest Dissertations and Theses Full Text, ProQuest Dissertations and Theses: UK and Ireland, PsycArticles, PsycInfo, Social Services Abstracts and Sociological Abstracts) | Up to 2016 | 2006–2016 | Synthesize existing literature on the relationship between peer cybervictimization and adolescents' internalizing and externalizing problems |

**Table 1 (continued) | Characteristics of the 56 included meta-analyses**

| Author(s) and citation | Document type | Country, region | Participants | Sample size | Number of studies | Number of sources searched | Date range of search | Date range of included studies | Research objective |
|---|---|---|---|---|---|---|---|---|---|
| Gaffney et al.[105] | Journal article | 10 countries (Australia, Europe and North America) | School-aged participants (aged 4–18 years) | 36,534 | 18 | Several databases (including DARE, ERIC, Google Scholar, PsycArticle, PsycInfo, Scopus, Web of Science), several journals (including Behavior and Social Networking, Computers in Human Behavior and Cyberpsychology) | 2000–2018 | 2012–2018 | Evaluate the effectiveness of existing cyberbullying intervention and prevention programmes |
| García-Hermoso et al.[106] | Journal article | 89 countries (Africa, Arab States, Asia-Pacific, Australia, Europe, Middle East, North America and South/ Latin America) | Children and young people (aged 10–16 years) | 386,740 | 18 | 4 databases (EMBASE, ERIC, PsycArticles and PubMed) and reference lists of retrieved articles | NA | 2007–2020 | Provide a quantitative analysis on the associations of physical activity and sedentary behaviour on bullying victimization among children and adolescents |
| Gardella et al.[85] | Journal article | United States (North America) | Adolescents (aged 12–17 years) | 26,906 | 25 | 10 databases (Dissertations and Theses at Vanderbilt University, ERIC, International Bibliography of the Social Sciences, ProQuest Central, ProQuest Dissertations and Theses: UK and Ireland, ProQuest Dissertations and Theses Full Text, PsycArticles, PsycInfo, Social Services Abstracts and Sociological Abstracts), conference proceedings, sites with technical reports and forward citation searches | 1985–2015 | 2009–2014 | Quantitatively synthesize relationships between peer cybervictimization and educational outcomes |
| Gilbar et al.[107] | Journal article | 10 countries (Africa, Australia, Europe, Middle East, North America and South/ Latin America) | Adults (aged above 18 years, mean age 18.06–44.80 years) | 27,491 | 44 | 7 databases (APA PsycArticles, Criminal Justice Abstracts, EBSCO, ERIC, MEDLINE, PsycInfo, Social Sciences Full Text (H.W. Wilson) and Web of Science) | Up to 2019 | 2010–2020 | Conduct an in-depth examination of the possibility of sex differences in different types of cyber-intimate partner violence, whether there are cyber-intimate partner violence associations to face-to-face intimate partner violence and an exploration of sex differences in the associations between cyber-intimate partner violence and face-to-face intimate partner violence among adults |
| Gini et al.[108] | Journal article | Australia, Europe and North America | Children and adolescents (mean age 12.05–16 years) | 90,877 | 19 | 8 databases (Dissertation Abstracts International, Open Access, ProQuest Dissertations and Theses Open, PsychInfo, Pubmed, Scopus and Web of Science), Google, reference sections of review, reference sections of collected articles, conference proceedings of the last biannual meetings of the Society for Research in Child Development and the Society for Research on Adolescence, and of the last four 'Workshop Aggression' (held in Europe) | Up to 2015 | 2009–2015 | Summarize the relations of traditional and cybervictimization with internalizing problems and identify whether these types of peer victimization were differentially related to such problems |

 

**Table 1 (continued) | Characteristics of the 56 included meta-analyses**

| Author(s) and citation | Document type | Country, region | Number of studies | Sample size | Participants | Number of sources searched | Date range of search | Date range of included studies | Research objective |
|---|---|---|---|---|---|---|---|---|---|
| Guo[37] | Journal article | Asia–Pacific, Australia, Europe, Middle East and North America | 77 | 129,278 | Juveniles and young adults in school settings (aged 9–24 years) | 7 databases (ERIC, Digital Dissertations, Google Scholar, National Criminal Justice Reference System, PsychInfo, PubMed and Web of Science), reference sections of review articles and direct referrals by authors | NA | 2004–2013 | Examine the relative magnitude of demographic, individual and contextual predictors of cyberbullying perpetration and victimization |
| Heerde and Hemphill[109] | Journal article | 22 countries (Asia–Pacific, Australia, Europe, Middle East and North America) | 27 | 156,284 | Adolescents (aged 11–19 years) | 14 databases (Australian Criminology Database, Australian Criminology Database—Health Subset, CINAHL, ERIC, Health Source, MEDLINE, Nursing/Academic Edition, ProQuest Criminal Justive, ProQuest Education Journals, ProQuest Psychology Journals, ProQuest Social Science Journals, PschInfo, Psychology and Behavioural Sciences Collection and Social Work Abstracts, SocIndex) | 1990–2018 | 2008–2017 | Consolidate studies investigating associations between bullying (traditional and cyberbullying perpetration and victimization) and deliberate self-harm in youth using a meta-analytic approach |
| Hu et al.[110] | Journal article | 13 countries (Africa, Asia–Pacific, Australia, Europe, North America and South/Latin America) | 57 | 53,183 | Minors (aged below 18 years) and adults (age range not included) | 8 databases (Chinese CNKI Database, EBSCO, Google Scholar, ProQuest Dissertations, PsycInfo, Springer, Wanfang database and Web of Science) | 2099–2022 | 2002–2022 | Examine the reliability of the effect size and a series of moderating effects between gender non-conformity and victimization |
| Hu et al.[93] | Journal article | 17 countries (Asia–Pacific, Europe, North America) | 57 | 105,440 | Adolescents and young adults (mean age 10.85–24.67 years) | 4 databases (APA PsycNet, Google Scholar, PsycInfo and PubMed) | Up to 2021 | 2009–2021 | Examine the effect of cyberbullying and victimization on depression |
| Huang et al.[111] | Journal article | 15 countries (Arab States, Asia–Pacific, Australia, Europe, North America and South/Latin America) | 26 | 73,191 | Children, adolescents, and adults (aged 11–85 years) | 11 databases (Chinese CNKI Database, Cochrane Library, EMBASE, EBSCO, ERIC, MEDLINE, PsycInfo, PubMed, Scopus, Wanfang Database and Web of Science), reference list of included studies and review articles | Up to November 2022 | 2020–2022 | Examine the effect of the coronavirus disease 2019 pandemic on cyberbullying, estimate the global cyberbullying prevalence and explore factors related to cyberbullying during the coronavirus disease 2019 pandemic |
| John et al.[35] | Journal article | 9 countries (Asia–Pacific, Australia, Europe and North America) | 23 | 156,384 | Adolescents and young adults (aged below 25 years, mean age 12.5–20 years) | 6 databases (Cochrane Library, Medical Literature Analysis and Retrieval System Online, PROSPERO, PsycInfo, PubMed and Scopus) health improvement sources (for example, Health Evidence Canada), topic-specific websites (for example, American Association of Suicidology), meta-search engines (Google) and direct referrals by authors | January 1996 to February 2017 | 2009–2016 | Systematically review and meta-analyse, when possible, the current evidence examining the association between cyberbullying involvement (as victim, perpetrator, or both) and self-harm and suicidal behaviours in children and young people (younger than 25 years) |

**Table 1 (continued) | Characteristics of the 56 included meta-analyses**

| Author(s) and citation | Document type | Country, region | Number of studies | Sample size | Participants | Number of sources searched | Date range of search | Date range of included studies | Research objective |
|---|---|---|---|---|---|---|---|---|---|
| Kamaruddin et al.[16] | Journal article | 2 countries (Australia and Thailand) | 2 | 2,954 | Non-school-aged children (aged 8–29 years) | 10 databases (Cambridge Journal Online, EBSCOHOST, ERIC, IEEE XPLORE, Oxford Journal Online, ProQuest Dissertations and Theses, PubMed (MEDLINE), Science Direct, Scopus and SpringerLink) | January 1995 to February 2022 | 2013–2022 | Empirically determine the effectiveness of programmes with non-school-aged samples with a specific focus on studies conducted within the Asia-Pacific region |
| Killer et al.[112] | Journal article | 13 countries (Asia-Pacific, Australia, Europe and North America) | 49 | 43,809 | Children and adolescents (aged 7–19 years) | 3 databases (Google Scholar, ProQuest Dissertations and Theses Global, and PsycInfo) | Up to 2018 | 2007–2018 | Examine the relationship between moral disengagement and key bullying roles (that is, defender, bystander and victim) |
| Kowalski et al.[28] | Journal article | Asia-Pacific, Australia, Europe and North America | 131 | 1,519 to 164,280 | School and college students (aged 6–88 years) | 14 databases, 15 journals, reference sections of review papers, direct referrals by authors | Up to 2012 | 2004–2013 | Synthesize the relationships among cyberbullying, cybervictimization and meaningful behavioural and psychological variables |
| Lan et al.[90] | Journal article | 7 countries (Australia and Europe) | 19 | 31,924 | Adolescents (aged 10–19 years) | 18 databases (Academic Search Complete, Australian Education Index, British Education Index, Business Source Complete, CINAHL Plus, Cochrane Library, Communication and Mass Media Complete, Criminal Justice Abstracts, Education Database, ERIC, Google Scholar, MEDLINE, PsycArticles, PsycInfo, PubMed, Scopus, Sociological Abstracts and Web of Science) | Up to March 2020 | 2012–2019 | Examine the effectiveness of anti-cyberbullying educational programmes in reducing cyber aggression and cyber victimization |
| Li et al.[113] | Journal article | Asia-Pacific, Australia, Europe, Middle East and North America | 42 | 266,888 | Children and youth (aged 8–10 years) | 3 databases (PsycInfo, PubMed and Web of Science) | 2010–2021 | 2010–2021 | Explore the prevalence trends of traditional bullying and cyberbullying for the past decade |
| Li et al.[114] | Journal article | 9 countries (Asia-Pacific, Europe, North America and South/Latin America) | 57 | 98,351 | Children, adolescents and young adults (age range not included) | 5 databases (Chinese CNKI Database, EBSCO-ASP, PubMed, Web of Science and Wiley Online Library), 9 journals (Computers in Human Behavior; Cyberpsychology, Behavior, and Social Networking; Child Abuse and Neglect; Journal of Interpersonal Violence; Journal of Child and Adolescent Trauma; Journal of Child Sexual Abuse; Child Maltreatment; Journal of Family Violence; and Chinese Mental Health Journal) and a reference list of included studies, reviews and meta-analyses | Up to May 2023 | 2016–2023 | Examine the extent to which childhood maltreatment is correlated with cyberbullying (perpetration and victimization) and whether these associations varied by sample, publication and research design characteristics |
| López-Barranco et al.[115] | Journal article | 4 countries (Europe, North America and South/Latin America) | 12 | 47,104 | Adolescents (aged 13–18 years) and young adults (aged 19–24 years) | 5 databases (Gender Studies, PsycInfo, PubMed, Scopus and Web of Science), reference list of shortlisted articles and contacting authors of articles of interest | January 2015 to January 2021 | 2015–2020 | Analyse the different types of violence perpetrated and experienced in dating relationships as a function of gender in adolescents and young adults |

**Table 1 (continued) | Characteristics of the 56 included meta-analyses**

| Author(s) and citation | Document type | Country, region | Number of studies | Sample size | Participants | Number of sources searched | Date range of search | Date range of included studies | Research objective |
|---|---|---|---|---|---|---|---|---|---|
| Lozano-Blasco et al.[116] | Journal article | 5 countries (Asia–Pacific, Australia and Europe) | 9 | 29,093 | Adolescents (aged 11.5–18 years) | 3 databases (PsycInfo, Science Direct and WOS) | 2015–2020 | 2017–2020 | Analyse the influence of family communication on cybervictims and the moderating role of different sociodemographic variables (age, gender, nationality and culture), as well as social, emotional and personality variables |
| Lozano-Blasco et al.[117] | Journal article | 11 countries (Asia–Pacific, Australia, Europe, North America and South/Latin America) | 32 | 238,977 | Adolescents (aged 10–19 years; mean age 13.68 years) | 4 databases (Scopus, PsycInfo, Science Direct and PubMed) | 2013–2019 | 2013–2019 | Investigate the significance of sex and age differences in cybervictimization |
| Lozano-Blasco et al.[86] | Journal article | 14 countries (Africa, Asia–Pacific, Europe, Middle East, North America and South/Latin America) | 22 | 47,836 | Adolescents (mean age range 11.72–16.5 years, mean age 14.64 years) | 3 databases (PsycInfo, Science Direct and Scopus) | 2014–2019 | 2014–2019 | Examine whether it was possible for someone to be both a cybervictim and a cyberbully |
| Marciano et al.[82] | Journal article | 13 countries (Asia–Pacific, Australia, Europe and North America) | 56 | 40,682 | Children and adolescents (mean age 10.5–16.7 years, mean age 13.4 years) | 13 databases (CENTRAL, CINAHL, Communication and Mass Media Complete, EMBASE, ERIC, Google Scholar, MEDLINE, ProQuest Dissertations and Theses, ProQuest Sociology, PsycArticles, PsycInfo, Psychology and Behavioral Sciences Collection, and Web of Science) | Up to 2018 | 2007–2017 | Quantitatively summarize exclusively longitudinal studies on the causes and consequences of cyberbullying perpetration and cybervictimization |
| Mills et al.[38] | Journal article | 4 countries (Asia–Pacific, Europe and North America) | 10 | 421 | Children, adolescents, and young adults (aged 8–25 years) | 4 databases (ProQuest, PubMed, Scopus and Web of Science) | 2002–2022 | 2010–2021 | Investigate the impacts of cyberbullying-like behaviours on psychophysiology, using electroencephalography as the measurement method via a meta-analysis |
| Modecki et al.[118] | Journal article | NA | 80 | 335,519 | Adolescents (aged 12–18 years) | 6 databases (Educational Resources Information Centre, Google Scholar, Proquest Dissertations and Theses, PsycInfo, PubMed and Scopus) and reference lists of eligible articles | No limit | 2004–2013 | Conduct a meta-analysis on the prevalence of bullying across cyber and traditional contexts among adolescents |
| Molero et al.[119] | Journal article | 6 countries (Asia–Pacific, Europe and North America) | 13 | 7,348 | Adolescents and young adults (aged 8–37 years) | 3 databases (PsycInfo, Scopus and Web of Science) | 2011–2021 | 2014–2020 | Analyse the relationship between cybervictimization, anxiety and depression in an adolescent population through a meta-analysis |

**Table 1 (continued)| Characteristics of the 56 included meta-analyses**

| Author(s) and citation | Document type | Country, region | Number of studies | Sample size | Participants | Number of sources searched | Date range of search | Date range of included studies | Research objective |
|---|---|---|---|---|---|---|---|---|---|
| Nesi et al.[120] | Journal article | 14 countries (Asia–Pacific, North America and South/Latin America) | 61 | 532 to 135,424 | Adolescents and adults (aged 11–35 years) | 3 databases (CINAHL, MEDLINE and PsycInfo) | Up to August, 2020 | 2010–2020 | Provide an overview of the current research and to examine associations between different aspects of social media use and self-injurious thoughts and behaviours |
| Ng et al.[25] | Journal article | 10 countries (Africa, Asia–Pacific, Australia, Europe, North America and South/Latin America) | 15 | 35,694 | Adolescents (aged 10–18 years) | 6 databases (Cumulative Index to Nursing and Allied Health Literature, EMBASE, Google Scholar, ProQuest Dissertations and Theses, PsycInfo and PubMed) and reference lists of review articles and eligible articles | Up to 2019 | 2000–2018 | Examine the effectiveness of anti-bullying educational interventions at reducing the frequencies of traditional bullying or cyberbullying and cybervictimization among adolescents |
| Oblad[65] | Dissertation/thesis | 27 countries (Asia–Pacific, Australia, Europe and North America) | 24 | 42,118 | NA | 4 databases (PsycInfo, Interlibrary Loan, Internet accessible databases (for example, Google Scholar) and direct referrals by authors in previous reviews) | 2000–2012 | 2004–2011 | Compare gender differences in cyberbullying and victimization |
| Polanin et al.,[26] | Journal article | United States and non-United States | 50 | 45,371 | School students (mean age 13 years) | 15 databases (Academic Search Complete, CrimeDoc, Education Full Text, ERIC, Grey Literature Database (Canadian), National Criminal Justice Reference Service Abstracts, ProQuest Criminal Justice, ProQuest Dissertations and Theses, ProQuest Education Journals, ProQuest Social Science Journals, PsycInfo, PubMed (MEDLINE), Social Care Online (United Kingdom), Social Sciences Abstracts and Social Science Research Network eLibrary), 5 journals (Aggressive Behavior, Child Development, Computers in Human Behavior, Journal of Interpersonal Violence, and Prevention Science), reference lists of eligible articles and records that cited eligible articles | 1995–2005 | 2004–2019 | Conduct a systematic review and meta-analysis that synthesized the effects of school-based programmes on cyberbullying perpetration or victimization outcomes |
| Pratt et al.[121] | Journal article | 30 countries (Asia–Pacific, Australia, Europe and North America) | 66 | 102,716 | Minors (aged below 18 years) and adults (age range not included) | 2 databases (Google Scholar and National Criminal Justice Reference Service) | Up to November 2012 | 1995–2014 | Conduct a meta-analysis on self-control and victimization |
| Resett and Mesurado[122] | Book chapter | Europe, North America and South/Latin America | 8 | 7,627 | Adolescents (aged 10–19 years) | 8 databases (Dialnet, EBSCO Host, JSTOR, Latindex, SciELO, ScienceDirect, NCBI and PsycInfo) | 2000 to July 2018 | 2000–2018 | Analyse the effectiveness of bullying and cyberbullying interventions in adolescents aged 10–19 years, published between 2000 and July 2018 inclusive, in English, Spanish and Portuguese |

**Table 1 (continued) | Characteristics of the 56 included meta-analyses**

| Author(s) and citation | Document type | Country, region | Number of studies | Sample size | Participants | Number of sources searched | Date range of search | Date range of included studies | Research objective |
|---|---|---|---|---|---|---|---|---|---|
| Sarier[123] | Journal article | Turkey (Europe) | 37 | 21,768 | Secondary and high school students (age range not included) | 2 databases (DergiPark and YOK thesis database) | Up to January 2020 | 2010–2019 | Combine the results of studies that reveal the relationship between cyberbullying and victimization with different demographic variables in Turkey using the meta-analysis method |
| Sun and Fan[66] | Journal article | Asia–Pacific, Europe and North America | 40 | 71,722 | NA | 8 databases (Academic Search Primer, Business Source Primer, Communication Source, ERIC, Google Scholar, PsycArticles, PsycInfo and PubMed) | Up to October 2013 | 2006–2013 | Examine what is the general gender group difference in cybervictimization as reported in the existing empirical studies |
| Tran et al.[124] | Journal article | 7 countries (Asia–Pacific, Australia, Europe and North America) | 17 | 79,202 | Adolescents (aged 10–19 years) | 2 databases (EMBASE and PubMed), reference sections of reviews and references that cited eligible studies | Up to 2021 | 2007–2020 | Investigate the relationship between cybervictimization and depression in adolescents |
| Van Cleemput et al.[125] | Journal article | 8 countries (Asia–Pacific, Australia, Europe and North America) | 8 | 11,921 | Adolescents (aged 10–18 years) | 11 databases (Arts and Humanities Index, Communication Abstracts, Conference Proceedings Index—Social Science and Humanities, ERIC, Google Scholar, MEDLINE, PsycInfo, Social Sciences Index, Social Services Abstracts, Sociological Abstracts and Web of Science), contacting researchers through personal connections and research networks | January 2003 to September 2014 | 2006–2013 | Conduct a systematic review and meta-analyses of cyberbullying prevention programs |
| van Geel et al.[126] | Journal article | NA | 34 | 284,375 | Children and adolescents (aged 9–21 years) | 3 databases (OvidMEDLINE, PsycInfo and Web of Science) and reference sections of review articles | January 1910 to January 2013 | 1999–2012 | Examine the relationship between peer victimization and suicidal ideation or suicide attempts in children and adolescents |
| Walters[127] | Journal article | 10 countries (Asia–Pacific, Australia, Europe and North America) | 22 | 1,048 | Samples aged below 18 years (mean age 9.5–15 years) | 12 databases (Academic Search Complete, Criminal Justice Abstracts, Dissertation Abstracts, ERIC, HeinOnline, JSTOR Journals, PsycArticles, Psychology and Behavior Sciences Collection, PsycInfo, Social Sciences Citation Index, SocIndex and Sociological Collection) | Up to April 2019 | 2008–2018 | Gauge the magnitude of the relationship between concurrent victimization and perpetration, assess the temporal direction of the association between victimization and perpetration, determine whether meaningful effect size differences exist between traditional bullying/victimization and cyberbullying/ victimization and whether the effect extends across types and investigate the effect of four moderator variables |

**Table 1 (continued) | Characteristics of the 56 included meta-analyses**

| Author(s) and citation | Document type | Country, region | Number of studies | Sample size | Participants | Number of sources searched | Date range of search | Date range of included studies | Research objective |
|---|---|---|---|---|---|---|---|---|---|
| Wang and Jiang[91] | Journal article | 8 countries (Australia, Europe and Middle East) | 11 | 29,859 | Adolescents (aged 10–19 years) | 7 databases (EBSCO, ERIC, ProQuest Dissertations and Theses, PsycInfo, PubMed, Scopus and Web of Science) | Up to May 2021 | 2012–2021 | Investigate the effectiveness of parent-related programmes in reducing the frequency of cyberbullying perpetration and victimization among adolescents |
| Wirth[128] | Dissertation/thesis | 8 countries (Australia, Europe and North America) | 12 | 32,004 | Children (aged 10–19 years) | 7 databases (CINAHL, EMBASE, ERIC, Informit, PsycInfo, Pubmed, CINAHL and Scopus) | Up to 2018 | 2010–2016 | Produce a systematic review and meta-analysis of the existing literature to discover whether cyberbullying intervention programmes are effective at reducing perpetration and victimization |
| Wissink et al.[67] | Journal article | 11 countries (Africa, Asia–Pacific, Australia, Europe and North America) | 48 | 109,402 | Juveniles (mean age 12–23 years) | 4 databases (ERIC, Google Scholar, PsycInfo and Web of Science) and reference list of included studies | Up to May 2019 | 2006–2019 | Identify risk factors for cyberstalking, hacking and sexting perpetrated by juveniles |
| Wong[129] | Dissertation/thesis | 27 countries (Africa, Asia–Pacific, Australia, Europe, Middle East, North America and South/Latin America) | 52 | 59,294 | Adolescents and emerging adults (aged below 30 years; mean age 12.08–24.14 years) | 5 databases (ERIC, Google Scholar, PsycInfo and Web of Science), conference websites | 2017–2020 | 2017–2020 | Conduct a meta-analysis to study the moderation of age in cyberbullying with a focus on the interaction across different aspects of information and communication technology affordances on social cognitive development, dominant developmental environments and developmental goals across adolescence and emerging adulthood |
| Yuchang et al.[130] | Journal article | Asia–Pacific, Europe and North America | 56 | 214,819 | Children and adolescents (aged 10–19 years) | 4 databases (China National Knowledge Infrastructure, Google Scholar, PsycArticles and PsycInfo) and reference section of review articles | NA | 1998–2016 | Examine cross-cultural perspectives to explore whether there are any differences between the effects of cybervictimization and traditional victimization on the presence of depression and anxiety in children and adolescents |

**Table 1 (continued) | Characteristics of the 56 included meta-analyses**

| Author(s) and citation | Document type | Country, region | Sample size | Number of studies | Participants | Number of sources searched | Date range of search | Date range of included studies | Research objective |
|---|---|---|---|---|---|---|---|---|---|
| Zhang and Chen[131] | Journal article | 11 countries (Asia–Pacific, Australia, Europe and North America) | 23,438 | 24 | School or university students (mean age 9.12–10.45 years) | 5 databases (Chinese CNKI Database, Google Scholar, ProQuest Dissertations and Theses, PubMed and Web of Science) | Up to March 2022 | 2010–2021 | Conduct a meta-analysis to evaluate the exact association between emotional intelligence and school bullying victimization |
| Zych et al.[94] | Journal article | 15 countries (Asia–Pacific, Europe and North America) | 25,268 | 25 | Children and adolescents (aged up to 18 years; age range 11.57–18 years) | 5 databases (Google Scholar, PsycInfo, PubMed, Scopus and Web of Science) | Up to November 2016 | 2009–2017 | Examine how empathy is related to different cyberbullying roles |
| Zych et al.[132] | Journal article | 4 countries (Africa, Europe and North America) | 55,445 | 23 | Children or adolescents (aged up to 21 years, age range not included) | 4 databases (Google Scholar, MEDLINE, Scopus and Web of Science) | Up to October 2016 | 2000–2016 | Assess whether involvement in bullying perpetration or victimization could be risk factors for perpetration or victimization in early romantic relationships |

'NA' is used in cases where the record did not provide the relevant information.

records (median $r = -0.01$, range $-0.09$ to 0.08). Across four of the five meta-analyses that examined minority status, members of both racial/ethnic and sexual minorities were more likely to become cyberbullying victims as compared with majority groups (Caucasians and heterosexuals, respectively) (median $r = 0.03$, range $-0.03$ to 0.20). Finally, two meta-analyses indicated that indicators of higher socioeconomic status (including parental education) were associated with higher exposure to cyberbullying victimization.

Positively valenced personality traits—such as agreeableness, extraversion and openness to experience—were associated with lower cyberbullying victimization (median $r = -0.09$, range $-0.18$ to $-0.06$), while negatively valenced personality traits—such as antisocial personality, dark personality traits, dominance and neuroticism—were associated with increased risk of exposure to cyberbullying victimization (median $r = 0.15$, range $-0.06$ to 0.23).

*Psychological predictors.* A total of 14 meta-analyses explored psychological factors predicting cyberbullying victimization (Fig. 3b). Across all meta-analyses, higher levels of mental health risk factors and behavioural problems were both associated with increased levels of cyberbullying victimization. Internalizing mental health problems associated with cyberbullying victimization included higher levels of anxiety, higher levels of depression, higher levels of moral disengagement and various psychiatric conditions, all of which were related to an increased tendency to be a victim of cyberbullying (median $r = 0.15$, range 0.07 to 0.38). All meta-analyses also indicated that high levels of externalizing problems, including anger and hostility, behavioural problems (including risky behaviours) and substance use, were positively related to cyberbullying victimization (median $r = 0.16$, range $-0.01$ to 0.57). In contrast, 16 out of 18 effect sizes indicated that positively valenced psychological factors such as emotional intelligence, better emotional management, empathy, higher self-control, higher self-efficacy, higher self-esteem and higher social intelligence served as protective factors against cyberbullying victimization (median $r = -0.06$, range $-0.22$ to 0.12).

*Contextual predictors.* A total of 29 meta-analyses reported various contextual predictors of cyberbullying victimization (Fig. 4a). Within them, 12 meta-analyses included parental and family relations as a contextual predictor. Overall, 13 out of 14 effect sizes indicated that a positive family environment was associated with lower levels of cyberbullying victimization. Higher levels of family support and parental monitoring, including parental control of technology, parental interaction, parental mediation and parental support, were also associated with lower risk of being subjected to cyberbullying victimization (median $r = -0.08$, range $-0.18$ to 0.01). In contrast, results from three meta-analyses indicated that unfavourable home environments, such as experiencing childhood maltreatment, offensive family communication or being part of single-parent households, were associated with increased exposure to cyberbullying victimization (median $r = 0.20$, range 0.16 to 0.24). Furthermore, three meta-analyses showed that being in an intimate relationship and characteristics of the relationship (including high violence perpetration and or/victimization within the relationship) were also associated with higher levels of cyberbullying victimization (median $r = 0.14$, range $-0.05$ to 0.44).

The association between school-related, peer-related and environmental factors and cyberbullying victimization was included in 11 meta-analyses (Fig. 4b). Five effect sizes indicated that negative school climates and lack of school safety were associated with higher cyberbullying victimization (median $r = 0.11$, range 0.01 to 0.22). Similarly, lower peer relationship quality, negative peer influence and being the perpetrator or victim of traditional peer bullying were associated with a higher cyberbullying victimization across all meta-analyses (median $r = 0.25$, range 0.09 to 0.49).

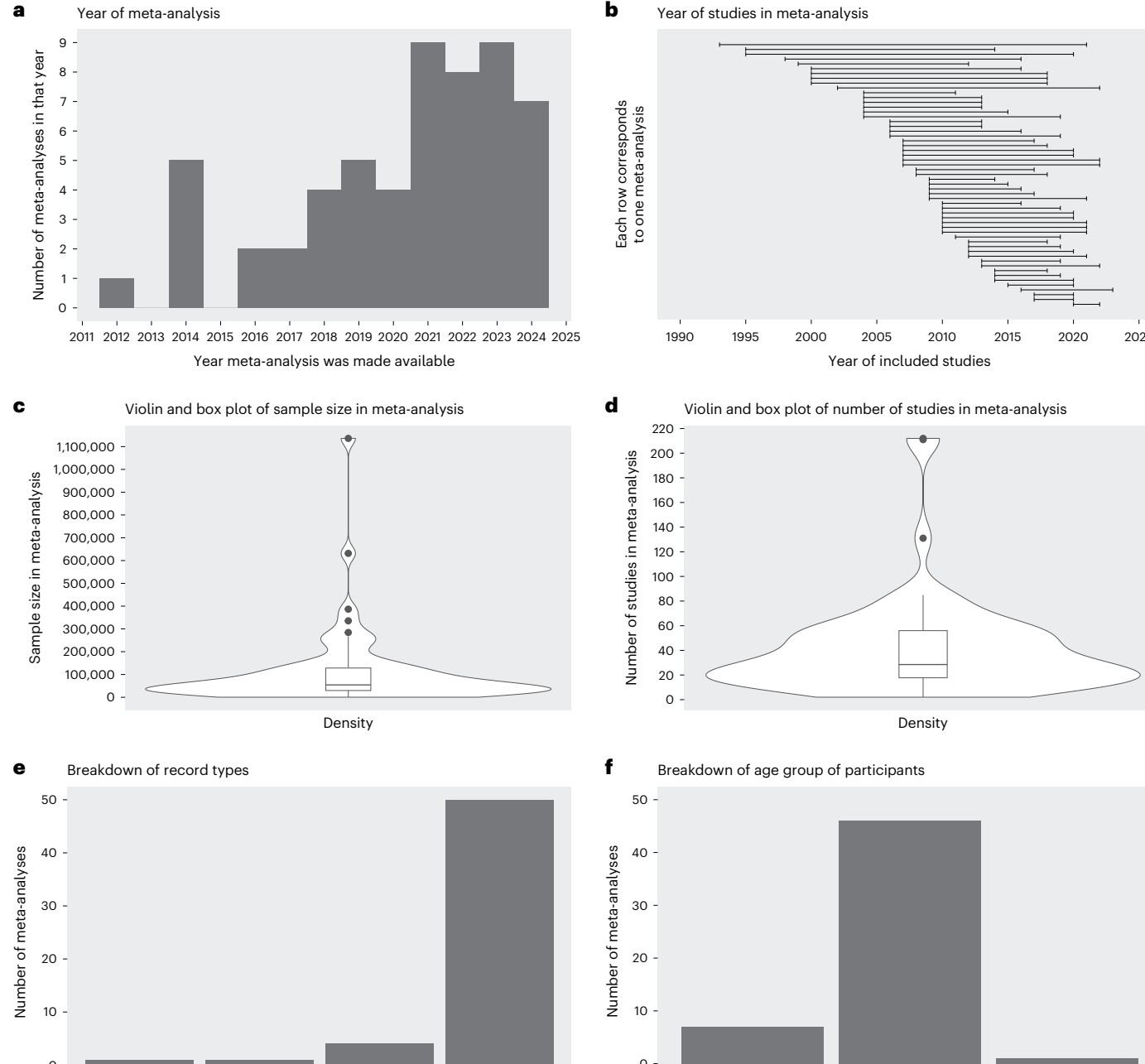

**Fig. 1 | Descriptive statistics of the 56 included meta-analyses. a**, The years in which the included meta-analyses were published or made available. **b**, The span of years of the studies included within the meta-analyses. **c**, A representation of the sample sizes across the included meta-analyses. The data were derived from the 56 meta-analyses included in the review (*n* = 56). The violin plot displays the density distribution of the data, while the overlaid box plot shows the median (54,314), interquartile range (98,724.75) and full range of the data (421–1,136,080). The black dots represent outliers, indicating sample sizes that deviate

noticeably from the rest of the data distribution. **d**, The spread of the number of studies within the included meta-analyses. The data were derived from 56 meta-analyses included in the review (*n* = 56). The violin plot displays the density distribution of the data, while the overlaid box plot shows the median (28.5), interquartile range (38.25) and full range of the data (2–212). The black dots represent outliers, indicating numbers of studies that deviate noticeably from the rest of the data distribution. **e**, The included meta-analyses by the type of publication. **f**, The age groups of the samples included within the meta-analyses.

Factors related to internet use were also common contextual predictors of cyberbullying victimization, as seen in nine meta-analyses (Fig. 5a). They included higher frequency and type of internet use, internet addiction, risky online behaviour and being perpetrators and victims of cyberbullying previously, all of which were associated

with increased cyberbullying victimization in 19 out of 23 effect sizes (median *r* = 0.19, range −0.11 to 0.87).

Finally, taking part in anti-cyberbullying interventions, including both school-based programmes and parental education programmes, was indicated by 11 meta-analyses as consistently associated with lower

**Table 2 | Methodological quality assessment of the included meta-analyses by record according to the JBI Critical Appraisal of Systematic Reviews and Research Synthesis**

| Author(s) and citation | Clear review question? | Appropriate inclusion criteria? | Appropriate search strategy? | Adequate use of sources? | Appropriate appraisal criteria? | Appraisal by two or more reviewers? | Methods to minimize error in data? | Appropriate method to combine studies? | Assessed publication bias? | Appropriate recommendations? | Appropriate new research directives? | Overall appraisal score |
|---|---|---|---|---|---|---|---|---|---|---|---|---|
| **Meta-analyses reporting predictors of cyberbullying victimization** | | | | | | | | | | | | |
| Abregú-Crespo et al.[98a] | ⊗ | ⊗ | ⊗ | ⊗ | ⊗ | ND | ⊗ | ⊗ | ⊗ | ⊗ | ⊗ | 9 |
| Barlett et al.[99a] | ⊗ | ⊗ | ⊗ | ⊗ | ⊗ | ⊗ | ⊗ | ⊗ | ⊗ | ⊗ | ⊗ | 6 |
| Caridade and Braga[64] | ⊗ | ⊗ | ⊗ | ⊗ | ⊗ | ⊗ | ⊗ | ⊗ | ⊗ | ⊗ | ⊗ | 11 |
| Chen et al.[73] | ⊗ | ⊗ | ⊗ | ⊗ | ⊗ | ⊗ | ⊗ | ⊗ | ⊗ | ⊗ | ⊗ | 8 |
| Chen et al.[89] | ⊗ | ⊗ | ⊗ | ⊗ | ⊗ | ⊗ | ⊗ | ⊗ | ⊗ | ⊗ | ⊗ | 11 |
| Christina et al.[101a] | ⊗ | ND | ⊗ | ⊗ | ⊗ | ⊗ | ⊗ | ⊗ | ⊗ | ⊗ | ⊗ | 8 |
| Doty et al.[102] | ⊗ | ⊗ | ⊗ | ⊗ | ⊗ | ⊗ | ⊗ | ⊗ | ⊗ | ⊗ | ⊗ | 9 |
| Eberle[103] | ⊗ | ⊗ | ⊗ | ⊗ | ⊗ | ⊗ | ⊗ | ⊗ | ⊗ | ⊗ | ⊗ | 6 |
| Erbiçer et al.[104a] | ⊗ | ⊗ | ⊗ | ⊗ | ⊗ | ⊗ | ⊗ | ⊗ | ⊗ | ⊗ | ⊗ | 11 |
| García-Hermoso et al.[106] | ⊗ | ⊗ | ⊗ | ⊗ | ⊗ | ND | ⊗ | ⊗ | ⊗ | ⊗ | ⊗ | 7 |
| Gilbar et al.[107] | ⊗ | ⊗ | ⊗ | ⊗ | ⊗ | ⊗ | ⊗ | ⊗ | ⊗ | ⊗ | ⊗ | 11 |
| Gaffney et al.[105] | ⊗ | ⊗ | ⊗ | ND | ⊗ | ⊗ | ⊗ | ⊗ | ⊗ | ⊗ | ⊗ | 7 |
| Gini et al.[108a] | ⊗ | ⊗ | ⊗ | ⊗ | ⊗ | ⊗ | ⊗ | ⊗ | ⊗ | ⊗ | ⊗ | 9 |
| Guo[37] | ⊗ | ⊗ | ⊗ | ⊗ | ⊗ | ⊗ | ⊗ | ⊗ | ⊗ | ⊗ | ⊗ | 9 |
| Hu et al.[110] | ⊗ | ⊗ | ⊗ | ⊗ | ⊗ | ⊗ | ⊗ | ⊗ | ⊗ | ⊗ | ⊗ | 9 |
| Huang et al.[111] | ⊗ | ⊗ | ⊗ | ⊗ | ⊗ | ⊗ | ⊗ | ⊗ | ⊗ | ⊗ | ⊗ | 10 |
| Kamaruddin et al.[16] | ⊗ | ⊗ | ⊗ | ⊗ | ⊗ | ⊗ | ⊗ | ⊗ | ⊗ | ⊗ | ⊗ | 11 |
| Kowalski et al.[28a] | ⊗ | ⊗ | ⊗ | ⊗ | ⊗ | ⊗ | ⊗ | ⊗ | ⊗ | ⊗ | ⊗ | 9 |
| Lan et al.[90] | ⊗ | ⊗ | ⊗ | ⊗ | ⊗ | ⊗ | ⊗ | ⊗ | ⊗ | ⊗ | ⊗ | 9 |
| Li et al.[114] | ⊗ | ⊗ | ⊗ | ⊗ | ⊗ | ⊗ | ⊗ | ⊗ | ⊗ | ⊗ | ⊗ | 9 |
| López-Barranco et al.[115] | ⊗ | ⊗ | ⊗ | ⊗ | ⊗ | ⊗ | ⊗ | ⊗ | ⊗ | ⊗ | ⊗ | 9 |
| Lozano-Blasco et al.[116] | ⊗ | ⊗ | ⊗ | ⊗ | ⊗ | ⊗ | ⊗ | ⊗ | ⊗ | ⊗ | ⊗ | 7 |
| Lozano-Blasco et al.[117] | ⊗ | ⊗ | ⊗ | ⊗ | ⊗ | ND | ⊗ | ⊗ | ⊗ | ⊗ | ⊗ | 9 |
| Marciano et al.[82a] | ⊗ | ⊗ | ⊗ | ⊗ | ⊗ | ND | ⊗ | ⊗ | ⊗ | ⊗ | ⊗ | 10 |
| Modecki et al.[118] | ⊗ | ⊗ | ⊗ | ⊗ | ⊗ | ⊗ | ⊗ | ⊗ | ⊗ | ⊗ | ⊗ | 8 |
| Ng et al.[25] | ⊗ | ⊗ | ⊗ | ⊗ | ⊗ | ⊗ | ⊗ | ⊗ | ⊗ | ⊗ | ⊗ | 11 |
| Oblad[65] | ⊗ | ⊗ | ⊗ | — | ⊗ | ⊗ | ⊗ | ⊗ | ⊗ | ⊗ | ⊗ | 6 |
| Polanin et al.[26] | ⊗ | ⊗ | ⊗ | ⊗ | ⊗ | ND | ⊗ | ⊗ | ⊗ | ⊗ | ⊗ | 10 |
| Pratt et al.[121] | ⊗ | ⊗ | ⊗ | ⊗ | ⊗ | ⊗ | ND | ⊗ | ND | ⊗ | ⊗ | 6 |
| Resett and Mesurado[122] | ⊗ | ⊗ | ⊗ | ⊗ | ⊗ | ⊗ | ⊗ | ⊗ | ⊗ | ⊗ | ⊗ | 8 |
| Sarier[123] | ⊗ | ⊗ | ND | ⊗ | ⊗ | ⊗ | ⊗ | ⊗ | ⊗ | ⊗ | ⊗ | 6 |
| Sun and Fan[66] | ⊗ | ⊗ | ⊗ | ⊗ | ⊗ | ⊗ | ⊗ | ⊗ | ⊗ | ⊗ | ⊗ | 7 |
| Van Cleemput et al.[125] | ⊗ | ⊗ | ⊗ | ⊗ | ⊗ | ⊗ | ⊗ | ⊗ | ⊗ | ⊗ | ⊗ | 8 |

**Table 2 (continued) | Methodological quality assessment of the included meta-analyses by record according to the JBI Critical Appraisal of Systematic Reviews and Research Synthesis**

| Author(s) and citation | Clear review question? | Appropriate inclusion criteria? | Appropriate search strategy? | Adequate use of sources? | Appropriate appraisal criteria? | Appraisal by two or more reviewers? | Methods to minimize error in data? | Appropriate method to combine studies? | Assessed publication bias? | Appropriate recommendations? | Appropriate new research directives? | Overall appraisal score |
|---|---|---|---|---|---|---|---|---|---|---|---|---|
| Walters[127a] | ⊗ | ⊗ | ⊗ | ⊗ | ⊗ | ⊗ | ⊗ | ⊗ | ⊗ | ⊗ | ⊗ | 7 |
| Wang and Jiang[91] | ⊗ | ⊗ | ⊗ | ⊗ | ⊗ | ⊗ | ⊗ | ⊗ | ⊗ | ⊗ | ⊗ | 9 |
| Wissink et al.[67] | ⊗ | ⊗ | ⊗ | ⊗ | ⊗ | ⊗ | ⊗ | ⊗ | ⊗ | ⊗ | ⊗ | 9 |
| Wirth[128] | ⊗ | ⊗ | ⊗ | ⊗ | ⊗ | ⊗ | ⊗ | ⊗ | ⊗ | ⊗ | ⊗ | 9 |
| Zhang and Chen[131] | ⊗ | ⊗ | ⊗ | ⊗ | ⊗ | ⊗ | ⊗ | ⊗ | ⊗ | ⊗ | ⊗ | 10 |
| Zych et al.[132] | ⊗ | ⊗ | ⊗ | ⊗ | ⊗ | ⊗ | ⊗ | ⊗ | ⊗ | ⊗ | ⊗ | 9 |
| **Meta-analyses reporting consequences of cyberbullying victimization** | | | | | | | | | | | | |
| Abregú-Crespo et al.[98a] | ⊗ | ⊗ | ⊗ | ⊗ | ⊗ | ND | ⊗ | ⊗ | ⊗ | ⊗ | ⊗ | 9 |
| Barlett et al.[99a] | ⊗ | ⊗ | ⊗ | ⊗ | ⊗ | ⊗ | ⊗ | ⊗ | ⊗ | ⊗ | ⊗ | 6 |
| Chen et al.[100] | ⊗ | ⊗ | ⊗ | ⊗ | ⊗ | ⊗ | ⊗ | ⊗ | ⊗ | ⊗ | ⊗ | 6 |
| Christina et al.[101a] | ⊗ | ND | ⊗ | ⊗ | ⊗ | ⊗ | ⊗ | ⊗ | ⊗ | ⊗ | ⊗ | 8 |
| Erbiçer et al.[104a] | ⊗ | ⊗ | ⊗ | ⊗ | ⊗ | ⊗ | ⊗ | ⊗ | ⊗ | ⊗ | ⊗ | 11 |
| Fisher et al.[36] | ⊗ | ⊗ | ⊗ | ⊗ | ⊗ | ⊗ | ⊗ | ⊗ | ⊗ | ⊗ | ⊗ | 9 |
| Gardella et al.[85] | ⊗ | ⊗ | ⊗ | ⊗ | ⊗ | ⊗ | ⊗ | ⊗ | ⊗ | ⊗ | ⊗ | 9 |
| Gini et al.[108a] | ⊗ | ⊗ | ⊗ | ⊗ | ⊗ | ⊗ | ⊗ | ⊗ | ⊗ | ⊗ | ⊗ | 9 |
| Heerde and Hemphill[109] | ⊗ | ⊗ | ⊗ | ⊗ | ⊗ | ⊗ | ⊗ | ⊗ | ⊗ | ⊗ | ⊗ | 8 |
| Hu et al.[93] | ⊗ | ⊗ | ⊗ | ⊗ | ⊗ | ND | ⊗ | ⊗ | ⊗ | ⊗ | ⊗ | 9 |
| John et al.[35] | ⊗ | ⊗ | ⊗ | ⊗ | ⊗ | ND | ND | ⊗ | ⊗ | ⊗ | ⊗ | 10 |
| Killer et al.[112] | ⊗ | ⊗ | ⊗ | ⊗ | ⊗ | ND | ⊗ | ⊗ | ⊗ | ⊗ | ⊗ | 9 |
| Kowalski et al.[28a] | ⊗ | ⊗ | ⊗ | ⊗ | ⊗ | ⊗ | ⊗ | ⊗ | ⊗ | ⊗ | ⊗ | 9 |
| Li et al.[113] | ⊗ | ⊗ | ⊗ | ⊗ | ⊗ | ⊗ | ⊗ | ⊗ | ⊗ | ⊗ | ⊗ | 10 |
| Lozano-Blasco et al.[86] | ⊗ | ⊗ | ⊗ | ⊗ | ⊗ | ND | ND | ⊗ | ⊗ | ⊗ | ⊗ | 8 |
| Marciano et al.[82a] | ⊗ | ⊗ | ⊗ | ⊗ | ⊗ | ND | ⊗ | ⊗ | ⊗ | ⊗ | ⊗ | 10 |
| Mills et al.[38] | ⊗ | ⊗ | ⊗ | ⊗ | ⊗ | ⊗ | ND | ⊗ | ⊗ | ⊗ | ⊗ | 6 |
| Molero et al.[119] | ⊗ | ⊗ | ⊗ | ⊗ | ⊗ | ⊗ | ND | ⊗ | ⊗ | ⊗ | ⊗ | 8 |
| Nesi et al.[120] | ⊗ | ⊗ | ⊗ | ⊗ | ⊗ | ⊗ | ⊗ | ⊗ | ⊗ | ⊗ | ⊗ | 8 |
| Tran et al.[124] | ⊗ | ⊗ | ⊗ | ⊗ | ⊗ | ⊗ | ⊗ | ⊗ | ⊗ | ⊗ | ⊗ | 10 |
| van Geel et al.[126] | ⊗ | ⊗ | ⊗ | ⊗ | ⊗ | ⊗ | ⊗ | ⊗ | ⊗ | ⊗ | ⊗ | 8 |
| Walters[129] | ⊗ | ⊗ | ⊗ | ⊗ | ⊗ | ⊗ | ⊗ | ⊗ | ⊗ | ⊗ | ⊗ | 7 |
| Wong[38] | ⊗ | ⊗ | ⊗ | ⊗ | ⊗ | ⊗ | ⊗ | ⊗ | ⊗ | ⊗ | ⊗ | 8 |
| Yuchang et al.[130] | ⊗ | ⊗ | ⊗ | ⊗ | ⊗ | ⊗ | ⊗ | ⊗ | ⊗ | ⊗ | ⊗ | 7 |
| Zych et al.[94] | ⊗ | ⊗ | ⊗ | ⊗ | ⊗ | ⊗ | ⊗ | ⊗ | ⊗ | ⊗ | ⊗ | 8 |

⊗, yes; ⊗, no; ND, not determined. [a]Records reporting both predictors and consequences of cyberbullying victimization.

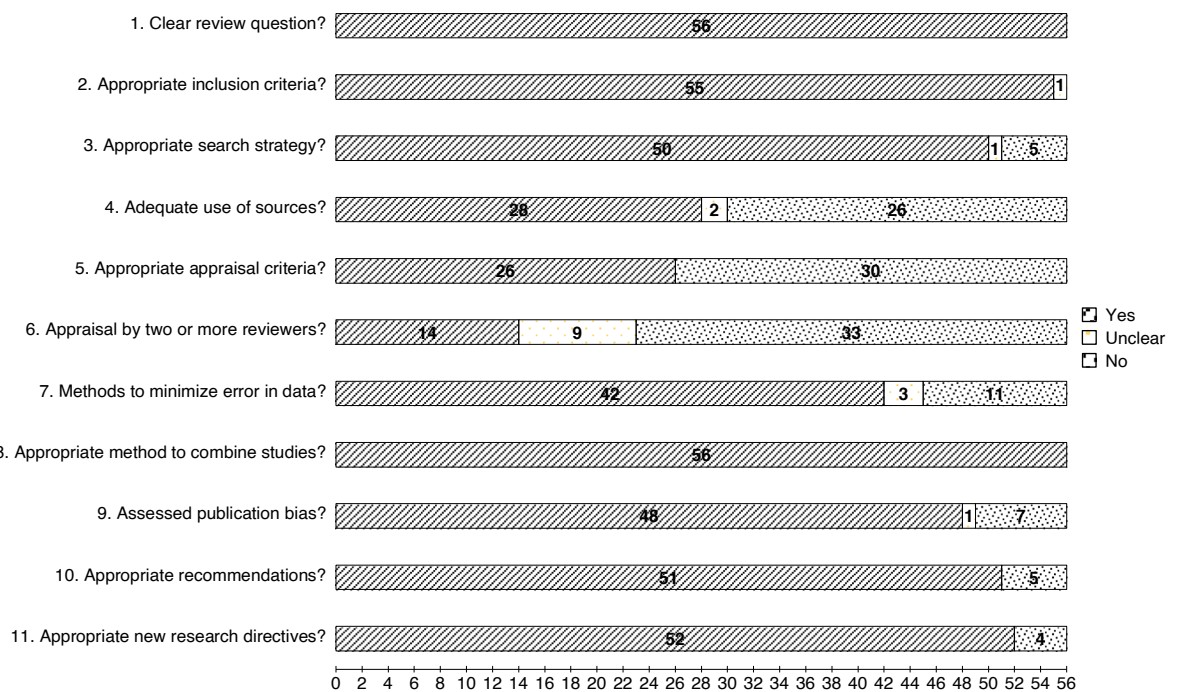

**Fig. 2 | Methodological quality assessment of the included meta-analyses.** An assessment using criteria according to the JBI Critical Appraisal of Systematic Reviews and Research Synthesis.

levels of cyberbullying victimization (median $r$ = −0.07, range −0.14 to −0.04) (Fig. 5b).

**Consequences of cyberbullying victimization.** In total, 25 out of the 56 included records included effect sizes on the relationship between cyberbullying victimization and its consequences.

*Psychological consequences.* A total of 21 meta-analyses provided effect sizes regarding associations between cyberbullying victimization and psychological consequences (Fig. 6). Overall, internalizing and emotional problems were common consequences of cyberbullying victimization. Individuals experiencing cyberbullying victimization displayed increased anxiety, depression, emotional problems, stress, loneliness and moral disengagement in 30 out of 31 effect sizes (median $r$ = 0.24, range −0.04 to 0.35). Furthermore, victims of cyberbullying were also more likely to show tendencies of self-harm and suicidal behaviour in all examined meta-analyses (median $r$ = 0.29, range 0.04 to 0.40). Conversely, cyberbullying victimization was negatively associated with positively valenced psychological variables in all meta-analyses (median $r$ = −0.150, range −0.310 to −0.003), with victims showing decreased levels of empathy, life satisfaction and self-esteem.

*Behavioural consequences.* Nine meta-analyses provided associations between cyberbullying victimization and behavioural predictors (Fig. 7). In 17 out of 18 effect sizes, higher cyberbullying victimization was associated with higher levels of externalizing behaviours and behavioural problems (median $r$ = 0.22, range −0.26 to 0.61), including aggressive behaviour and traditional bullying perpetration, cyberbullying perpetration, conduct problems, increased social problems, less prosocial behaviour, risky sexual behaviour and increased drug and alcohol use.

Four meta-analyses also indicated associations between higher levels of cyberbullying victimization and school-related outcomes (median $r$ = 0.14, range 0.06 to 0.36). Being subjected to increased levels of cybervictimization was associated with decreased levels

of academic achievement, lower school attendance and worse peer relationships, as well as being subjected to both traditional and cyberbullying in the long term.

## Discussion
The current systematic review examines meta-analyses on the predictors and consequences associated with cyberbullying victimization. A total of 56 meta-analyses, with a total of 296 effect sizes, were reviewed within the current work. The umbrella review approach made it possible to consider a broad scope of factors investigated by scholars and consider whether consensus in the field has been met on the factors that cause cyberbullying victimization and its consequences[96,97]. Our findings begin with a detailed analysis of the sociodemographic predictors, revealing nuanced differences in vulnerability among various groups. The subsequent sections delve into the psychological and contextual factors, each highlighted by distinct patterns and relationships that emerge from the meta-analytical data. The central findings derived from the analysis provide a holistic view of the potential predictors and consequences of cyberbullying victimization and serve as a basis for future research as well as interventions. We now discuss the ten central findings of the current review.

### Females are more likely to be subjected to cyberbullying victimization but are over-represented in cyberbullying research
Meta-analyses consistently show that females (versus males) are at a slightly higher risk of cyberbullying victimization[37,64–66,103,107,117]. Females engage more with cyberbullying as both perpetrators and victims owing to their higher involvement in indirect forms of aggression[133,134] and more frequent use of social networking sites[135,136]. Their tendency to share more personal information online increases their vulnerability[137,138]. Furthermore, females may interpret online comments as hurtful more quickly than males[139], contributing to higher reported levels of victimization and may, therefore, be overrepresented in cyberbullying research[140,141].

## Age appears to have a curvilinear relationship with cyberbullying victimization

Existing findings indicate a nonlinear relationship between age and cyberbullying victimization, with victimization rates increasing with age[28,37,67,117,123] but only until adulthood[64]. Research shows that as children and adolescents age, their increased use of computers, integration into social media and exposure to digital devices heighten their cyberbullying risk[140]. However, victimization rates flatten in older adults, potentially owing to cyberbullying's lower prevalence in this group and a general decrease in aggressive behaviours with age[13,48,62]. These trends suggest that the impact of age on cyberbullying must be cautiously interpreted, recognizing that, while youth are increasingly vulnerable, older adults may be less affected.

## Cyberbullying victims are likely to become cyberbullying perpetrators in future

Cyberbullying victimization was associated with future perpetration across three meta-analyses[67,82,99,127]. Unlike traditional bullying, which often involves physical disparities[142], cyberbullying occurs online, enabling victims to become bullies more easily due to the absence of physical disadvantage[143]. Online anonymity complicates identifying bullies and victims, allowing victims to adopt the role of bullies as a form of retaliation[51,144]. This ability to switch roles contributes to a vicious cycle of cyberbullying, escalating its negative impact within the online sphere.

## Cyberbullying victimization is associated with negative psychological outcomes and lower school performance, which may lead to maladaptive behaviours

Four meta-analyses consistently show that cyberbullying victims often display lower school performance[28,82,85,99], leading to considerable psychological distress and maladaptive coping behaviours[28,36,82,94,130]. Hurtful online comments can make victims feel isolated and emotionally distressed, resulting in feelings of hopelessness, lowered self-esteem and increased anxiety, which often culminate in depression[69,145]. These emotional burdens can reduce attendance and participation in school and social activities[85], further impacting academic performance. Consequently, victims may engage in deviant behaviours such as aggression, substance use and risky sexual behaviour[28,36,82,116] as coping mechanisms to offset psychological and academic issues[146]. This cycle of adverse effects is supported by the developmental cascades model[147], which links the snowball effect of stressors, such as cyberbullying, to escalating externalizing behaviour[148].

## Negative psychological consequences of cyberbullying victimization increase the possibility of future victimization

Fourteen meta-analyses reveal that cyberbullying victimization was associated with considerable psychological impacts, including anxiety, depression, low empathy, reduced life satisfaction, loneliness, low self-esteem and stress, serving both as outcomes and predictors[28,36,82,93,94,98,99,101,104,112,113,119,124,130]. Victims often experience isolation and negative emotions[69], leading to hopelessness and depression[145], which are, in turn, linked to self-harm and suicidal ideation[149]. Further, emotional vulnerabilities, such as poor anger management, antisocial tendencies or externalizing behaviours, can increase the likelihood of becoming a cyberbullying victim[150]. Research by Guo[37] supports that higher aggressive cognition predicts increased cyberbullying victimization. Victims, often distanced from social groups owing to such antisocial or aggressive tendencies, appear more susceptible to bullies and are prone to seek interactions through online media[69,70], increasing their risk of encountering perpetrators[151]. This dynamic underscores the cyclical nature of cyberbullying, where the psychological effects also become risk factors, perpetuating victim vulnerability.

## Parental support is a consistent protective factor against cyberbullying victimization, but the effect tends to be small

Nine meta-analyses indicate a small positive correlation between strong family relationships and a reduced risk of cyberbullying victimization[28,37,64,67,73,99,103,104,123]. Children and adolescents with involved parents, who monitor their internet use and are informed about their online experiences, are less likely to be victimized[28,152]. This parental mediation acts as a protective factor, aligning with the Routine Activity Theory, which emphasizes the role of capable guardians as a protective factor against experiencing deviant acts[74–76,153]. However, the effect remains limited as children's cyber activities often extend beyond parental supervision, especially in settings such as schools[154].

## Individuals in non-supportive romantic relationships are at higher risk of cyberbullying victimization

Results from three meta-analyses indicate that negative relationships with intimate partners significantly increase the risk of cyberbullying victimization to a small extent[64,67,107]. This heightened risk often stems from the fact that the perpetrator of cyberbullying is frequently the same individual involved in negative in-person interactions, especially in cases of cyber-dating harassment[155,156]. Interestingly, Wissink et al.[67] observed an association, albeit non-significant, of having younger partners being linked to higher cyberbullying victimization. This is possibly because younger couples, being more active online, may encounter cyberbullying more frequently[157]. This observation is noteworthy, as it highlights potential age-related dynamics in cyberbullying within intimate relationships.

## Lack of teacher–student interactions in school are associated with higher levels of cyberbullying victimization

Ten meta-analyses reveal that unfavourable school climates lacking proper teacher–student interactions have been consistently associated with small increases in cyberbullying victimization[28,37,67,73,82,99,104,108,118,132]. These environments, which also foster traditional bullying due to minimal supervision, allow unrestricted access to digital media and school devices, exacerbating cyberbullying risks[158–160]. According to the Routine Activity Theory, such settings enable cyberbullies to operate unimpeded and leave victims vulnerable without teacher support[74–76]. Additionally, traditional bullying victimization and perpetration are both significantly associated with increased cyberbullying victimization[28,73], and as negative school climates facilitate traditional bullying, it can indirectly have a further heightening effect on the risk of cyberbullying victimization.

## Active internet users are more likely to become cyberbullying victims, especially when they engage in risky online behaviour

A small but significant association was observed between increased internet and digital media use and higher cyberbullying victimization rates across seven meta-analyses[37,67,73,82,99,104,127]. Active internet users are more more likely to encounter cyberbullying perpetrators[161,162], particularly when engaging in risky behaviours such as revealing private details online or visiting unverified websites. For example, sharing personal photos or details online increases vulnerability to attacks[163,164], and visiting new websites without verifying their safety can expose personal information, attracting cyberbullying[165].

## Anti-cyberbullying intervention programmes are effective in reducing cyberbullying

Participation in cyberbullying intervention programmes has been consistently shown across 11 meta-analyses to have a small but significant effect in reducing victimization[25,26,89–91,102,105,122,125,128]. These findings were consistent, regardless of whether the programme was school based, targeting children and adolescents[16,89,90,105] or home based, aimed at increasing parental awareness[91]. These interventions typically focus on

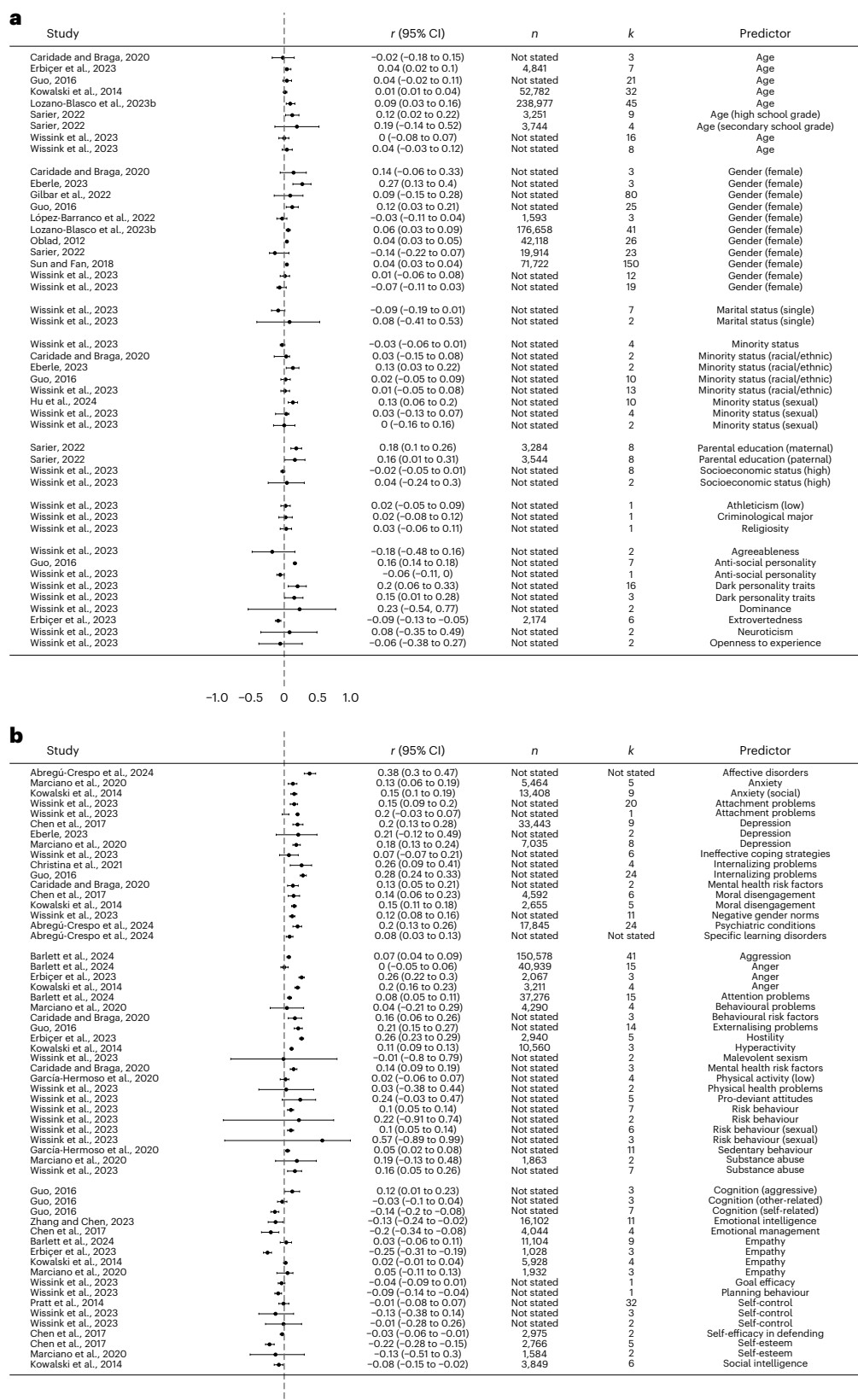

**Fig. 3 | Forest plot for the association between various predictors and cyberbullying victimization based on each individual review. a**, The association between sociodemographic and personality predictors and cyberbullying victimization based on each individual review. **b**, The association between psychological predictors and cyberbullying victimization based on each individual review. For **a** and **b**, *r* and the 95% confidence interval (CI) refer to the correlation between cyberbullying victimization and the predictor of interest. *n* refers to the sample size corresponding to each row ('not stated' is used in cases where meta-analyses did not provide relevant information), and *k* refers to the number of effect sizes used to calculate the correlation in each row.

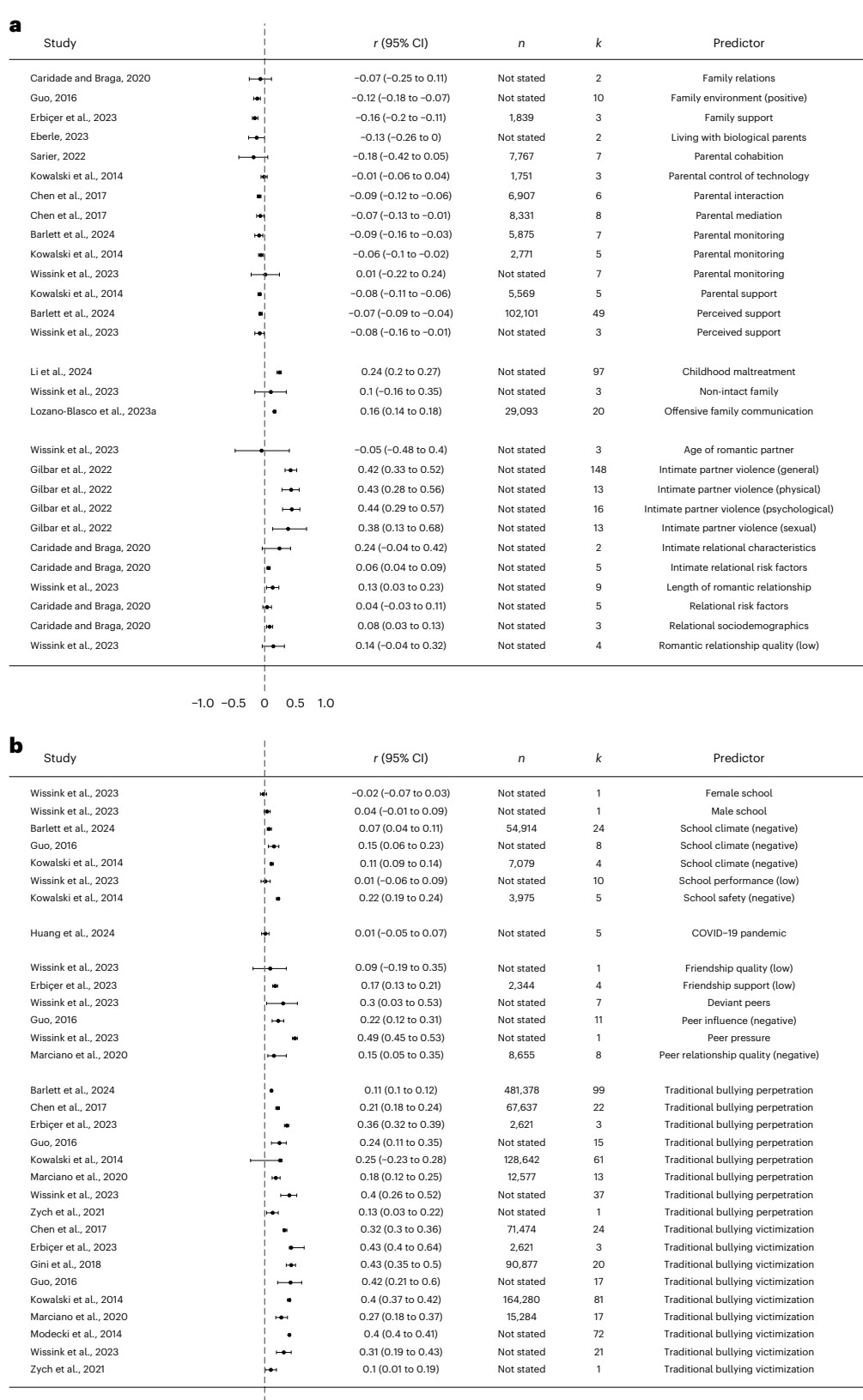

**Fig. 4 | Forest plot for the association between various predictors and cyberbullying victimization based on each individual review. a**, The association between parental and family relations and cyberbullying victimization based on each individual review. **b**, The association between school-related, peer-related and environmental factors and cyberbullying victimization based on each individual review.

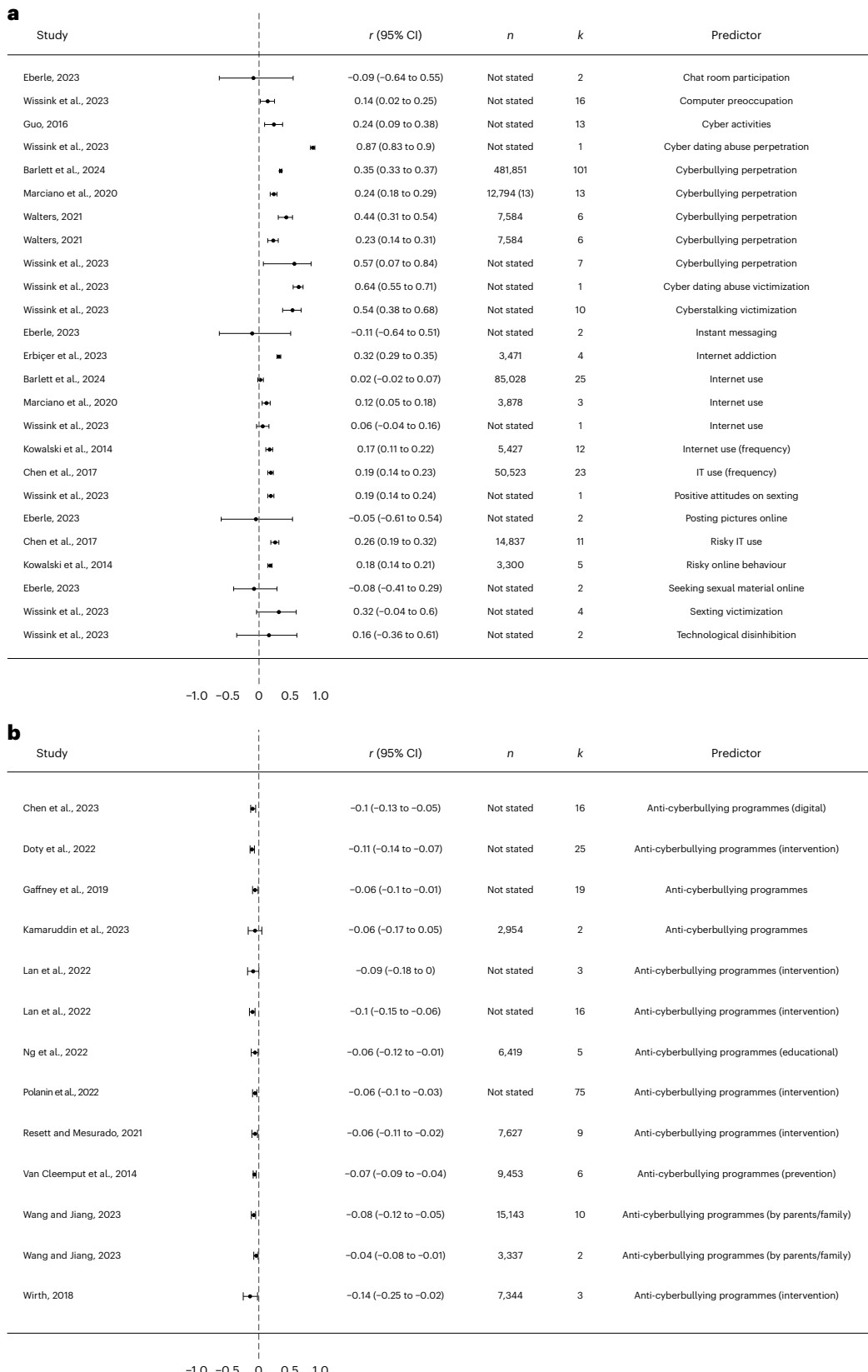

**Fig. 5 | Forest plot for the association between various predictors and cyberbullying victimization based on each individual review. a**, The association between factors related to internet use and cyberbullying victimization based on each individual review. **b**, The association between participating in anti-cyberbullying programmes and cyberbullying victimization based on each individual review. IT, information technology.

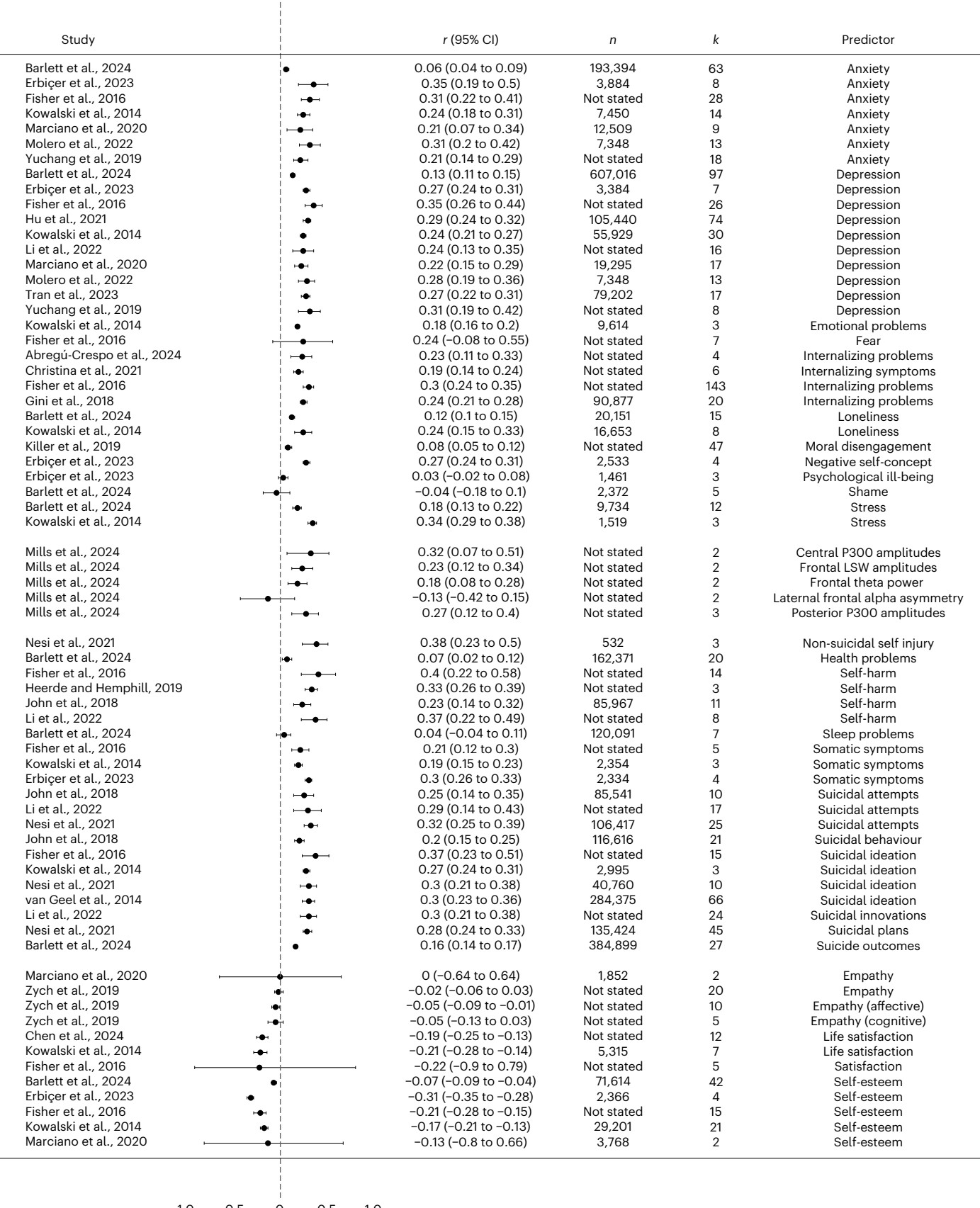

**Fig. 6 | Forest plot for the association cyberbullying victimization and various consequences based on each individual review.** The association between cyberbullying victimization and psychological predictors based on each individual review.

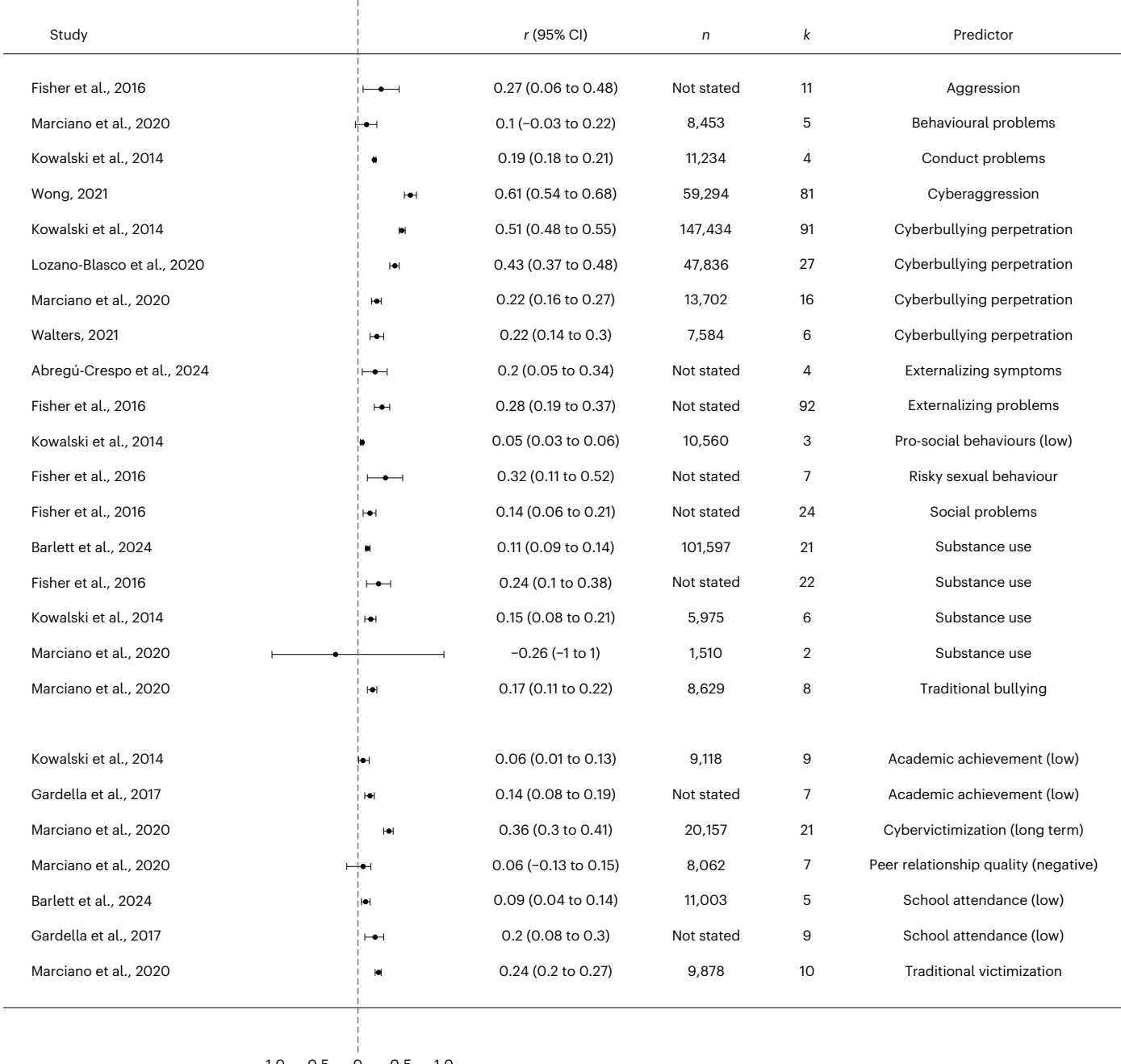

| Study | | r (95% CI) | n | k | Predictor |
|---|---|---|---|---|---|
| Fisher et al., 2016 | | 0.27 (0.06 to 0.48) | Not stated | 11 | Aggression |
| Marciano et al., 2020 | | 0.1 (−0.03 to 0.22) | 8,453 | 5 | Behavioural problems |
| Kowalski et al., 2014 | | 0.19 (0.18 to 0.21) | 11,234 | 4 | Conduct problems |
| Wong, 2021 | | 0.61 (0.54 to 0.68) | 59,294 | 81 | Cyberaggression |
| Kowalski et al., 2014 | | 0.51 (0.48 to 0.55) | 147,434 | 91 | Cyberbullying perpetration |
| Lozano-Blasco et al., 2020 | | 0.43 (0.37 to 0.48) | 47,836 | 27 | Cyberbullying perpetration |
| Marciano et al., 2020 | | 0.22 (0.16 to 0.27) | 13,702 | 16 | Cyberbullying perpetration |
| Walters, 2021 | | 0.22 (0.14 to 0.3) | 7,584 | 6 | Cyberbullying perpetration |
| Abregú-Crespo et al., 2024 | | 0.2 (0.05 to 0.34) | Not stated | 4 | Externalizing symptoms |
| Fisher et al., 2016 | | 0.28 (0.19 to 0.37) | Not stated | 92 | Externalizing problems |
| Kowalski et al., 2014 | | 0.05 (0.03 to 0.06) | 10,560 | 3 | Pro-social behaviours (low) |
| Fisher et al., 2016 | | 0.32 (0.11 to 0.52) | Not stated | 7 | Risky sexual behaviour |
| Fisher et al., 2016 | | 0.14 (0.06 to 0.21) | Not stated | 24 | Social problems |
| Barlett et al., 2024 | | 0.11 (0.09 to 0.14) | 101,597 | 21 | Substance use |
| Fisher et al., 2016 | | 0.24 (0.1 to 0.38) | Not stated | 22 | Substance use |
| Kowalski et al., 2014 | | 0.15 (0.08 to 0.21) | 5,975 | 6 | Substance use |
| Marciano et al., 2020 | | −0.26 (−1 to 1) | 1,510 | 2 | Substance use |
| Marciano et al., 2020 | | 0.17 (0.11 to 0.22) | 8,629 | 8 | Traditional bullying |
| Kowalski et al., 2014 | | 0.06 (0.01 to 0.13) | 9,118 | 9 | Academic achievement (low) |
| Gardella et al., 2017 | | 0.14 (0.08 to 0.19) | Not stated | 7 | Academic achievement (low) |
| Marciano et al., 2020 | | 0.36 (0.3 to 0.41) | 20,157 | 21 | Cybervictimization (long term) |
| Marciano et al., 2020 | | 0.06 (−0.13 to 0.15) | 8,062 | 7 | Peer relationship quality (negative) |
| Barlett et al., 2024 | | 0.09 (0.04 to 0.14) | 11,003 | 5 | School attendance (low) |
| Gardella et al., 2017 | | 0.2 (0.08 to 0.3) | Not stated | 9 | School attendance (low) |
| Marciano et al., 2020 | | 0.24 (0.2 to 0.27) | 9,878 | 10 | Traditional victimization |

−1.0  −0.5   0   0.5   1.0

**Fig. 7 | Forest plot for the association between cyberbullying victimization and various consequences based on each individual review.** The association between cyberbullying victimization and behavioural predictors based on each individual review.

educating about cyberbullying, identifying and mitigating risky behaviours and, sometimes, include a component for parental training[26,166–168]. By addressing such risk and protective factors, these programmes effectively reduce the likelihood of cyberbullying victimization.

## Limitations

The current review has several limitations. First, although this review provides an overview of a wide range of findings, it is unable to study the finer details included in either the meta-analyses or the original primary studies. While the umbrella review approach allows studying aggregated findings to reveal more precise and generalizable results that could not be arrived at via analysing single empirical studies[169], it does not facilitate the studying of more detailed aspects of various studies (for example, different moderators, types of measure utilized

and response time frames). As such, it is important to consider these nuances by directing attention to the individual meta-analyses contained in the current review, as well as the various studies cited within them. Second, the current review only considered associations between cyberbullying victimization and various predictors and consequences in the form of correlations. As a majority of the included meta-analyses did not report directional or otherwise lagged findings, it was not possible to consider the directional relationship between factors within the scope of the current review.

## Research gaps and potential future research

The current review highlights several research gaps in cyberbullying victimization literature. First, most meta-analyses focus on child or adolescent populations. Given that cyberbullying victimization differs

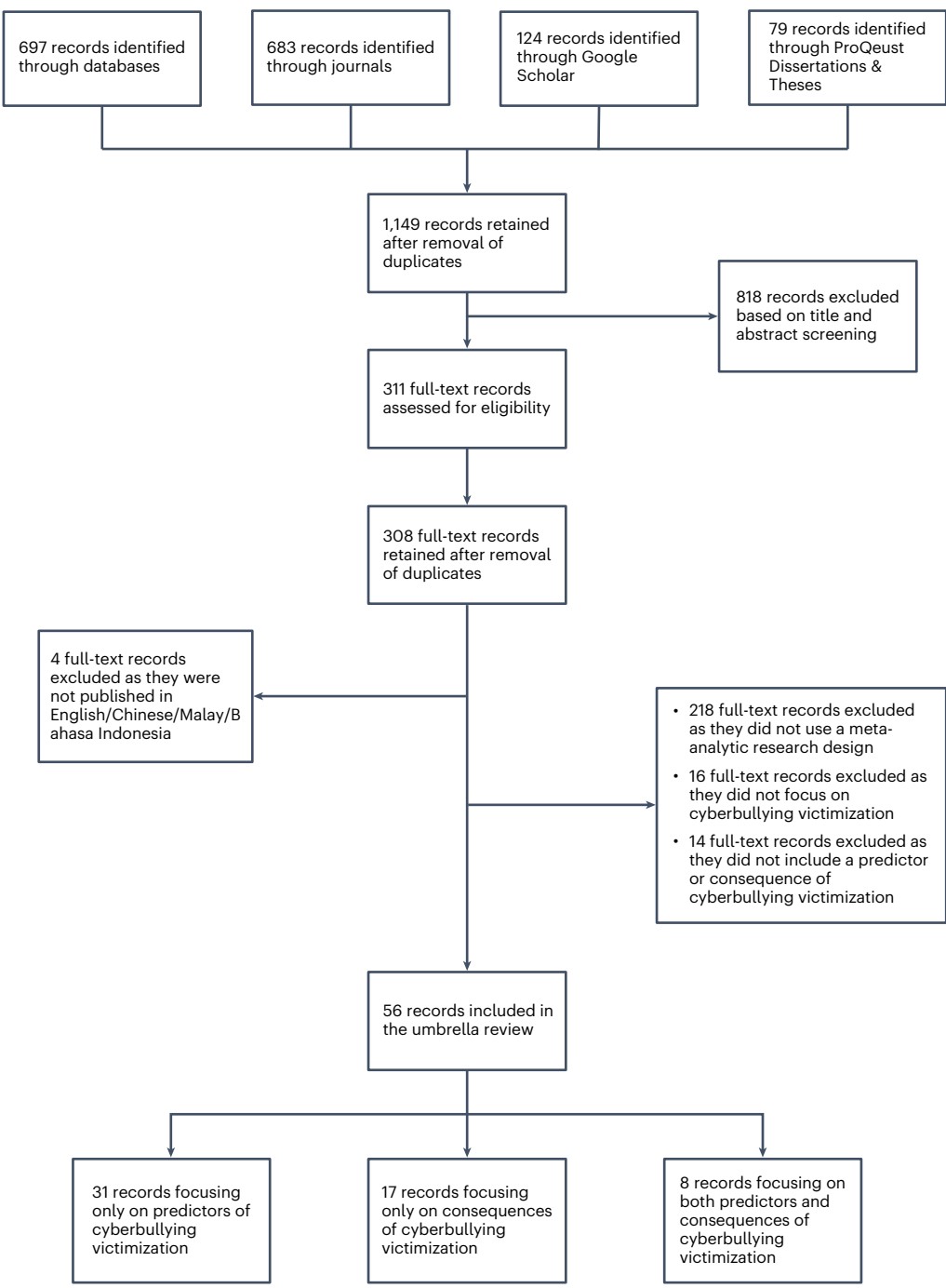

**Fig. 8 | PRISMA flowchart.** The PRISMA flowchart illustrate the record selection process, including the number of studies indentified or retained at eachstage of screening.

based on age group[13,62], and older adults may have different reactions to cyberbullying victimization than younger populations, research involving broader demographics is needed to better understand its impact on different age groups.

Second, the meta-analyses included in this review mainly defined cyberbullying by focusing on the different media platforms through which cyberbullying occurs, rather than on the different acts of cyberbullying[28,82,104,119]. However, research suggests that individuals do not distinguish cyberbullying based on the medium used but rather on the nature of the bullying acts themselves[170]. Therefore, future work should aim to refine definitions that emphasize behaviours involved in cyberbullying and incorporate behavioural measurements within cyberbullying scales to more accurately capture the phenomenon.

Third, there was a lack of meta-analyses on cyberbullying related to intimate-partner relations. A majority of the included meta-analyses focused on peer-cyberbullying victimization within young samples and were, therefore, unable to examine intimate partner relations. However, as the current review reveals that intimate partner relations can have considerable impacts on an individual's tendency to become a cyberbullying victim, it is important for future research to consider cyber-related intimate partner violence as a branch of cyberbullying and explore further into its risk factors.

Lastly, while interventions were identified as a protective against cyberbullying victimization, the meta-analyses lacked long-term follow-up data. As analysing long-term impacts of anti-cyberbullying interventions is important to better understand the impact of such

programmes, it is critical for future research to consider follow-up analysis to gain a better idea of the impact of interventions.

## Conclusion

The growth of internet and social media as a communication platform has increased the incidence of cyberbullying victimization. While there has been much research exploring the various predictors and consequences of cyberbullying victimization, most has focused on a narrow range of variables or contexts. As such, the current review aims to conduct systematic and comprehensive review of meta-analyses to reconcile literature on the various predictors and consequences of cyberbullying victimization. Findings suggest that females, school-aged populations, individuals who experienced traditional bullying and individuals who use the internet more are more likely to be cyberbullied. Unregulated school environments and unsupportive parental relationships are also associated with higher levels of cyberbullying victimization. Cyberbullying victimization is consistently associated with negative psychological outcomes such as anxiety, depression and loneliness, as well as lower school performance and maladaptive coping behaviours. The systematic identification of these robust predictors and consequences provides crucial insights that can aid stakeholders—educators, policymakers and community leaders—in developing targeted interventions that are grounded in empirical evidence. For instance, knowing specific risk factors allows for the design of prevention programmes tailored to protect vulnerable groups, while understanding the psychological impacts helps in structuring appropriate therapeutic responses. Developing such interventions is especially important, as the current review found that cyberbullying interventions show promising results. This underscores the urgent need to devote adequate resources towards developing and implementing informed evidence-based strategies to effectively combat cyberbullying victimization.

## Methods

### Transparency and openness

The current review was conducted in accordance with the PRISMA guidelines[171]. The design and synthesis plan of the current review was not pre-registered. Ethical approval was not required, as the study design (umbrella review) is exempted by the local institutional review board. Mendeley Desktop version 1.19.8 (ref. [172]) was used to remove duplicates from the records obtained after the retrieval process.

In cases where effect sizes were not reported in the form of correlations, conversions were conducted using R version 3.6.3 (ref. [173]). 'effectsize' version 0.0.6.1 (ref. [174]) was used to convert Cohen's $d$ and odds ratios to Pearson's $r$. 'psych' version 2.2.5 (ref. [175]) was used to convert Fisher's $z$ to Pearson's $r$. In cases where Hedge's $g$ was provided, it was converted to Cohen's $d$ using the following formula $d = \frac{g}{\left(1 - \frac{3}{4(n_1 + n_2) - 9}\right)}$

(http://dlinares.org/cohend.html), where $n_1$ and $n_2$ refer to the sample sizes of the two groups used to calculate the effect size. The result was then converted into Pearson's $r$ using the 'effectsize' package. For meta-analyses that presented Hedge's $g$ and did not disclose $n_1$ and $n_2$, we assume that the meta-analyses included a large total sample size and treated Hedge's $g$ and Cohen's $d$ as equivalent[176]. Forest plots for the visualization of results were created using Microsoft Excel version 16.78 (ref. [177]). The R analytic code used to convert effect sizes as well as all screening records and data extraction records of the current review are publicly available on Researchbox no. 1364 (https://researchbox.org/1364).

### Study design

The current work was conducted as an umbrella review, a distinct form of systematic review designed to compile data from multiple meta-analyses addressing the same research questions[96,97]. This approach allows for a comprehensive synthesis of evidence across studies, enhancing our understanding by comparing and contrasting results from different meta-analyses. By aggregating findings across these studies, an umbrella review helps identify patterns, strengths and gaps in the literature, providing a robust analysis of extensive datasets. This methodology is particularly suitable for fields with a vast array of studies and varying outcomes, such as cyberbullying victimization, where it can effectively distil broad insights from diverse research findings.

### Search strategy

A search strategy was developed by the first author and agreed upon by the first, second and last authors to capture relevant records from each of the sources. Systematic searches were conducted by the first author on various sources for meta-analyses available up to 7 April 2024. Main sources comprised five databases (EBSCOhost ERIC, EBSCOhost PsycInfo, PubMed, Scopus and Web of Science) and 13 journals related to the field of cyberbullying (*Adolescent Research Review*; *Aggression and Violent Behavior*; *Aggressive Behavior*; *Children and Youth Services Review*; *Computers in Human Behavior*; *Cyberpsychology, Behavior, and Social Networking*; *Deviant Behavior*; *Journal of Adolescence*; *Journal of Pediatric Nursing*; *Journal of School Violence*; *New Media and Society*; *School Psychology Review*; *Trauma, Violence, and Abuse*). The journals were selected based on search strategies of previous meta-analyses on the topic[28], as well as by selecting journals that had recently published meta-analyses on the field of cyberbullying. To augment the search, two other sources (ProQuest Dissertations and Theses, Google Scholar) were also searched to retrieve additional published literature, as well as relevant unpublished literature.

The following keywords were used to conduct the systematic literature search within the five databases: ('meta-analy*' OR 'meta analy*' OR 'quantitative synthesis' OR 'review*') AND (cyber* OR internet OR net OR online OR chat OR electronic OR mobile OR 'social network' OR media OR Facebook OR Twitter OR Blog* OR Youtube OR Tumblr OR Discord OR Reddit OR Instagram OR Tiktok OR Snapchat OR Pinterest OR LinkedIn) AND (harass* OR bully* OR bulli* OR victim* OR aggres* OR abus* OR maltreat* OR incivil* OR toxic* OR violen* OR delinquen* OR devian* OR ragging OR hazing OR mobbing OR intimidat*). A simplified search string containing the following keywords was used to search the relevant journals and other sources: (meta-analysis OR 'meta analysis' OR review) AND (cyber OR internet OR online OR 'social media') AND (bully OR victim).

### Selection criteria

Following the literature search, the retrieved records were screened for potential inclusion independently by the first and third author or by the first author and a trained research assistant (see Fig. 8 for the PRISMA flowchart[178]). Any disagreements in the screening process were resolved through discussion between the two authors, and upon consensus, irrelevant and duplicate records were removed.

First, titles and abstracts were evaluated based on a preliminary set of criteria, which looked at whether each record (1) was published in English, Chinese, Malay or Bahasa Indonesia, (2) was a meta-analysis, (3) mentioned cyberbullying victimization and (4) mentioned at least one predictor or consequence in relation to cyberbullying victimization (94.5% overall inter-rater agreement between the first and third author and 92.76% overall inter-rater agreement between the firth author and research assistant).

Subsequently, the remaining records were assessed for inclusion based on their full-texts by the same authors and research assistant as per the following criteria (94.65% overall inter-rater agreement between the first and third author, 95.23% overall inter-rater agreement between the first author and research assistant):

1. Records were included if they were published in English, Chinese, Malay or Bahasa Indonesia.

**Table 3 | Categorization of predictors and consequences of cyberbullying victimization analysed in the review**

| Category | Variables |
| --- | --- |
| **Predictors** | |
| Sociodemographic and personality predictors | Age, gender, racial/ethnic minority, sexual minority, marital status, paternal education, maternal education, socioeconomic status, athleticism, religiosity, agreeableness, antisocial personality, dark personality traits (that is, Machiavellianism, psychopathy and narcissism), dominance, extraversion, neuroticism and openness to experience |
| Psychological predictors | Affective disorders, aggression, anger, anxiety, attachment problems, behavioural problems, cognition, depression, emotional intelligence, emotional management, empathy, externalizing problems, goal efficacy, hostility, hyperactivity, ineffective coping, internalizing problems, malevolent sexism, mental health, moral disengagement, negative gender norms (for example, norms/attitudes of violence towards the opposite gender), planning behaviour, pro-deviant attitudes, psychiatric conditions, risky behaviour, sedentary behaviour, self-control, self-efficacy in defending (that is, ability to effectively defend oneself), self-esteem, social intelligence, specific learning disorders and substance abuse |
| Contextual predictors | |
| Parental and family relations | Childhood maltreatment, family environment, intimate and family relations, intimate partner age, intimate relationship characteristics, intimate partner violence, length of romantic relationship, living with parents, non-intact family (that is, household structures other than two-parent households), offensive family communication, parental cohabitation, parental control of technology, parental interaction, parental mediation, parental support, perceived support and relationship quality |
| School, peer relations and other environmental contexts | Attending female-only or male-only schools, coronavirus disease 2019 pandemic, friendship quality and support, peer influence, peer pressure, peer relationship quality, school climate, school performance, school safety, traditional bullying perpetration and traditional victimization |
| Factors related to internet use | Chat room participation, computer preoccupation, cyber activities (that is, communication or personal activities using any form of technological device), cyberbullying perpetration (including cyber dating abuse), cyberbullying victimization (including cyber-dating abuse and cyberstalking), frequency of internet use and internet addiction, instant messaging, risky online behaviour (including sexting, posting online pictures and seeking sexual material) and technological disinhibition |
| Cyberbullying interventions | Participating in anti-cyberbullying interventions (for both potential victims and parents) |
| **Consequences** | |
| Psychological consequences | Anxiety, depression, emotional problems, empathy (including affective and cognitive empathy), fear, internalizing problems, life satisfaction, loneliness, moral disengagement, negative self-concept, neurological outcomes related to anger, distress, and emotional regulation, non-suicidal self-injury, psychological ill-being, self-esteem, self-harm, shame, sleep problems, somatic symptoms, stress, suicidal ideation and suicide attempts |
| Behavioural consequences | |
| Externalizing behaviour | Behavioural problems, conduct problems, cyberbullying perpetration, risky sexual behaviour, substance abuse and traditional bullying perpetration |
| School-related academic and social outcomes | Academic achievement, peer relationship quality, prosocial behaviour, school attendance, social problems with peers, traditional bullying and long-term cyberbullying victimization |

2. Records were included if they used a meta-analytic research design.
3. Meta-analyses were included if they focused on any type of cyberbullying victimization. Cyberbullying victimization was defined as being subjected to any aggressive or bullying behaviour (for example, threatening, harassing, abusing or disrespecting) aimed directly either towards themselves or a group involving them using electronic means. Cyberbullying victimization also includes being subjected to acts such as public posts or information aimed to defame or embarrass themselves or a group they are part of. Common types of cyberbullying victimization include (but are not limited to): cyber harassment or online harassment, cyber-aggression, peer-cyberbullying victimization and cyber partner abuse and online dating violence. The records were excluded if they focused only on cyberbullying perpetration (that is, carrying out acts of cyberbullying rather than being the victim of it).
4. Meta-analyses were included if they reported at least one predictor or consequence of cyberbullying victimization.
   a. Common examples for predictors of cyberbullying victimization include (but are not limited to) age, gender, culture, frequency of internet use/technology use, parental monitoring, school climate and exposure to traditional bullying. Interventions aimed at preventing cyberbullying were also considered predictors, as they are designed to reduce the

incidence or impact of cyberbullying and, therefore, may influence the likelihood or impact of an individual's cyberbullying victimization experience.
   b. Common examples for consequences of cyberbullying includ (but are not limited to) depression, anxiety, suicidal ideation, self-esteem, loneliness and academic achievement. Consequences of cyberbullying victimization across all domains were considered (that is, not only limited to mental health outcomes but also included other outcomes, such as educational achievement and drug and alcohol use).
5. Meta-analyses were included if they examined humans. No other restrictions were placed on any sample characteristics such as age, gender, health or country.
6. Meta-analyses were included regardless of the peer review status of meta-analyses (that is, meta-analyses were included whether or not they were peer reviewed). However, if two versions of the same meta-analyses were available (for example, as part of a thesis and as part of a journal article), only the peer-reviewed version was retained.
7. Meta-analyses were included if they reported sufficient statistical information (that is, effect sizes and variance or sample size). All types of effect size were accepted. If a meta-analysis did not report the necessary information, data were requested from the relevant authors via email, ResearchGate and/or other online communication channels.

**Article** https://doi.org/10.1038/s41562-024-02011-6

## Quality assessment

The quality of each included meta-analysis was assessed independently by the first and third author or by the first author and a trained research assistant using the JBI Critical Appraisal Instrument for Systematic Reviews and Research Syntheses[179]. The records were evaluated using an 11-item checklist, with each item rated according to four categories ('yes', 'no', 'unclear' and 'not applicable') based on how closely the records adhered to each criterion. The criteria guiding the methodological evaluation of each record were (1) clarity of review question, (2) use of appropriate inclusion criteria, (3) use of appropriate search strategies, (4) adequacy of sources and resources to search for studies, (5) use of appropriate criteria for appraisal of studies, (6) independent critical appraisal of studies, (7) employment of methods to minimize errors in data extraction, (8) use of appropriate data synthesis methods, (9) assessment of the likelihood of publication bias, (10) have recommendations for policy and/or practice backed by data reported and (11) use of appropriate specific directives for new research. Each record was then given a quality score based on how many 'yes' responses were accorded (that is, the number of 'yes' ratings out of 11). The inter-rater agreement was generally excellent on average across all criteria, with an overall agreement rate of 96% (range 92–100%) between the first and third author and an overall agreement rate of 94% (range 90–100%) between the first author and research assistant. Any remaining discrepancies or disagreements were resolved through discussion between the reviewers.

## Data extraction

The following information was independently extracted from the final list of included meta-analyses by either the first and third author or by the first author and a research assistant: author(s), year of publication, title of publication, countries and regions covered by the review, participant demographics, total number of studies, total unique sample size, cyberbullying definition and type of cyberbullying victimization measured, predictors and/or consequences of cyberbullying victimization and the relevant effect sizes denoting the association between cyberbullying victimization and the predictor and/or consequence of cyberbullying victimization explored within each meta-analysis. Regional classification of the different countries followed the listing by Wikimedia, Meta-Wiki[180] (2022). Effect sizes were extracted as given within each meta-analysis without any conversions. The inter-rater agreement for all variables was generally excellent for all variables (range 77.46–100% between the first and third author and range 81.54–100% between the first author and research assistant).

## Data analysis

The records included in the current review were expected to include a diverse range of predictors and consequences of cyberbullying victimization across multiple domains that were distinct from each other (for example, sociodemographic predictors, psychological predictors/consequences and behavioural consequences). Furthermore, the included meta-analyses were expected to display high levels of heterogeneity in terms of the study aims and types of cyberbullying measured. Due to these factors, it was not appropriate to synthesize results statistically. Thus, the included meta-analyses and their subsequent applicable findings were synthesized narratively by investigating the overall effect sizes denoting the association between cyberbullying victimization and the different predictors and consequences of cyberbullying victimization based on the primary findings of each meta-analysis (attempts to conduct subgroup analyses to explore heterogeneity among study results were not feasible due to an insufficient number of meta-analyses analysing identical subgroups for the same outcomes).

To better compare effect sizes, all extracted effect sizes were converted into Pearson's $r$ correlations by the first author. It was decided to use Pearson's $r$ as majority of the meta-analyses included in the current review reported correlational effect sizes (refer to 'Transparency and openness' for further details on the conversion process).

To synthesize associations between cyberbullying victimization and predictors of cyberbullying victimization in a theoretically appropriate manner, both predictors and consequences of cyberbullying victimization were further divided into different categories based on the domain of each variable (Table 3).

## Reporting summary

Further information on research design is available in the Nature Portfolio Reporting Summary linked to this article.

## Data availability

All screening records of the current review are publicly available via Researchbox (https://researchbox.org/1364).

## Code availability

The R analytic code used to convert effect sizes of the current review are publicly available via Researchbox (https://researchbox.org/1364).

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

## Acknowledgements

The authors received no specific funding for this work. We thank X. Ci Soh for her assistance in data extraction.

## Author contributions

Conceptualization was done by K.T.A.S.K. and A.H. The literature search was conducted by K.T.A.S.K. The screening of literature and data extraction was conducted by K.T.A.S.K. and C.H.Y.C. Analysis was conducted by K.T.A.S.K. Draft paper preparation was done by K.T.A.S.K. All authors contributed to reviewing and editing the paper. Visualizations were done by K.T.A.S.K. and N.M.M. Supervision was done by A.H., E.M.W.T. and N.M.M. All authors read and approved the final paper.

## Competing interests

The authors declare no competing interests.

## Additional information

**Correspondence and requests for materials** should be addressed to K. T. A. Sandeeshwara Kasturiratna or Andree Hartanto.

¹Singapore Management University, Singapore, Singapore. ²Department of Psychology, National University of Singapore, Singapore, Singapore. ³Social Service Research Centre, National University of Singapore, Singapore, Singapore. ✉e-mail: skasturirat.2023@phdps.smu.edu.sg; andreeh@smu.edu.sg

# Reporting Summary

## Statistics

For all statistical analyses, confirm that the following items are present in the figure legend, table legend, main text, or Methods section.

| n/a | Confirmed | |
|---|---|---|
| ☒ | ☐ | The exact sample size (*n*) for each experimental group/condition, given as a discrete number and unit of measurement |
| ☒ | ☐ | A statement on whether measurements were taken from distinct samples or whether the same sample was measured repeatedly |
| ☒ | ☐ | The statistical test(s) used AND whether they are one- or two-sided *Only common tests should be described solely by name; describe more complex techniques in the Methods section.* |
| ☒ | ☐ | A description of all covariates tested |
| ☒ | ☐ | A description of any assumptions or corrections, such as tests of normality and adjustment for multiple comparisons |
| ☐ | ☒ | A full description of the statistical parameters including central tendency (e.g. means) or other basic estimates (e.g. regression coefficient) AND variation (e.g. standard deviation) or associated estimates of uncertainty (e.g. confidence intervals) |
| ☒ | ☐ | For null hypothesis testing, the test statistic (e.g. *F*, *t*, *r*) with confidence intervals, effect sizes, degrees of freedom and *P* value noted *Give P values as exact values whenever suitable.* |
| ☒ | ☐ | For Bayesian analysis, information on the choice of priors and Markov chain Monte Carlo settings |
| ☒ | ☐ | For hierarchical and complex designs, identification of the appropriate level for tests and full reporting of outcomes |
| ☐ | ☒ | Estimates of effect sizes (e.g. Cohen's *d*, Pearson's *r*), indicating how they were calculated |

*Our web collection on statistics for biologists contains articles on many of the points above.*

## Software and code

Policy information about availability of computer code

| Data collection | No software was used for data collection. |
|---|---|
| Data analysis | In the literature search for the systematic review, Mendeley Desktop version 1.19.8 (Mendeley, n.d.) was used in order to remove duplicates from the records obtained after the retrieval process. In cases where effect sizes were not reported in the form of correlations, conversions were conducted using R version 3.6.3 (R Core Team, 2020). effectsize version 0.0.6.1 (Ben-Shachar et al., 2020) was used to convert Cohen's d and odds ratios to Pearson's r. psych version 2.2.5 (Revelle, 2021) was used to convert Fisher's z to Pearson's r. In cases where Hedge's g was provided, it was converted to Cohen's d using the following formula, d= g/((1- 3/(4(n_1+ n_2 )-9))) (http://dlinares.org/cohend.html), where n1 and n2 refer to the sample sizes of the two groups used to calculate the effect size. The result was then converted into Pearson's r using the effectsize package. |

For manuscripts utilizing custom algorithms or software that are central to the research but not yet described in published literature, software must be made available to editors and reviewers. We strongly encourage code deposition in a community repository (e.g. GitHub). See the Nature Portfolio guidelines for submitting code & software for further information.

## Data

Policy information about availability of data

All manuscripts must include a data availability statement. This statement should provide the following information, where applicable:

- Accession codes, unique identifiers, or web links for publicly available datasets
- A description of any restrictions on data availability
- For clinical datasets or third party data, please ensure that the statement adheres to our policy

> The R analytic code used to convert effect sizes as well as all screening records of the current review are publicly available on Researchbox #1364 (https://researchbox.org/1364&PEER_REVIEW_passcode=FWAIHM).

## Research involving human participants, their data, or biological material

Policy information about studies with human participants or human data. See also policy information about sex, gender (identity/presentation), and sexual orientation and race, ethnicity and racism.

| | |
|---|---|
| Reporting on sex and gender | The analysis was conducted from data extracted from the meta-analyses included in the umbrella review. All references to sex and gender were in-line with how each meta-analyses referred to the variables.<br><br>The effect of gender as a predictor of cyberbullying victimisation was narratively synthesised within the results. |
| Reporting on race, ethnicity, or other socially relevant groupings | The analysis was conducted from data extracted from the meta-analyses included in the umbrella review. All references to age, race and ethnicity were in-line with how each meta-analyses referred to the variables.<br><br>The effect of age and race as predictors of cyberbullying victimisation was narratively synthesised within the results. |
| Population characteristics | No data was collected for the study as the research utilised a systematic review design. Data were extracted from the 56 meta-analyses included in the systematic review. Sample sizes of the included meta-analyses ranged from 421 to 1,136,080 (Mdn=53,183), covering all regions including Africa, Arab States, Asia-Pacific, Europe, Middle East, North America and South America. 50 records (89%) were journal articles, while 1 record (2%) was a book chapter, 1 record (2%) was a conference piece, and 4 records (7%) were dissertations/theses. Out of the 56 included records, 47 records (84%) focused specifically on children and/or adolescents and young adults (including those focused on school settings), and 6 records (11%) focused on both children/adolescent and adult samples, while only 1 record (2%) focused solely on an adult sample (2 records did not provide information on their participant type). Statistics regarding sample age or female proportion were not provided by the majority of the meta-analyses. |
| Recruitment | No participants were recruited as the research utilised an umbrella review methodology. |
| Ethics oversight | Ethical approval was not required as the study design (umbrella review) was exempted from the local Institutional Review Board. |

Note that full information on the approval of the study protocol must also be provided in the manuscript.

# Field-specific reporting

Please select the one below that is the best fit for your research. If you are not sure, read the appropriate sections before making your selection.

☐ Life sciences ☒ Behavioural & social sciences ☐ Ecological, evolutionary & environmental sciences

For a reference copy of the document with all sections, see nature.com/documents/nr-reporting-summary-flat.pdf

# Behavioural & social sciences study design

All studies must disclose on these points even when the disclosure is negative.

| | |
|---|---|
| Study description | The study utilizes an umbrella review method, and conducts an umbrella review of meta-analyses on the predictors and consequences of cyberbullying victimisation. |
| Research sample | A total of 56 records were included in the final review. Records were made available from 2012 to 2024 inclusive, and included meta-analyses covering studies from 1993 to 2023 inclusive. Sample sizes ranged from 421 to 1,136,080 (Mdn=53,183), covering all regions including Africa, Arab States, Asia-Pacific, Europe, Middle East, North America and South America. 50 records (89%) were journal articles, while 1 record (2%) was a book chapter, 1 record (2%) was a conference piece, and 4 records (7%) were dissertations/theses. |
| Sampling strategy | A search strategy was developed by the first author and agreed upon by the first, second and last authors in order to capture relevant records from each of the sources. Systematic searches were conducted by the first author on various sources for meta-analyses available up to 7 April 2024. Main sources comprised five databases (EBSCOhost ERIC, EBSCOhost PsycInfo, PubMed, |

Scopus, Web of Science) and 13 journals related to the field of cyberbullying (Adolescent Research Review; Aggression and Violent Behavior; Aggressive Behavior; Children and Youth Services Review; Computers in Human Behavior; Cyberpsychology, Behavior, and Social Networking; Deviant Behavior; Journal of Adolescence; Journal of Pediatric Nursing; Journal of School Violence; New Media and Society; School Psychology Review; Trauma, Violence, & Abuse). The journals were selected based on search strategies of previous meta-analyses on the topic (Kowalski et al., 2014) ) as well as by selecting journals that had recently published meta-analyses on the field of cyberbullying. To supplement the research, two other sources (ProQuest Dissertations and Theses, Google Scholar) were also searched to retrieve additional published literature as well as relevant unpublished literature.

The following keywords were used to conduct the systematic literature search within the five databases: ("meta-analy*" OR "meta analy*" OR "quantitative synthesis" OR "review*") AND (cyber* OR internet OR net OR online OR chat OR electronic OR mobile OR "social network" OR media OR Facebook OR Twitter OR Blog* OR Youtube OR Tumblr OR Discord OR Reddit OR Instagram OR Tiktok OR Snapchat OR Pinterest OR LinkedIn) AND (harass* OR bully* OR bulli* OR victim* OR aggres* OR abus* OR maltreat* OR incivil* OR toxic* OR violen* OR delinquen* OR devian* OR ragging OR hazing OR mobbing OR intimidat*). A simplified search string containing the following keywords was used to search the relevant journals and other sources: (meta-analysis OR "meta analysis" OR review) AND (cyber OR internet OR online OR "social media") AND (bully OR victim).

| Data collection | The initial search returned 1583 records, of which 1149 remained after the removal of duplicates. Title and abstract screening resulted in the removal of a further 818 records. Full text-screening resulted in the removal of 331 records, leaving a final total of 56 records.<br><br>The following information was independently extracted from the final list of included meta-analyses by either the first and third author or the first author and a research assistant: author(s), year of publication, title of publication, countries and regions covered by the review, participant demographics, total number of studies, total unique sample size, cyberbullying definition and type of cyberbullying victimisation measured, predictors and/or consequences of cyberbullying victimisation, and the relevant effect sizes denoting the association between cyberbullying victimisation and the predictor and/or consequence of cyberbullying victimisation explored within each meta-analysis. Regional classification of the different countries followed the listing by Wikimedia, Meta-Wiki (2022). Effect sizes were extracted as given within each meta-analysis without any conversions. |
|---|---|
| Timing | The literature search was conducted for all papers published up to 7th April 2024. |
| Data exclusions | No data were exlucded |
| Non-participation | No participants dropped out. |
| Randomization | Participants were not allocated to experimental groups. |

# Reporting for specific materials, systems and methods

We require information from authors about some types of materials, experimental systems and methods used in many studies. Here, indicate whether each material, system or method listed is relevant to your study. If you are not sure if a list item applies to your research, read the appropriate section before selecting a response.

## Materials & experimental systems

| n/a | Involved in the study |
|---|---|
| ☒ | Antibodies |
| ☒ | Eukaryotic cell lines |
| ☒ | Palaeontology and archaeology |
| ☒ | Animals and other organisms |
| ☒ | Clinical data |
| ☒ | Dual use research of concern |
| ☒ | Plants |

## Methods

| n/a | Involved in the study |
|---|---|
| ☒ | ChIP-seq |
| ☒ | Flow cytometry |
| ☒ | MRI-based neuroimaging |

# Plants

| Seed stocks | *Report on the source of all seed stocks or other plant material used. If applicable, state the seed stock centre and catalogue number. If plant specimens were collected from the field, describe the collection location, date and sampling procedures.* |
|---|---|
| Novel plant genotypes | *Describe the methods by which all novel plant genotypes were produced. This includes those generated by transgenic approaches, gene editing, chemical/radiation-based mutagenesis and hybridization. For transgenic lines, describe the transformation method, the number of independent lines analyzed and the generation upon which experiments were performed. For gene-edited lines, describe the editor used, the endogenous sequence targeted for editing, the targeting guide RNA sequence (if applicable) and how the editor was applied.* |
| Authentication | *Describe any authentication procedures for each seed stock used or novel genotype generated. Describe any experiments used to assess the effect of a mutation and, where applicable, how potential secondary effects (e.g. second site T-DNA insertions, mosaicism, off-target gene editing) were examined.* |

