## [Peer Review File · Nature Human Behaviour]

Peer Review Information

Journal: Nature Human Behaviour

Manuscript Title: Umbrella Review of Meta-Analyses on the Risk Factors, Protective Factors, Consequences, and Interventions of Cyberbullying Victimization

Corresponding author name(s):

Reviewer Comments & Decisions:

Decision Letter, initial version:
--

7th March 2024

Dear Ms Kasturiratna,

Thank you once again for your manuscript, entitled "Systematic Review of Meta-Analyses on Cyberbullying Victimization: A Critical Synthesis of Risk & Protective Factors, Consequences, and Interventions," and for your patience during the peer review process.

Your manuscript has now been evaluated by 2 reviewers, whose comments are included at the end of this letter. While we had recruited a third reviewer, they unfortunately never submitted comments. Although the reviewers find your work to be of interest, they also raise some important concerns. We are very interested in the possibility of publishing your study in Nature Human Behaviour, but would like to consider your response to these concerns in the form of a revised manuscript before we make a decision on publication.

To guide the scope of the revisions, the editors discuss the referee reports in detail within the team, including with the chief editor, with a view to (1) identifying key priorities that should be addressed in revision and (2) overruling referee requests that are deemed beyond the scope of the current study. We hope that you will find the prioritised set of referee points to be useful when revising your study. Please do not hesitate to get in touch if you would like to discuss these issues further.

1. Reviewer 1 points out that one relevant meta-analysis appears to have been missed in your search. Please re-examine your search terms (and the journals included in your search -- see Reviewer 2's relevant comment) and re-run your search to ensure that all relevant meta-analyses published until the receipt date of this decision letter are included in your umbrella review.

2. Reviewer 2 points out that some key items in the PRISMA checklist have not been attended to (and do not appear to have been reported in your manuscript). Please ensure that your PRISMA checklist is fully completed and that an explanation as to why a certain item is not reported is provided both in the checklist and within the manuscript itself.

3. We ask that you streamline your Introduction and Discussion section to focus on cyberbullying in specific. Please follow Reviewer 1's suggestion and discuss individual and external predictors separately and motivate the adaptation of a broad definition of cyberbullying.

Finally, your revised manuscript must comply fully with our editorial policies and formatting requirements. Failure to do so will result in your manuscript being returned to you, which will

delay its consideration. To assist you in this process, I have attached a checklist that lists all of our requirements. If you have any questions about any of our policies or formatting, please don't hesitate to contact me.

In sum, we invite you to revise your manuscript taking into account all reviewer and editor comments. We are committed to providing a fair and constructive peer-review process. Do not hesitate to contact us if there are specific requests from the reviewers that you believe are technically impossible or unlikely to yield a meaningful outcome.

We hope to receive your revised manuscript within two months. I would be grateful if you could contact us as soon as possible if you foresee difficulties with meeting this target resubmission date.

- Include a "Response to the editors and reviewers" document detailing, point-by-point, how you addressed each editor and referee comment. If no action was taken to address a point, you must provide a compelling argument. When formatting this document, please respond to each reviewer comment individually, including the full text of the reviewer comment verbatim followed by your response to the individual point. This response will be used by the editors to evaluate your revision and sent back to the reviewers along with the revised manuscript.
- Highlight all changes made to your manuscript or provide us with a version that tracks changes.

[REDACTED]

We look forward to seeing the revised manuscript and thank you for the opportunity to review your work. Please do not hesitate to contact me if you have any questions or would like to discuss these revisions further.

Sincerely,

[REDACTED]

Reviewer expertise:

Reviewer #1: antecedents and consequences of cyberbullying ; evidence synthesis

Reviewer #2: antecedents and consequences of cyberbullying ; evidence synthesis

Reviewer #3: AWOL

REVIEWER COMMENTS:

Reviewer #1:

Remarks to the Author:

This review has potential to have a significant impact on the field because it is a comprehensive overview of the predictors and consequences of cyberbullying. The following recommendations are meant as constructive feedback to improve the manuscript.

1. Growth has increased cyberbullying—this statement is imprecise to assume growth because internet use is increasing. Some researchers have found that cyberbullying is decreasing over time.
2. If the paper is really focused on cyberbullying, the first para. on general internet use doesn't seem needed, except maybe the end of it.
3. Consider separating individual predictors and family/school predictors in the introduction, and maybe adding in some subheadings to separate these categories out visually.
4. Generally, adding in headings would make the introduction easier to read.
5. In a comprehensive integration of cyberbullying literature, it seems important to acknowledge differences between cyberbullying and bullying as well as measurement issues that may have affected prevalence and outcomes across cyberbullying studies. Your review takes a very broad definition of cyberbullying. These decisions need to be justified. For example including intimate partner violence perpetrated online, may typically be under stalking literature or sexual harassment literature. Why do you include it in cyberbullying?
6. Additionally, the majority of the papers focus on children and youth. The decision to include adults should be justified, and developmental considerations should be noted.
7. If Routine Activity Theory guided your study (it's brought up in the discussion a couple times), it should be introduced in the introduction.
8. It would be helpful to explicitly identify research questions at the end of the introduction.

Methods

9. It would be important to define umbrella review at the start of the methods.
10. On pg 10, line 195, it would be good to specify that this is a list of preliminary inclusion criteria.
11. Use of PRISMA is excellent. Was this review registered? The search strategy and search terms are fairly comprehensive, though at least one meta-analysis was not included: "The dosage, context, and modality of interventions to prevent cyberbullying perpetration and victimization: A systematic review."
12. Interventions were called out in the abstract—were they conceptualized as predictors of cyberbullying victimization? It would be good to make that explicit in the description of inclusion criteria.
13. Good use of Joanna Briggs Institute (JBI) critical appraisal instrument.
14. School and peer related outcomes seem like they would be two separate domains rather than one domain, but I understand that there were few of each. Is there a better way to label this?
15. On pg. 13, lines 260-263, by overall agreement rate, do you mean average for each record?
16. Per APA guidelines, please limit repetition of material that is in the tables and figures (e.g., prisma flowchart)

Results

17. Some studies include the age range in the table others do not—this would be important to include or note with NA that age range was not included.
18. The finding of age needs contextualization by including the developmental stage of samples that the three records analyzed—no adult studies were included in this finding, and it seems unlikely that this finding would apply to adults. You discuss this later, but it's important to set the discussion up by including a brief description of the studies. Also, it would seem to be important to report the proportion of studies that found each outcome. For example, $\frac{3}{4}$ found age differences—all of these had adolescent and young adult samples.

Discussion

19. I recommend starting with fifth sentence about the current study (line 528)—the first two sentences over generalize and the next two justify the study, which was done in the introduction. It's better to get right to the point, especially in the discussion. This first paragraph should cover

what you did in this study and what you found.

20. The discussion is very long, and it rehashes out well-known findings (e.g., gender, prior cyberbullying victimization). I suggest focusing on the main 3-4 findings that are novel and make sure to be consistent in commenting on the size of effect sizes for each outcome you discuss. Then, you could close with paragraphs that 1. sums up additional demographic/victimization findings in the review and 2. implications of findings.

21. Right now, the review feels like it fails to address the “so what” question at the end—what good is this information? It seems strange to include research gaps in the same section as limitations—this should be separated out.

22. Minor—editing for grammar (e.g., comma use) is needed throughout the manuscript.

Reviewer #2:

Remarks to the Author:

See attached comments

Reviewer #3:

None

Author Rebuttal to Initial comments

Ms. Ref. No.: NATHUMBEHAV-23103327

Title: A Systematic Review of Meta-Analyses on the Risk Factors, Protective Factors, Consequences, and Interventions of Cyberbullying Victimization

We thank the Reviewers for their thorough and incisive reviews, which have significantly strengthened the revised paper. We strove to carefully address both Reviewers’ concerns and suggestions and hope that this revision is much stronger. Our point-by-point responses are listed and detailed below, along with the Reviewers’ comments.

Thank you once again for your consideration of our paper.

Reviewer #1

This review has potential to have a significant impact on the field because it is a comprehensive overview of the predictors and consequences of cyberbullying. The following recommendations are meant as constructive feedback to improve the manuscript.

1. *Growth has increased cyberbullying—this statement is imprecise to assume growth because internet use is increasing. Some researchers have found that cyberbullying is decreasing over time.*

Response:

We thank the Reviewer for the positive feedback and all the constructive comments, which we have incorporated in our revision to enhance the quality of our manuscript. We agree with the reviewer that increase in internet use does not necessarily increase cyberbullying prevalence, and have edited our abstract to reflect this (p. 2). We have also clarified in our Introduction that increasing internet use does not always lead to increased cyberbullying prevalence (p. 3). Specifically, we elaborate that the increasing prevalence of cyberbullying with greater digital media use does not uniformly indicate that more internet usage directly leads to higher instances of cyberbullying (Jang et al., 2014; Livingstone et al., 2016). Indeed, the relationship between increased digital activity and cyberbullying is influenced by a variety of factors, such as digital literacy (Seçkin Kapucu et al., 2021; Tao et al., 2022), social support networks (Aoyama et al., 2011; Elgar et al., 2014; Sampasa-Kanyinga et al., 2020), and the effectiveness of preventative measures (Gaffney et al., 2019; Ng et al., 2022; Polanin et al., 2022) which can vary widely across different social and cultural contexts, thus making it essential to understand the nuanced mechanisms that facilitate cyberbullying.

- 2. If the paper is really focused on cyberbullying, the first para. on general internet use doesn't seem needed, except maybe the end of it.*

Response:

We thank the Reviewer for the comment. Following the Reviewer's suggestion, we have revised the first paragraph of the Introduction to be more concise and focus directly on cyberbullying (p. 3). The revised paragraph is included below:

Amidst rapid technological advancements, the internet and social media have become prevalent platforms for social interaction, particularly among youth and adolescents (Guan & Subrahmanayam, 2009; Petrič, 2006; Wong et al., 2022). These digital environments, while fostering connections and personal expression (Bailey et al., 2020; Best et al., 2014; Brown & Greenfield, 2021; Subrahmanyam & Greenfield, 2008; Subrahmanyam & Šmahel, 2010), also present new challenges (Joinson et al., 2007), including cyberbullying—a significant and growing concern as digital interactions increase (Völlink et al., 2016). Referring to intentional acts of aggression carried out repeatedly via online or electronic media against an individual or a group of individuals (Wade & Beran, 2011), cyberbullying has become a commonplace social issue in recent times (Agustina, 2015; Wang et al., 2019). In the US, nearly half of adolescents have experienced at least one instance of cyberbullying victimisation (Vogels, 2022). Globally, around 4 in 10 adults who use the internet have experienced cyberbullying victimisation (Petrosyan, 2022). Within Asia, countries such as Singapore, China, Malaysia, and South Korea all report high prevalence rates close to 50% (Kamaruddin et al., 2023; Kwan & Skoric, 2013).

3. *Consider separating individual predictors and family/school predictors in the introduction, and maybe adding in some subheadings to separate these categories out visually.*

Response:

We thank the Reviewer for the helpful suggestion. Following the Reviewer's suggestion, we have restructured the content (pp. 7-8) to distinctly categorise individual and external predictors of cyberbullying, enhancing the clarity of our literature review. Each predictor is now discussed in detail under specific subheadings, providing a clear distinction that aids in understanding their individual impacts. For example, under Sociodemographic and Personality Predictors (p. 7), we discuss that research has shown that females and minorities were more likely to be subjected to cyberbullying victimisation (Foody et al., 2017; Caridade & Braga, 2020; Oblad, 2012; Sun & Fan, 2018).

Personality traits also contribute to the likelihood of being cyberbullied; attributes like neuroticism, antisocial personality, and low agreeableness can increase vulnerability by affecting how individuals interact online and perceive hostile interactions (Wissink et al., 2023). There is also evidence that individuals with higher levels of anxiety, depression, and anger were more likely to become victims of cyberbullying (Kokkinos et al., 2014; Kowalski et al., 2014; Pabian & Vandebosch, 2016), as they tend to be distanced from social groups and resort more to online media, increasing the likelihood of encountering cyberbullies (O'Day & Heimberg, 2021; Olenik-Shemesh et al., 2012).

Separately, under Contextual and Environmental Predictors (p. 7), we discuss that research examining contextual predictors found that variables such as unregulated family and school climates, as well as unrestricted internet use were prominent contextual risk factors that were associated with higher levels of cyberbullying victimisation (Arató et al., 2022; Balakrishnan, 2015; Caridade & Braga, 2020; Chen et al., 2017). Unregulated home and school environments provide vulnerable targets and allow cyberbullies to perpetrate unrestrainedly in the absence of parental guardians or

teacher, consistent with the Routine Activity Theory that posits that deviant behaviours such as cyberbullying occur in the presence of motivated offenders, suitable targets and an absence of capable guardians (Aizenkot, 2022; Akgül, 2023; Kalia & Aleema, 2017). The Routine Activity Theory suggests that the lack of effective supervision from the environment increases the opportunity for cyberbullying, emphasising the importance of considering environmental factors as a predictor of cyberbullying victimisation.

In addition, we have also elaborated on interventions as a predictor of cyberbullying victimisation (p. 8). Within the Introduction, we discuss how many intervention programs focus on educating individuals about cyberbullying and equipping them with coping strategies to handle its risk factors (Chen et al., 2023; Lan et al., 2022). Additionally, some studies also highlight various program types that incorporate digital interventions (Chen et al., 2023) and emphasise the involvement of specific social groups, such as families and parents (Wang & Jiang, 2023). However, the effectiveness of these anti-cyberbullying programs remains uncertain, as indicated by previous reviews that report mixed results (Mishna et al., 2011; Polanin et al., 2022), which necessitates the synthesising of results across different interventions to arrive at more robust conclusions.

4. *Generally, adding in headings would make the introduction easier to read.*

Response:

Thank you for your valuable suggestion to add headings within the introduction to enhance its readability. We agree that structured headings can significantly improve the navigation and comprehension of the content. Following the suggestion, we have incorporated clear and descriptive headings throughout the introduction (pp. 3-8). Specifically, we have added the heading ‘Defining

Cyberbullying’ to the section that elaborates on the definition of cyberbullying (p. 3), followed by the heading “Cyberbullying vs Traditional Bullying” to the subsequent section that discusses the differences between cyberbullying and traditional bullying (p. 4). Next, for the section that discusses common issues in cyberbullying measures, we have added the heading “Measurement Issues in Cyberbullying Research” (p. 5). The headings “Sociodemographic and Psychological Predictors” (p. 7), “Contextual and Environmental Predictors” (p. 7), “Psychological and Behavioural Consequences” (p. 8), and “Effectiveness of Interventions” (p. 8) were added for the next four sections where the four major research areas within cyberbullying victimisation research were discussed.

- 5. In a comprehensive integration of cyberbullying literature, it seems important to acknowledge differences between cyberbullying and bullying as well as measurement issues that may have affected prevalence and outcomes across cyberbullying studies. Your review takes a very broad definition of cyberbullying. These decisions need to be justified. For example including intimate partner violence perpetrated online, may typically be under stalking literature or sexual harassment literature. Why do you include it in cyberbullying?*

Response:

We thank the Reviewer for the constructive suggestion. Following the Reviewer’s suggestion, we have expanded our Introduction section (pp. 3-6) to discuss the definition of cyberbullying, as well as differences from traditional bullying and measurement issues.

Firstly, we note that there is currently no consensus within research on the precise definition of cyberbullying (Alipan et al., 2015; Kowalski et al., 2014). However, there are some universally accepted elements. First, it is widely recognised that cyberbullying involves bullying using electronic

media (Kowalski et al., 2014). The term "electronic media" itself is broad, and while some definitions restrict it to the use of the internet and mobile phones (Almeida et al., 2012; Aricak et al., 2008; Brighi et al., 2012; Didden et al., 2009), others apply a more detailed taxonomy of technology (Agatston et al., 2007; Dempsey et al., 2009). Given the rapid evolution of digital media, it is pragmatic to adopt a broader definition that encompasses both current technologies and any forthcoming developments by using a definition such as 'actions carried out via any electronic means' rather than specifying devices through which cyberbullying can occur.

Next, it is also generally agreed that cyberbullying involves a form of aggression directed toward an individual or a group (John et al., 2018; Kowalski et al., 2014). However, different studies operationalise the type of aggression differently. For instance, a majority of research identifies cyberbullying through behaviours such as sending aggressive or threatening messages online (Fisher et al., 2016; Guo, 2016; John et al., 2018; Kowalski et al., 2014). In contrast, Mills et al. (2024) operationalised cyberbullying as online social exclusion. Willard (2007) developed a comprehensive taxonomy of cyberbullying that includes flaming (online arguments), harassment (sending repetitive, offensive messages), outing and trickery (distributing someone's personal information without consent), exclusion (blocking someone from digital social circles), impersonation (posing as someone else to communicate derogatively), cyber-stalking (sending repetitive threatening communications online), and sexting (distributing nude pictures without consent).

Taking all these perspectives into account, the current work opted to define cyberbullying as any aggressive or bullying behaviour aimed directly at an individual or a group using any electronic means. This definition encompasses aspects such as sextortion (threatening to use an explicit photo or video of someone to make demands or pressure them; O'Malley & Holt, 2022), online social exclusion (excluding an individual via blocking or distancing over online means; Willard, 2007), and

cyberdating abuse (a form of control and harassment by the dating partner using electronic media; Zweig et al., 2014), as these behaviours also involve acts of aggression via electronic media (pp. 3-4).

We also elaborate on the differences between traditional bullying and cyberbullying within the Introduction (pp. 4-5). We note that it is widely accepted that cyberbullying is an extension of traditional bullying (Alipan et al., 2020), with many researchers modelling their definitions of cyberbullying on the main characteristics traditionally associated with aggression (Englander et al., 2017): intention, repetition, and power imbalance (Olweus, 1995). While there is a high correlation between traditional bullying and cyberbullying (Erdur-Baker, 2010; Hinduja & Patchin, 2008; Kowalski et al., 2014) and both involve aggressive acts, significant differences exist between the two. Firstly, the intention behind cyberbullying can often be ambiguous due to the lack of non-verbal cues; actions perceived as humorous by the perpetrator might be interpreted as hurtful by the recipient (Alipan et al., 2015; Vandebosch & Van Cleemput, 2008). Secondly, the concept of repetition differs within the online realm; a perpetrator may only commit a single act of aggression to a victim, but that one post, comment, or image does not need to be reposted by the original perpetrator to be considered repetitive (Slonje & Smith, 2008; Smith, 2011). It can instead be shared or forwarded by others, continually harming the victim without further direct action from the perpetrator. Thirdly, a power imbalance is not always a prerequisite for cyberbullying (Alipan et al., 2020). Due to the anonymity afforded by digital platforms and the lack of physical confrontation, individuals who may not typically engage in face-to-face bullying can easily perpetrate online harassment (Vandebosch & Van Cleemput, 2008). While research on cyberbullying has attempted to define power imbalance in terms of technological savvy or digital literacy (Langos, 2012), this may not necessarily confer a significant advantage in the current environment as the proliferation of various platforms and their ease of use has simplified the act of bullying online (Alipan et al., 2015).

Most crucially, cyberbullying eliminates the need for face-to-face interaction, often allowing perpetrators to remain anonymous (Barlett, 2015; Barlett et al., 2016). According to Suler's 2005 online disinhibition effect theory, the Internet, which offers anonymity by allowing users to adopt contrived usernames, grants individuals the power to separate their online actions from their offline identity. This reduces the sense of responsibility for their online actions and motivates perpetrators to engage in more cyberbullying, which then increases the possible incidence of victimisation (Suler, 2005) and allows victims themselves to become cyberbullies in the future (Holfeld & Mishna, 2019; Pabian & Vandebosch, 2016). As such, it is imperative to better understand cyberbullying as a phenomenon distinct from traditional bullying in order to prevent creating a vicious cycle of internet-based aggressive behaviour that perpetuates negative consequences.

Lastly, we also appreciate the Reviewer's suggestion to discuss measurement issues within the field of cyberbullying, and agree that this would significantly enhance the quality of our review. Following the suggestion, we have elaborated on challenges with regards to measuring cyberbullying research in our Introduction (pp. 5-7). Specifically, we elaborate on how the absence of a unified definition complicates measurement as studies often adopt divergent definitions and employ varying scales that may not fully capture the phenomenon. For instance, some studies limit cyberbullying to online peer victimisation (Dempsey et al., 2009; Smith, 2019), while others do not (Agatston et al., 2007; Almeida et al., 2012). Additionally, older studies frequently omit a clear definition of bullying (Berne et al., 2013; Kowalski et al., 2014), and while newer studies tend to provide one, they vary significantly in terminology, using terms like 'cyberaggression,' 'cyberstalking,' or 'cyberbullying,' which can confuse respondents and hinder comparability across studies (Chun et al., 2020). The development of cyberbullying scales also shows inconsistencies, with many not adhering to recommended guidelines for item development and only about half reporting validity statistics (Chun et al., 2020). Moreover, the rapid evolution of digital communication platforms continually outdates

traditional cyberbullying scales that may not focus on newer methods of cyberbullying (Menesini et al., 2011).

6. *Additionally, the majority of the papers focus on children and youth. The decision to include adults should be justified, and developmental considerations should be noted.*

Response:

We thank the reviewer for the constructive comment regarding the focus of the reviewed papers primarily on children and youth, and the inclusion of studies involving adults in our systematic review. We recognise the importance of justifying the inclusion of all age groups to encompass the full spectrum of cyberbullying experiences. In response to the feedback, we have further elaborated on the reasons for including studies on adults in the Introduction section of our manuscript (p. 7).

In the revision, we acknowledge that children, adolescents, and adults can experience and interpret cyberbullying in fundamentally different ways due to their developmental cognitive and social capacities (Cohen-Almagor, 2018). For example, younger children may lack the emotional maturity to accurately identify and report cyberbullying incidents (Walker et al., 2013), whereas adolescents and young adults, as they become more integrated with society, may both experience it more and also be able to identify it (Cohen-Almagor, 2018). Adults, on the other hand, might interpret interactions differently based on life experiences and maturity, influencing their responses to potential cyberbullying scenarios (Barlett & Chamberlin, 2017; M.-J. Wang et al., 2019). This variability across age groups necessitates the synthesising of unique and common factors of cyberbullying to develop more robust cyberbullying measures and identify universally applicable predictors and consequences. Although a significant majority of cyberbullying research concentrates on children to youth, the

phenomena also significantly impact adults. This inclusion of both younger samples as well as adult samples allows our review to provide a more comprehensive overview of cyberbullying across the lifespan.

7. *If Routine Activity Theory guided your study (it's brought up in the discussion a couple times), it should be introduced in the introduction.*

Response:

We thank the Reviewer for this comment. We agree that introducing this theoretical framework in the introduction would provide a clearer foundation for understanding the context of our review.

Following the suggestion, we have integrated an explanation of the Routine Activity Theory in the Introduction (pp. 7-8). Specifically, in the Introduction (p. 7), we elaborate that the Routine Activity Theory asserts that deviant behaviours, such as cyberbullying, are more likely to occur when there are motivated offenders, suitable targets, and an absence of capable guardians (Aizenkot, 2022; Akgül, 2023; Kalia & Aleema, 2017). Furthermore, we highlighted that this framework is critical for understanding the environmental conditions that facilitate cyberbullying, as it highlights the interplay between offender motivation, target vulnerability, and the lack of supervision or intervention by responsible figures. Factors such as unregulated home and school environments create a context conducive to cyberbullying, as these settings often lack adequate supervision, which, according to the Routine Activity Theory, allow cyberbullies to act with impunity. The absence of capable guardians—be they parents, teachers, or institutional safeguards—significantly increases the risks associated with becoming a target of cyberbullying. This theoretical perspective helps elucidate why certain environments are more prone to cyberbullying incidents, reinforcing the need for proactive measures to enhance supervision and guardianship in home and school environments.

8. *It would be helpful to explicitly identify research questions at the end of the introduction.*

Response:

We thank the reviewer for the helpful suggestion. We agree that clearly specifying the research questions would enhance the structure of our manuscript. We have revised the introduction to include a clear statement of the research questions that guide our systematic review. These questions are now clearly listed at the end of the introduction to ensure they are both visible and impactful to our readers (p. 9). Specifically, we address the following three critical questions: Specifically, this review will address the following critical questions: 1) What are the crucial sociodemographic and psychological profiles of cyberbullying victims? 2) What critical contextual and environmental factors are associated with cyberbullying victimisation? 3) What are the key psychological and behavioural consequences of cyberbullying for victims? 4) How effective are existing interventions in mitigating the impacts of cyberbullying? These questions aim to summarise the core focus of our review and guide the subsequent sections of the manuscript.

Methods

9. *It would be important to define umbrella review at the start of the methods.*

Response:

We thank the Reviewer for this helpful suggestion. We agree that clearly explaining our methodology at the outset would enhance the clarity of the manuscript. Following the suggestion, we have revised the Method section (p. 22) to include a detailed explanation of what constitutes an umbrella review.

Specifically, in our Method section (p. 22), we have elaborated that an umbrella review is a distinct form of systematic review that compiles data from multiple meta-analyses addressing the same research questions. This methodology allows for a comprehensive synthesis of evidence across studies, which is crucial for enhancing our understanding by comparing and contrasting results from different meta-analyses. By aggregating findings across these studies, an umbrella review helps identify patterns, strengths, and gaps in the literature, providing a robust analysis of extensive datasets. This approach is particularly suitable for fields like cyberbullying victimisation, which involves a vast array of studies with varying outcomes, allowing us to distil broad insights from diverse research findings effectively.

10. On pg 10, line 195, it would be good to specify that this is a list of preliminary inclusion criteria.

Response:

We thank the Reviewer for the helpful suggestion. We have edited the manuscript to specifically indicate that the titles and abstracts were evaluated based on a preliminary set of criteria (p. 24).

11. Use of PRISMA is excellent. Was this review registered? The search strategy and search terms are fairly comprehensive, though at least one meta-analysis was not included: “The dosage, context, and modality of interventions to prevent cyberbullying perpetration and victimization: A systematic review.”

Response:

We thank the reviewer for this important comment. With regards to the pre-registration, it was not completed for this investigation. Preregistration is indeed a best practice that enhances transparency and reduces the risk of bias by pre-defining research questions and analysis strategies. We acknowledge this oversight and understand that preregistration could have added an additional layer of rigor to our systematic review. We have acknowledged in our Transparency and Openness section that the current review was not pre-registered (p. 21). We appreciate the reviewers pointing out this omission, and we will consider preregistration for future systematic reviews to better align with contemporary research standards.

We also thank the Reviewer for pointing out the meta-analysis by Doty et al. (2022), which was omitted in our previous literature search. We conducted an additional round of searches within the selected databases (*EBSCOhost ERIC, EBSCOhost PsycInfo, PubMed, Scopus, Web of Science*), selected journals (*Adolescent Research Review; Aggression and Violent Behavior; Aggressive Behavior; Children and Youth Services Review; Computers in Human Behavior; Cyberpsychology, Behavior, and Social Networking; Deviant Behavior; Journal of Adolescence; Journal of Pediatric Nursing; Journal of School Violence; New Media and Society; School Psychology Review; Trauma, Violence, & Abuse*), Google Scholar, and ProQuest Dissertations and Theses to identify relevant meta-analyses missed in our previous search (p. 23). We also attempted to incorporate journals that are more relevant to today's scholarship on cyberbullying by including six new journals (*Adolescent Research Review; Aggression and Violent Behavior; Children and Youth Services Review; Deviant Behavior; Journal of Pediatric Nursing; Trauma, Violence, & Abuse*) that recently published meta-analyses within the field of cyberbullying (p. 23).

The updated search resulted in the retrieval of 373 new records out of which 63 records were identified as duplicates and removed, resulting in 310 records to be screened for potential inclusion. Title and abstract screening was conducted by the first author and a research assistant (92.76% inter-

rater agreement), and resulted in the removal of a further 219 records. For remaining 91 records, full-text screening was conducted by the first author and a research assistant (95.23% inter-rater agreement). Another 69 records were eliminated during full-text screening, leaving a total of 22 new records (overall 56 records) to be included in the final review. The PRISMA flow-chart with the updated numbers (p. 27) incorporating both the original search as well as the updated search is included below (Figure 1).

Following the search, quality assessment was conducted for the newly included records using the Joanna Briggs Institute critical appraisal instrument for Systematic Reviews and Research Syntheses (JBI, 2017). The quality of each newly included meta-analysis was assessed independently by the first author and a trained research assistant (inter-rater agreement 94%; range 90–100%). Finally, data extraction for the new records was conducted by the first author and a trained research assistant (inter-rater agreement range=81.54%–100%). We have further updated our results (pp. 10-14) and discussion (pp. 14-19) to reflect the changes from the updated search.

Figure 1

PRISMA Flowchart

12. Interventions were called out in the abstract—were they conceptualized as predictors of cyberbullying victimization? It would be good to make that explicit in the description of inclusion criteria.

Response:

We thank the reviewer for this important comment, and agree that the role of interventions should indeed be explicitly conceptualized as predictors of cyberbullying victimisation. To clarify this in our manuscript, we have amended the inclusion criteria (p. 25) to specify that interventions aimed at preventing cyberbullying were also considered predictors, as they are designed to reduce the incidence or impact of cyberbullying, and therefore may influence the likelihood or impact of an individual's cyberbullying victimisation experience.

13. Good use of Joanna Briggs Institute (JBI) critical appraisal instrument.

Response:

We thank the review for the comment, and appreciate your positive feedback on our work.

14. School and peer related outcomes seem like they would be two separate domains rather than one domain, but I understand that there were few of each. Is there a better way to label this?

Response:

We thank the reviewer for this comment. Following the suggestion, we have reconsidered the categorisation of school and peer-related outcomes. The outcomes investigated within this category—peer relationship quality, prosocial behaviour, academic achievement, school attendance, and traditional bullying victimisation—are all from child or adolescent samples and are inherently linked to the school context, affecting and reflecting both academic and social dimensions of school life. Therefore, we have renamed the category as 'School-Related Academic and Social Outcomes' (p. 32) to more accurately represent these intertwined aspects. This label not only captures the dual focus on academic performance and social interactions within schools but also underscores how these elements are often influenced by similar factors, such as the school environment and peer relationships. By using this unified category, we hope that our synthesis on how cyberbullying victimisation impacts various facets of school life becomes clearer.

15. On pg. 13, lines 260-263, by overall agreement rate, do you mean average for each record?

Response:

We thank the reviewer for the comment. We have specified that the overall agreement rate refers to the inter-rater agreement between the two coders on average across all criteria of the JBI critical appraisal instrument (p. 28).

16. Per APA guidelines, please limit repetition of material that is in the tables and figures (e.g., prisma flowchart)

Response:

We thank the reviewer for this helpful comment. We agree that reducing redundancy can enhance the readability and conciseness of our manuscript. Following the suggestion, we have revised the manuscript to ensure that the descriptions of data presented in our tables and figures, particularly the PRISMA flowchart, are more succinctly summarized in the text (p. 10). For instance, the revised text now briefly outlines the search and selection process as follows: "The initial search returned 1583 records, of which 1149 remained after the removal of duplicates. Title and abstract screening resulted in the removal of a further 818 records. Full text-screening resulted in the removal of 331 records, leaving a final total of 56 records included in the review (for full details refer to PRISMA flowchart in Method section). (p. 10)"

Results

17. Some studies include the age range in the table others do not—this would be important to include or note with NA that age range was not included.

Response:

We thank the Reviewer for this comment. We agree that understanding age-related nuances is crucial in the context of cyberbullying research, and have revised the tables to ensure that the age range of participants is consistently reported (pp. 34-50). Where age data were not provided by the original studies, we have noted this as 'NA' to maintain clarity and uniformity across the dataset. The table is included below for ease of reference:

Table 4*Characteristics of the 56 Included Meta-Analyses*

Author, Publication year	Document type	Country, Region	Number of studies	Sample size	Participants	Number of sources searched	Date range of search	Date range of included studies	Research objective
Abregú-Crespo et al., 2024	Journal article	Europe, North America (73%)	212	631,523	Children and adolescents (aged 4-17 years) with neurodevelopmental or psychiatric conditions	6 databases (ERIC, PsycArticles, PsycInfo, Psychology and Behavioural Sciences Collection, PubMed, Web of Science Core Collection)	Up to August, 2023	NA	Assess the odds of bullying involvement and its association with mental health measures in these populations
Barlett et al., 2024	Journal article	62 countries (Africa, Asia-Pacific, Australia, Europe, Middle East, North America, South/Latin America)	211	1,136,080	Youth and adult populations (mean age 9.00-37.04 years)	13 databases (Academic Search Complete, Business Source Complete, Communication & Mass Media Complete, Criminal Justice Abstracts, Education Research Complete, Family Studies Abstracts, HealthSource: Nursing/Academic Edition, Human Resources Abstracts,	NA	2007–2022	Examine the correlations between cyberbullying and other variables while statistically controlling for traditional bullying.

MEDLINE, PsycArticles, PsycInfo, SocINDEX, Social Sciences Full Text), 15 journals (Aggressive Behavior; British Journal of Developmental Psychology; Computers in Human Behavior; Cyberpsychology, Behavior, and Social Networking; Developmental Psychology; European Journal of Developmental Psychology; Journal of Adolescence; Journal of Adolescent Health; Journal of School Psychology; Journal of School Violence; Journal of Youth and Adolescence; New Media & Society; School Psychology International; School Psychology Quarterly; School Psychology Review)

Caridade & Braga, 2020	Journal article	4 countries (Europe, North America)	16	12,760	Adolescents and young adults (mean	9 databases (Academic Research Complete, Business Source Complete, Complementary Index,	Up to February, 2019	2014–2018	Identify risk and protective factors associated with youth cyber-dating abuse
-----------------	-------------------------------------	----	--------	------------------------------------	---	----------------------	-----------	---

					age 10-26 years)	EBSCOhost, ERIC, Psychology and Behavioral Sciences Collection, PubMed, Science Direct, Scopus, Social Sciences Citation), reference lists of review articles, direct referrals by authors			
L. Chen et al., 2017	Journal article	Asia-Pacific, Europe, North America	81	99,741	Children, adolescents and adults (age range not included)	8 databases (Communication & Mass Media Complete, EBSCO, ERIC, MEDLINE, Nursing & Allied Health Source, PsycInfo, PubMed, Web of Science Direct) and reference lists of reviews	NA	2004–2015	Systematically examine the predictors of cyberbullying from a social cognitive and media effects approach
Q. Chen et al., 2023	Journal article	12 countries (Asia-Pacific, Australia, Europe, North America)	18	55,473	Primary school to college students (age range not included)	6 databases (EMBASE, ERIC, MEDLINE, PsycInfo, Social Service Abstracts, Sociological Abstracts)	Up to January, 2021	2000–2018	Examine and compare the effectiveness of digital health interventions (DHIs) in reducing bullying and cyberbullying
X. Chen et al., 2024	Journal article	25 countries (Africa, Asia-Pacific,	39	128,097	Adolescents (aged 10-18 years)	9 databases (ERIC, PsycArticles, PsycInfo, Psychology and Behavioral	Up to April, 2022	1993–2021	Examine the relation between experiences of bullying and

		Australia, Europe, Middle East, North America, South/Latin America)			without special attributes (e.g., clinical populations)	Sciences Collection, PubMed, ScienceDirect, Scopus, Web of Science, Wiley Online Library)			victimisation and life satisfaction among adolescents
Christina et al., 2021	Journal article	13 countries (Africa, Asia-Pacific, Europe, North America)	85	117,520	School-aged participants (younger children and young adults excluded) (mean age 6.3-16 years)	5 databases (Academic Search Premier, MEDLINE, PsychInfo, Scopus, Web of Science)	Up to May, 2020	1995–2020	Examine the bidirectional effects between internalising problems and peer victimisation within a meta-analytic framework
Doty et al., 2022	Journal article	11 countries (Australia, Europe, Middle East, North America)	30	48,548	Youth (aged up to 24 years)	7 databases (Academic Search Premier, Compendex, ERIC, PsycInfo, Psychology and Behavioural Sciences Collection, PubMed, Web of Science)	2010–2019	2011–2019	Describe cyberbullying preventative interventions in relation to intervention characteristics and risk of bias, qualitatively illustrate the dosage, modalities, and contexts of existing cyberbullying preventative interventions using harvest plots, and quantitatively examine the effect of cyberbullying preventative

interventions on perpetration and victimisation by dosage, modalities, and contexts

Eberle, 2023	Dissertation/ Thesis	USA (North America)	8	10,509	Minors (aged below 18 years) and adults (age range not included)	7 databases (ERIC, PAIS Index, ProQuest One Academic, PsycArticles, PsycInfo, PTSDpubs, Sociological Abstracts)	Up to August, 2023	2007–2022	Conduct a meta-analysis to identify the risk and protective markers of sextortion to identify points of prevention and intervention
Erbiçer et al., 2023	Journal article	Turkey (Europe)	59	34,968	Children and young people (aged 10-16 years)	8 databases (DergiPark, ERIC, National Thesis Center of Türkiye, ProQuest, PubMed, Scopus, Turkish Psychiatry Index, Web of Science)	Up to December, 2020	2012–2020	Conduct a systematic review and meta-analysis of the studies conducted in Turkey to clarify risk and protective factors and outcomes of cyberbullying perpetration and victimisation
Fisher et al., 2016	Journal article	NA	55	257,678	Adolescents (aged 12-18 years)	9 (ERIC, International Bibliography of the Social Sciences, ProQuest Central, ProQuest Dissertations and Theses Full Text, ProQuest Dissertations and Theses: UK and Ireland, PsycArticles, PsycInfo,	Up to 2016	2006–2016	Synthesise existing literature on the relationship between peer cybervictimisation and adolescents' internalizing and externalising problems

Social Services Abstracts,
Sociological Abstracts)

Gaffney et al., 2019	Journal article	10 countries (Australia, Europe, North America)	18	36,534	School-aged participants (aged 4-18 years)	Several databases (including DARE, ERIC, Google Scholar, PsycArticle, PsycInfo, Scopus, Web of Science), several journals (including Behavior and Social Networking, Computers in Human Behavior, Cyberpsychology)	2000–2018	2012–2018	Evaluate the effectiveness of existing cyberbullying intervention and prevention programmes
García-Hermoso et al., 2020	Journal article	89 countries (Africa, Arab States, Asia-Pacific, Australia, Europe, Middle East, North America, South/Latin America)	18	386,740	Children and young people (aged 10-16 years)	4 databases (EMBASE, ERIC, PsycArticles, PubMed), reference lists of retrieved articles quantitatively	NA	2007–2020	Provide a quantitative analysis on the associations of physical activity and sedentary behaviour on bullying victimisation among children and adolescents

Gardella et al., 2017	Journal article	USA (North America)	25	26,906	Adolescents (aged 12-17 years)	10 databases (Dissertations & Theses @ Vanderbilt University, ERIC, International Bibliography of the Social Sciences, ProQuest Central, ProQuest Dissertations & Theses: UK & Ireland, ProQuest Dissertations & Theses Full Text, PsycArticles, PsycInfo, Social Services Abstracts, Sociological Abstracts), conference proceedings, sites with technical reports, forward citation searches	1985– 2015	2009–2014	Quantitatively synthesise relationships between peer cybervictimisation and educational outcomes
Gilbar et al., 2023	Journal article	10 countries (Africa, Australia, Europe, Middle East, North America, South/Latin America)	44	27,491	Adults (aged above 18 years; mean age 18.06- 44.80 years)	7 (APA PsycArticles, Criminal Justice Abstracts, EBSCO, ERIC, MEDLINE, PsycInfo, Social Sciences Full Text (H.W. Wilson), Web of Science)	Up to 2019	2010–2020	Conduct an in-depth examination of the possibility of sex differences in different types of cyber-intimate partner violence, whether there are cyber-intimate partner violence associations to face-to-face intimate partner violence, and an exploration of sex differences in the associations between cyber-intimate partner violence and face-to-face intimate partner violence among adults

Gini et al., 2018	Journal article	Australia, Europe, North America	19	90,877	Children and adolescents (mean age 12.05-16 years)	8 databases (Dissertation Abstracts International, Open Access, ProQuest Dissertations and Theses Open, PsychInfo, Pubmed, Scopus, Web of Science), Google, reference sections of review, reference sections of collected articles, conference proceedings of the last biannual meetings of the Society for Research in Child Development and the Society for Research on Adolescence, and of the last four 'Workshop Aggression' (held in Europe)	Up to 2015	2009–2015	Summarise the relations of traditional and cyber-victimisation with internalising problems, and identify whether these types of peer victimisation were differentially related to such problems
Guo, 2016	Journal article	Asia-Pacific, Australia, Europe, Middle East, North America	77	129,278	Juveniles and young adults in school settings (aged 9-24 years)	7 databases (ERIC, Digital Dissertations, Google Scholar, National Criminal Justice Reference System, PsychInfo, PubMed, Web of Science), reference sections of review articles, direct referrals by authors	NA	2004–2013	Examine the relative magnitude of demographic, individual, and contextual predictors of cyberbullying perpetration and victimisation

Heerde & Hemphill, 2019	Journal article	22 countries (Asia-Pacific, Australia, Europe, Middle East, North America)	27	156,284	Adolescents (aged 11-19 years)	14 databases (Australian Criminology Database, Australian Criminology Database - Health Subset, CINAHL, ERIC, Health Source, MEDLINE, Nursing/Academic Edition, ProQuest Criminal Justice, ProQuest Education Journals, ProQuest Psychology Journals, ProQuest Social Science Journals, PschInfo, Psychology and Behavioural Sciences Collection, Social Work Abstracts, SocIndex)	1990–2018	2008–2017	Consolidate studies investigating associations between bullying (traditional and cyber-bullying perpetration and victimisation) and DSH in youth using a meta-analytic approach
Hu et al., 2024	Journal article	13 countries (Africa, Asia-Pacific, Australia, Europe, North America, South/Latin America)	57	53,183	Minors (aged below 18 years) and adults (age range not included)	8 databases (Chinese CNKI Database, EBSCO, Google Scholar, ProQuest Dissertations, PsycInfo, Springer, Wanfang database, Web of Science)	1999–2022	2002–2022	Examine the reliability of the effect size and a series of moderating effects between gender nonconformity and victimisation

Y. Hu et al., 2021	Journal article	17 countries (Asia-Pacific, Europe, North America)	57	105,440	Adolescents and young adults (mean age 10.85-24.67 years)	4 (APA PsycNet, Google Scholar, PsycInfo, PubMed)	Up to 2021	2009–2021	Examine the effect of cyberbullying and victimisation on depression
Huang et al., 2024	Journal article	15 countries (Arab States, Asia-Pacific, Australia, Europe, North America, South/Latin America)	126	73,191	Children, adolescents, and adults (aged 11-85 years)	11 databases (Chinese CNKI Database, Cochrane Library, EMBASE, EBSCO, ERIC, MEDLINE, PsycInfo, PubMed, Scopus, Wanfang database, Web of Science), reference list of included studies and review articles	Up to November, 2022	2020–2022	Examine the effect of the COVID-19 pandemic on cyberbullying, estimate the global cyberbullying prevalence and explore factors related to cyberbullying during the COVID-19 pandemic
John et al., 2018	Journal article	9 countries (Asia-Pacific, Australia, Europe, North America)	23	156,384	Adolescents and young adults (aged below 25 years; mean age 12.5-20 years)	6 databases (Cochrane Library, Medical Literature Analysis and Retrieval System Online, PROSPERO, PsycInfo, PubMed, Scopus) health improvement sources (e.g., Health Evidence Canada), topic-specific websites (e.g., American Association of Suicidology), meta-search engines	January, 1996–February, 2017	2009–2016	Systematically review and meta-analyse, when possible, the current evidence examining the association between cyberbullying involvement (as victim, perpetrator, or both) and self-harm and suicidal behaviours in children and young people (younger than 25 years)

(Google), direct referrals by authors

Kamaruddin et al., 2023	Journal article	2 countries (Australia, Thailand)	2	2,954	Non-school aged children (aged 8-29 years)	10 (Cambridge Journal Online, EBSCOHOST, ERIC, IEEE XPLORE, Oxford Journal Online, ProQuest Dissertations and Theses, PubMed (MEDLINE), Science Direct, Scopus, SpringerLink)	January, 1995– February, 2022	2013–2022	Empirically determine the effectiveness of programs with non-school-aged samples with a specific focus on studies conducted within the Asia-Pacific region
Killer et al., 2019	Journal article	13 countries (Asia-Pacific, Australia, Europe, North America)	49	43,809	Children and adolescents (aged 7-19 years)	3 (Google Scholar, ProQuest Dissertations and These Global, PsycInfo)	Up to 2018	2007–2018	Examine the relationship between moral disengagement and key bullying roles (i.e., defender, bystander, and victim)
Kowalski et al., 2014	Journal article	Asia-Pacific, Australia, Europe, North America	131	1,519 to 164,280	School and college students (aged 6-88 years)	14 databases, 15 journals, reference sections of review papers, direct referrals by authors	Up to 2012	2004–2013	Synthesise the relationships among cyberbullying, cybervictimisation, and meaningful behavioural and psychological variables

Lan et al., 2022	Journal article	7 countries (Australia, Europe)	19	31,924	Adolescents (aged 10-19 years)	18 databases (Academic Search Complete, Australian Education Index, British Education Index, Business Source Complete, CINAHL Plus, Cochrane Library, Communication & Mass Media Complete, Criminal Justice Abstracts, Education Database, ERIC, Google Scholar, MEDLINE, PsycArticles, PsycInfo, PubMed, Scopus, Sociological Abstracts, Web of Science)	Up to March, 2020	2012–2019	Examine the effectiveness of anti-cyberbullying educational programs in reducing cyber aggression and cyber victimisation
C. Li et al., 2022	Journal article	Asia-Pacific, Australia, Europe, Middle East, North America	42	266,888	Children and youth (aged 8-10 years)	3 databases (PsycInfo, PubMed, Web of Science)	2010–2021	2010–2021	Explore the prevalence trends of traditional bullying and cyberbullying for the past decade
J. Li et al., 2024	Journal article	9 countries (Asia-Pacific, Europe, North America,	57	98,351	Children, adolescents, and young adults (age	5 databases (Chinese CNKI Database, EBSCO-ASP, PubMed, Web of Science, Wiley Online Library), 9 journals (Computers in	Up to May, 2023	2016–2023	Examine the extent to which childhood maltreatment is correlated with cyberbullying (perpetration and victimisation), and whether these associations

		South/Latin America)			range not included)	Human Behavior; Cyberpsychology, Behavior, and Social Networking; Child Abuse and Neglect; Journal of Interpersonal Violence; Journal of Child and Adolescent Trauma; Journal of Child Sexual Abuse; Child Maltreatment; Journal of Family Violence; and Chinese Mental Health Journal), reference list of included studies, reviews, meta-analyses			varied by sample, publication, and research design characteristics
López-Barranco et al., 2022	Journal article	4 countries (Europe, North America, South/Latin America)	12	47,104	Adolescents (aged 13-18 years) and young adults (aged 19-24 years)	5 databases (Gender Studies, PsycInfo, PubMed, Scopus, Web of Science), reference list of shortlisted articles, contacting authors of articles of interest	January, 2015–January, 2021	2015–2020	Analyse the different types of violence perpetrated and experienced in dating relationship as a function of gender in adolescents and young adults
Lozano-Blaso, Barreiro-Collazo, et al., 2023	Journal article	5 countries (Asia-Pacific, Australia, Europe)	9	19,093	Adolescents (aged 11.5-18 years)	3 databases (PsycInfo, Science Direct, WOS)	2015–2020	2017–2020	Analyse the influence of family communication on cyber-victims and the moderating role of different sociodemographic variables (age, gender, nationality, and culture), as well

as social, emotional, and personality variables

Lozano-Blasco, Quilez-Robres, et al., 2023	Journal article	11 countries (Asia-Pacific, Australia, Europe, North America, South/Latin America)	32	238,977	Adolescents (aged 10-19 years; mean age 13.68 years)	4 databases (Scopus, PsycInfo, Science Direct, PubMed)	2013–2019	2013–2019	Investigate the significance of sex and age differences in cyber-victimisation
	Journal article	14 countries (Africa, Asia-Pacific, Europe, Middle East, North America, South/Latin America)	22	47,836	Adolescents (mean age range 11.72-16.5 years; mean age 14.64 years)	3 databases (PsycInfo, Science Direct, Scopus)	2014–2019	2014–2019	Examine whether it was possible for someone to be both a cybervictim and a cyberbully

Marciano et al., 2020	Journal article	13 countries (Asia-Pacific, Australia, Europe, North America)	56	40,682	Children and adolescents (mean age 10.5-16.7 years; mean age 13.4 years)	13 databases (CENTRAL, CINAHL, Communication and Mass Media Complete, EMBASE, ERIC, Google Scholar, MEDLINE, ProQuest Dissertations and Theses, ProQuest Sociology, PsycArticles, PsycInfo, Psychology and Behavioral Sciences Collection, Web of Science)	Up to 2018	2007–2017	Quantitatively summarise exclusively longitudinal studies on the causes and consequences of cyberbullying perpetration and cybervictimisation
Mills et al., 2024	Journal article	4 countries (Asia-Pacific, Europe, North America)	10	421	Children, adolescents, and young adults (aged 8-25 years)	4 databases (ProQuest, PubMed, Scopus, Web of Science)	2002–2022	2010–2021	Investigate the impacts of cyberbullying-like behaviours on psychophysiology, using EEG as the measurement method via a meta-analysis
Modecki et al., 2014	Journal article	NA	80	335,519	Adolescents (aged 12-18 years)	6 (Educational Resources Information Centre, Google Scholar, Proquest Dissertations and Theses, PsycInfo, PubMed, Scopus) and reference lists of eligible articles	No limit	2004–2013	Conduct a meta-analysis on the prevalence of bullying across cyber and traditional contexts among adolescents

Molero et al., 2022	Journal article	6 countries (Asia-Pacific, Europe, North America)	13	7,348	Adolescents and young adults (aged 8-37 years)	3 databases (PsycInfo, Scopus, Web of Science)	2011–2021	2014–2020	Analyse the relationship between cybervictimisation, anxiety, and depression in an adolescent population through a meta-analysis
Nesi et al., 2021	Journal article	14 countries (Asia-Pacific, North America, South/Latin America)	61	532 to 135,424	Adolescents and adults (aged 11-35.1 years)	3 databases (CINAHL, MEDLINE, PsycInfo)	Up to August, 2020	2010–2020	Provide an overview of the current research and to examine associations between different aspects of social media use and self-injurious thoughts and behaviours
Ng et al., 2022	Journal article	10 countries (Africa, Asia-Pacific, Australia, Europe, North America, South/Latin America)	15	35,694	Adolescents (aged 10-18 years)	6 databases (Cumulative Index to Nursing and Allied Health Literature, EMBASE, Google Scholar, ProQuest Dissertations and Theses, PsycInfo, PubMed) and reference list of review articles and eligible articles	Up to 2019	2000–2018	Examine the effectiveness of anti-bullying educational interventions at reducing the frequencies of traditional bullying or cyberbullying and cybervictimisation among adolescents
Oblad, 2012	Dissertation/ Thesis	27 countries (Asia-Pacific, Australia, Europe, North America)	24	42,118	NA	4 (PsycInfo, Interlibrary Loan, Internet accessible databases (e.g., Google Scholar), direct referrals by authors in previous reviews)	2000–2012	2004–2011	Compare gender differences in cyberbullying and victimisation

Polanin et al., 2022	Journal article	USA and non-USA	50	45,371	School students (mean age 13 years)	15 databases (Academic Search Complete, CrimeDoc, Education Full Text, ERIC, Grey Literature Database (Canadian), National Criminal Justice Reference Service Abstracts, ProQuest Criminal Justice, ProQuest Dissertations and Theses, ProQuest Education Journals, ProQuest Social Science Journals, PsycInfo, PubMed (MEDLINE), Social Care Online (UK), Social Sciences Abstracts, Social Science Research Network eLibrary), 5 journals (Aggressive Behavior, Child Development, Computers in Human Behavior, Journal of Interpersonal Violence, Computers in Human Behavior, Prevention Science), reference list of eligible articles and records that cited eligible articles	1995–2005	2004–2019	Conduct a systematic review and meta-analysis that synthesised the effects of school-based programs on cyberbullying perpetration or victimisation outcomes
-----------------	-----------------	----	--------	-------------------------------------	---	-----------	-----------	---

Pratt et al., 2014	Journal article	30 countries (Asia-Pacific, Australia, Europe, North America)	66	102,716	Minors (aged below 18 years) and adults (age range not included)	2 databases (Google Scholar, National Criminal Justice Reference Service)	Up to November, 2012	1995–2014	Conduct a meta-analysis on self-control and victimisation
Resett & Mesurado, 2021	Book chapter	Europe, North America, South/Latin America	8	7,627	Adolescents (aged 10-19 years)	8 databases (Dialnet, EBSCO Host, JSTOR, Latindex, SciELO, ScienceDirect, NCBI, PsycInfo)	2000–July, 2018	2000–2018	Analyse the effectiveness of bullying and cyberbullying interventions in adolescents aged 10–19 years, published between 2000 and July 2018 inclusive in English, Spanish, and Portuguese
Sarier, 2022	Journal article	Turkey (Europe)	37	21,768	Secondary and high school students (age range not included)	2 databases (DergiPark, YÖK thesis database)	Up to January, 2020	2010–2019	Combine the results of studies that reveal the relationship between cyberbullying and victimisation with different demographic variables in Turkey using the meta-analysis method
Sun & Fan, 2018	Journal article	Asia-Pacific, Europe, North America	40	71,722	NA	8 (Academic Search Primer, Business Source Primer, Communication Source, ERIC, Google Scholar, PsycArticles, PsycInfo, PubMed)	Up to October, 2013	2006–2013	Examine what is the general gender group difference in cybervictimisation as reported in the existing empirical studies

Tran et al., 2023	Journal article	7 countries (Asia-Pacific, Australia, Europe, North America)	17	79,202	Adolescents (aged 10-19 years)	2 databases (EMBASE, PubMed), reference sections of reviews and references that cited eligible studies	Up to 2021	2007–2020	Investigate the relationship between cybervictimisation and depression in adolescents
Van Cleemput et al., 2014	Conference piece	8 countries (Asia-Pacific, Australia, Europe, North America)	8	11,921	Adolescents (aged 10-18 years)	11 databases (Arts & Humanities Index, Communication Abstracts, Conference Proceedings Index - Social Science & Humanities, ERIC, Google Scholar, MEDLINE, PsycInfo, Social Sciences Index, Social Services Abstracts, Sociological Abstracts, Web of Science), contacting researchers through personal connections and research networks	January, 2003–September, 2014	2006–2013	Examine the relationship between peer victimisation, and suicidal ideation or suicide attempts in children and adolescents
Van Geel et al., 2014	Journal article	NA	34	284,375	Children and adolescents (aged 9-21 years)	3 (OvidMEDLINE, PsycInfo, Web of Science) and reference sections of review articles	January, 1910–January, 2013	1999–2012	Examine the relationship between peer victimisation, and suicidal ideation or suicide attempts in children and adolescents

Walters, 2021	Journal article	10 countries (Asia-Pacific, Australia, Europe, North America)	22	1,048	Samples aged below 18 years (mean age 9.5-15 years)	12 databases (Academic Search Complete, Criminal Justice Abstracts, Dissertation Abstracts, ERIC, HeinOnline, JSTOR Journals, PsycArticles, Psychology and Behavior Sciences Collection, PsycInfo, Social Sciences Citation Index, SocIndex, Sociological Collection)	Up to April, 2019	2008–2018	Gauge the magnitude of relationship between concurrent victimisation and perpetration, assess the temporal direction of the association between victimisation and perpetration, determine whether meaningful effect size differences exist between traditional bullying/victimisation and cyberbullying/victimisation and whether the effect extends across types and investigate the effect of four moderator variables
Wang & Jiang, 2023	Journal article	8 countries (Australia, Europe, Middle East)	11	29,859	Adolescents (aged 10-19 years)	7 databases (EBSCO, ERIC, ProQuest Dissertations and Theses, PsycInfo, PubMed, Scopus, Web of Science)	Up to May, 2021	2012–2021	Investigate the effectiveness of parent-related programs in reducing the frequency of cyberbullying perpetration and victimisation among adolescents
Wirth, 2018	Dissertation/ Thesis	8 countries (Australia, Europe, North America)	12	32,004	Children (aged 10-19 years)	7 databases (CINAHL, EMBASE, ERIC, Informit, PsycInfo, Pubmed, CINAHL, Scopus)	Up to 2018	2010–2016	Produce a systematic review and meta-analysis of the existing literature to discover whether cyberbullying intervention programs are effective at reducing perpetration and victimisation

Wissink et al., 2023	Journal article	11 countries (Africa, Asia-Pacific, Australia, Europe, North America)	48	11,968,667	Juveniles (mean age 12-23 years)	4 databases (ERIC, Google Scholar, PsycInfo, Web of Science), reference list of included studies	Up to May, 2019	2006–2019	Identify risk factors for cyberstalking, hacking, and sexting perpetrated by juveniles
Wong, 2021	Dissertation/ Thesis	27 countries (Africa, Asia-Pacific, Australia, Europe, Middle East, North America, South/Latin America)	52	59,294	Adolescents and emerging adults (aged below 30 years; mean age 12.08-24.14 years)	5 databases (ERIC, Google Scholar, PsycInfo, Web of Science), conference websites	2017–2020	2017–2020	Conduct a meta-analysis to study the moderation of age in cyberbullying with a focus on the interaction across different aspects of information and communication technology affordances on social cognitive development, dominant developmental environments, and developmental goals across adolescence and emerging adulthood
Yuchang et al., 2019	Journal article	Asia-Pacific, Europe, North America	56	214,819	Children and adolescents (aged 10-19 years)	4 databases (China National Knowledge Infrastructure, Google Scholar, PsycArticles, PsycInfo), reference section of review articles	NA	1998–2016	Examine cross-cultural perspectives to explore whether there are any differences between the effects of cyber victimisation, and traditional victimisation on the presence of depression and

Zhang & Chen, 2023	Journal article	11 countries (Asia-Pacific, Australia, Europe, North America)	24	27,438	School or university students (mean age 9.12-10.45 years)	5 databases (Chinese CNKI Database, Google Scholar, ProQuest Dissertations and Theses, PubMed, Web of Science)	Up to March, 2022	2010–2021	Conduct a meta-analysis to evaluate the exact association between emotional intelligence and school bullying victimisation
Zych et al., 2019	Journal article	15 countries (Asia-Pacific, Europe, North America)	25	25,268	Children and adolescents (aged up to 18 years; age range 11.57-18)	5 databases (Google Scholar, PsycInfo, PubMed, Scopus, Web of Science)	Up to November, 2016	2009–2017	Examine how empathy is related to different cyberbullying roles
Zych et al., 2021	Journal article	4 countries (Africa, Europe, North America)	23	55,445	Children or adolescents (aged up to 21 years; age range not included)	4 databases (Google Scholar, MEDLINE, Scopus, Web of Science)	Up to October, 2016	2000–2016	Assess whether involvement in bullying perpetration or victimisation could be risk factors for perpetration or victimisation in early romantic relationships

Note. NA is used in cases where the record did not provide the relevant information.

18. *The finding of age needs contextualization by including the developmental stage of samples that the three records analyzed—no adult studies were included in this finding, and it seems unlikely that this finding would apply to adults. You discuss this later, but it's important to set the discussion up by including a brief description of the studies. Also, it would seem to be important to report the proportion of studies that found each outcome. For example, ¾ found age differences—all of these had adolescent and young adult samples.*

Response:

We thank the reviewer for this constructive comment. We agree that it is crucial to provide a clear understanding of the developmental stages of the samples included in the analyses to accurately interpret these findings. Following the suggestion, we have revised the manuscript to include more detailed descriptions of the studies that contributed to our findings on age (pp. 10-11). Specifically, we have clarified that six out of the seven meta-analyses focusing on age, all of which exclusively involved children, adolescents, or college-aged samples, indicated that younger age was associated with higher risks of becoming a cyberbullying victim (p. 11).

Additionally, we have included information about the proportion of studies or effect sizes that found each specific outcome to provide a clearer statistical context (pp. 10-14).

Discussion

19. I recommend starting with fifth sentence about the current study (line 528)—the first two sentences over generalize and the next two justify the study, which was done in the introduction. It's better to get right to the point, especially in the discussion. This first paragraph should cover what you did in this study and what you found.

Response:

We thank the reviewer for the helpful recommendation. We agree that starting the discussion with a clear and concise overview of the results of the current review enhances clarity.

Following the recommendation, we have revised the opening of the discussion to eliminate generalizations and justifications which had been previously addressed in the introduction (p. 14). The discussion now starts directly with a focused summary of our study and its findings, as suggested: "The current review provides a systematic review of meta-analyses on the predictors and consequences associated with cyberbullying victimisation. 56 meta-analyses, with a total of 296 effect sizes, were featured within the current review. The umbrella review approach adopted in the current review made it possible to consider a broad scope of factors investigated by scholars and consider whether consensus in the field has been met on the factors that cause cyberbullying victimisation and its consequences (Belbasis et al., 2022; Koh et al., 2022). Our findings begin with a detailed analysis of the sociodemographic predictors, revealing nuanced differences in vulnerability among various groups. The subsequent sections delve into the psychological and contextual factors, each highlighted by distinct patterns and relationships that emerge from the meta-analytical data. The central findings derived from the analysis provide a holistic view of the potential predictors and consequences of cyberbullying

victimisation, and serve as a basis for future research as well as intervention programmes. In the following, the ten central findings of the current review are discussed."

20. The discussion is very long, and it rehashes out well-known findings (e.g., gender, prior cyberbullying victimization). I suggest focusing on the main 3-4 findings that are novel and make sure to be consistent in commenting on the size of effect sizes for each outcome you discuss. Then, you could close with paragraphs that 1. sums up additional demographic/victimization findings in the review and 2. implications of findings.

Response:

We thank the reviewer for the constructive feedback regarding the length and focus of the discussion section of our manuscript. We greatly appreciate the Reviewer's suggestion to emphasise the main novel findings and streamline the presentation for enhanced clarity and impact. Upon careful consideration of the comment, we have made substantial revisions to the discussion to enhance its readability and focus (pp. 15-21). While we have condensed the discussion around well-known findings such as gender impacts and prior victimisation and gone into more detail on the more novel conclusions drawn from the results, overall we have chosen to retain a detailed overview of all pertinent findings. This approach was taken to ensure that the review remains a comprehensive overview in the field of cyberbullying victimisation, which we believe is crucial for providing a foundational understanding for both new and seasoned researchers.

In order to address the Reviewer's comment about the length, we have significantly reduced the word count for each point discussed, ensuring that each is presented succinctly while still highlighting the most novel aspects of our research. This allows us to devote more attention to the more novel findings, enhancing the contribution of our review to the field. Moreover, we have ensured consistent commentary on the size of effect sizes for each outcome discussed, aiming to provide a clear and precise understanding of the implications of our findings. Although we did not remove the original findings, we believe that the revisions made in response to the Reviewer's comments improve the discussion's focus and readability, ensuring that critical and novel findings are highlighted effectively. The discussion thus serves not only to advance the field by introducing new insights but also to affirm and contextualize well-established knowledge within the broader landscape of cyberbullying research.

21. Right now, the review feels like it fails to address the “so what” question at the end—what good is this information? It seems strange to include research gaps in the same section as limitations—this should be separated out.

Response:

We thank the reviewer for the very important comment and suggestion. We acknowledge the importance of clearly addressing the relevance and applicability of our findings in the field of cyberbullying research. In response to the first comment regarding implications, we have revised the manuscript to include a more pronounced emphasis on the practical implications and significance of our findings (p. 21). We have elaborated on how the ten key findings from our systematic review can inform policy, educational interventions, and future research directions. Specifically, we now clearly outline how the identification

of robust predictors and consequences of cyberbullying can aid stakeholders in developing targeted interventions that are grounded in empirical evidence. This is crucial for formulating effective anti-cyberbullying strategies and policies that can be implemented in educational and social settings to mitigate the impact of cyberbullying.

Regarding the separation of research gaps and limitations, we appreciate your recommendation for clarity and have adjusted the structure of our manuscript accordingly. We now present the research gaps and future directions in a distinct section following the discussion of study limitations (pp. 19-20).

22. Minor—editing for grammar (e.g., comma use) is needed throughout the manuscript.

Response:

Reviewer #2

The present meta-analysis carried out a comprehensive review and synthesis based on 34 published meta-analyses and 153 effect sizes from work carried out across the globe. Key findings include that females who use the Internet frequently and have had a traditional bullying experience are at greater risk for cyberbullying. In addition, the authors also identified unregulated school environments, uninvolved family backgrounds and unsupportive parental relationships as risk factors for cyberbullying victimisation. Furthermore, they also identified a number of deleterious consequences following cyberbullying victimisation including anxiety, Depression, loneliness, low levels of school performance, and higher levels of maladaptive coping behaviors. Abstract seems appropriate.

1. *The introduction and study rationale are well developed and reused key pieces of scholarship. It also includes prevalence estimates about digital technology use as well as rates of cyberbullying. Next the literature review discusses both predictors of as well as consequences of cyber bullying victimization. This includes mention of some theoretical work including general strain theory and online disinhibition effect theory. The principal rationale is simply to efficiently synthesize the wealth of information that has been published on the general topic of correlates and consequences of cyberbullying victimization which is what the present study sets out to do. Importantly the authors differentiate the work that focuses on contextual variations of how cyberbullying victimization is conceptualized or defined. This includes peer cyber bullying as well as cyberbullying victimization related to dating and intimate partner violence. Thus, in essence, the present synthesis combines these separate areas of inquiry.*

Response:

We thank the Reviewer for the positive feedback and all the constructive comments, which we have incorporated in our revision to enhance the quality of our manuscript.

2. *The search strategy as well as selection criteria were clearly explained, in a sufficient amount of detail. The search parameters included five main databases as well as seven journals that the authors identified. The authors describe that these journals were selected based on search strategies of previous meta-analyses, citing Kowalski and colleagues (2014). Whether or not the*

seven journals are still the most relevant today for scholarship on cyber bullying is debatable, but they certainly cover a large proportion of scholarship published on the topic.

Response:

We thank the reviewer for the constructive suggestion, and have revised our search strategy to include a wider range of journals that are more relevant to today's research on cyberbullying. In addition to the journals we had already included (*Aggressive Behavior; Computers in Human Behavior; Cyberpsychology, Behavior, and Social Networking; Journal of Adolescence; Journal of School Violence; New Media and Society; School Psychology Review*), we have added six new journals that have recently published meta-analyses on cyberbullying-related topics (*Adolescent Research Review; Aggression and Violent Behavior; Children and Youth Services Review; Deviant Behavior; Journal of Pediatric Nursing; Trauma, Violence, & Abuse*) (p. 23).

The updated search resulted in the retrieval of 373 new records out of which 63 records were identified as duplicates and removed, resulting in 310 records to be screened for potential inclusion. Title and abstract screening was conducted by the first author and a research assistant (92.76% inter-rater agreement), and resulted in the removal of a further 219 records. For remaining 91 records, full-text screening was conducted by the first author and a research assistant (95.23% inter-rater agreement). Another 69 records were eliminated during full-text screening, leaving a total of 22 new records (overall 56 records) to be included in the final review. The PRISMA flow-chart with the updated numbers (p. 27) incorporating both the original search as well as the updated search is included below (Figure 1).

Following the search, quality assessment was conducted for the newly included records using the Joanna Briggs Institute critical appraisal instrument for Systematic Reviews and Research Syntheses (JBI, 2017). The quality of each newly included meta-analysis was assessed independently by the first author and a trained research assistant (inter-rater agreement 94%; range 90–100%). Finally, data extraction for the new records was conducted by the first author and a trained research assistant (inter-rater agreement range=81.54%–100%). We have further updated our results (pp. 10-14) and discussion (pp. 14-19) to reflect the changes from the updated search.

Figure 1

PRISMA Flowchart

3. *It's important to note that the selection criteria included meta-analyses if they focused on at least one predictor or at least one consequence of cyberbullying victimization. The quality of each included meta-analysis was assessed independently by two of the authors using a critical appraisal instrument for systematic reviews and research syntheses. The authors made a decision to only include work that was published in English, Chinese, Malay, or Bahasa Indonesia. It would be interesting and important to note based on this work how many additional meta-analyses existed in other target languages other than the ones included in the present study.*

Response:

We thank the reviewer for bringing up this important point. We have we have clarified the process of record exclusion in our manuscript (p. 27). Specifically, 11 records were excluded during abstract screening based on the language criteria, and 4 records were excluded during full-text screening based on the same criteria. Additionally, we conducted English translations of the abstracts of these records using Google Translate and confirmed that none of these excluded records were meta-analyses. Moreover, several reviews have found that including non-English language studies does not significantly alter review outcomes as compared to when no non-English language works are included (Moher et al., 2000; Morrison et al., 2012). Given this, coupled with the fact that the samples of the meta-analyses included

within our review were drawn from all global regions and therefore had significant cultural diversity, it is less likely that the exclusion of non-English studies a limitation of our review.

4. *The design and synthesis plan of the current review was not preregistered as disclosed by the authors. It is not entirely clear why the authors opted to not preregister the current investigation. The authors completed a PRISMA 2020 checklist and reporting summary. Some PRISMA Checklist items are blank, related to risk of bias assessment, reporting biases, certainty of evidence, heterogeneity of results etc., a few are not applicable as noted.*

Response:

We thank the reviewer for this important comment. With regards to the pre-registration, it was not completed for this investigation. Preregistration is indeed a best practice that enhances transparency and reduces the risk of bias by pre-defining research questions and analysis strategies. We acknowledge this oversight and understand that preregistration could have added an additional layer of rigor to our systematic review. We have acknowledged in our Transparency and Openness section that the current review was not pre-registered (p. 21). We appreciate the reviewers pointing out this omission, and we will consider preregistration for future systematic reviews to better align with contemporary research standards.

We have also endeavoured to adhere closely to the PRISMA guidelines in the current revision. We have completed all items in the PRISMA checklist and have attached it with the manuscript. Furthermore, we recognise the necessity of clearly explaining the reasons for non-applicability of certain items due to

the specific nature of our systematic review, which did not involve a new quantitative analysis but rather a synthesis of existing meta-analyses.

Regarding the items marked as 'NA' related to sensitivity analyses and assessment of the certainty of evidence (item #14, item #15, item #20d, item #21, item #22, item #24c, item #25, item #26), these elements typically pertain to original quantitative meta-analyses where primary data is pooled and analysed afresh. Our review synthesized findings from existing meta-analyses; hence, direct application of these items was not relevant.

Furthermore, concerning the heterogeneity analysis, we attempted to conduct subgroup analyses to explore possible causes of heterogeneity among the study results. Unfortunately, these were not feasible due to an insufficient number of meta-analyses analysing identical subgroups for the same outcomes. This limitation is noted in the Method section (p. 29, Footnote 1). We have also acknowledged this as a limitation in our Discussion (p. 19), and specify that we were unable to study the finer details included in either the meta-analyses or the original sources that studies included. The current review specifically focused only on meta-analyses, and aimed to provide a broad overview of cyberbullying victimisation research. While the umbrella review approach utilised in the current work does not facilitate the studying of more detailed aspects of various studies (e.g., different moderators, types of measures utilised, response time frames, etc.), it is important to consider these nuances by directing attention to the individual meta-analyses contained in the current review, as well as the various studies cited within them.

- 5. The authors spend quite a bit of time evaluating the quality of the different meta-analyses included, assessing their methodological adequacy which are described, analyzed, and presented in great detail in Tables 2 and 3. This is a great strength of the effort. The final focus included 20*

studies of the 34 that examined the link between cyberbullying victimization and its correlates or predictors. Seven of these focused on sociodemographic predictors, five on psychological predictors, and 17 on contextual predictors. Fourteen of 31 (it is not clear why the authors indicate 31 when the total final number was 34 here) included studies focused on consequences of cyberbullying victimization. The authors differentiated between psychological consequences as well as behavioral consequences.

Response:

We thank the review for the comment, and appreciate your positive feedback on our work. We are also grateful for the reviewer for pointing out the error in our manuscript regarding the number of total records included, which we have now corrected to accurately represent the final number of records (p. 13).

- 6. For each of these sub areas, the authors estimated (Mdn?) median associations and provided ranges and presented the findings of the effect sizes in forest plots, broken down by each source study meta- analysis. The presentation of the findings in this manner is both intuitive, thorough, and descriptive; they permit a direct comparison of the observed effects across different meta-analyses. These plots also include effect size ranges.*

Response:

We thank the review for the positive evaluation of our methodology.

7. *The discussion is well organized and highlights 10 central findings of the investigation. These will not be repeated here. The conclusions reached based on this work seem valid and robust, given that they are based on a large number of previously published meta-analyses.*

Response:

We thank the review for the encouraging feedback on our discussion.

8. *In conclusion, this is a very interesting and important study that capitalizes on a tremendous amount of published information on both the correlates as well as consequences of cyberbullying victimization. It's a tremendously comprehensive and large effort that seems to have been carried out in a very efficient and planful manner; it provides highly consequential and important findings about the correlates and consequences of cyber bullying victimization. No additional analyses are suggested or recommended. In its present form, the present synthesis of meta-analyses advances our knowledge and understanding about the correlates and consequences of cyberbullying victimization given the sheer scope and approach used. Thus, the work very meaningfully contributes to the extant literature on cyberbullying victimization.*

Response:

We thank the review for the comment, and appreciate your positive feedback on our work.

References

- Abregú-Crespo, R., Garriz-Luis, A., Ayora, M., Martín-Martínez, N., Cavone, V., Carrasco, M. Á., Fraguas, D., Martín-Babarro, J., Arango, C., & Díaz-Caneja, C. M. (2024). School bullying in children and adolescents with neurodevelopmental and psychiatric conditions: A systematic review and meta-analysis. *The Lancet Child & Adolescent Health*, 8(2), 122–134. [https://doi.org/10.1016/S2352-4642\(23\)00289-4](https://doi.org/10.1016/S2352-4642(23)00289-4)
- Agatston, P. W., Kowalski, R., & Limber, S. (2007). Students' perspectives on cyber bullying. *Journal of Adolescent Health*, 41(6), S59–S60. <https://doi.org/10.1016/j.jadohealth.2007.09.003>
- Agustina, J. R. (2015). *Understanding cyberVictimization: Digital architectures andThe disinhibition effect*. <https://doi.org/10.5281/ZENODO.22239>
- Aizenkot, D. (2022). The predictability of routine activity theory for cyberbullying victimization among children and youth: Risk and protective factors. *Journal of Interpersonal Violence*, 37(13–14), NP11857–NP11882. <https://doi.org/10.1177/0886260521997433>
- Akgül, G. (2023). Routine Activities Theory in cyber victimization and cyberbullying experiences of Turkish adolescents. *International Journal of School & Educational Psychology*, 11(2), 135–144. <https://doi.org/10.1080/21683603.2021.1980475>
- Alipan, A., Skues, J. L., Theiler, S., & Wise, L. (2020). Defining cyberbullying: A multifaceted definition based on the perspectives of emerging adults. *International Journal of Bullying Prevention*, 2(2), 79–92. <https://doi.org/10.1007/s42380-019-00018-6>

- Alipan, A., Skues, J., Theiler, S., & Wise, L. (2015). Defining cyberbullying: A multiple perspectives approach. *Studies in Health Technology and Informatics*, 219, 9-.
<https://doi.org/10.3233/978-1-61499-595-1-9>
- Almeida, A., Correia, I., Marinho, S., & Garcia, D. (2012). Virtual but not less real: A study of cyberbullying and its relations to moral disengagement and empathy. In Q. Li, D. Cross, & P. K. Smith (Eds.), *Cyberbullying in the global playground: Research from international perspectives*. Malden, MA: Blackwell.
- Aoyama, I., Saxon, T. F., & Fearon, D. D. (2011). Internalizing problems among cyberbullying victims and moderator effects of friendship quality. *Multicultural Education & Technology Journal*, 5(2), 92–105. <https://doi.org/10.1108/17504971111142637>
- Arató, N., Zsidó, A. N., Rivnyák, A., Péley, B., & Lábadi, B. (2022). Risk and protective factors in cyberbullying: The role of family, social support and emotion regulation. *International Journal of Bullying Prevention*, 4(2), 160–173. <https://doi.org/10.1007/s42380-021-00097-4>
- Bailey, E. R., Matz, S. C., Youyou, W., & Iyengar, S. S. (2020). Authentic self-expression on social media is associated with greater subjective well-being. *Nature Communications*, 11(1), 4889. <https://doi.org/10.1038/s41467-020-18539-w>
- Balakrishnan, V. (2015). Cyberbullying among young adults in Malaysia: The roles of gender, age and Internet frequency. *Computers in Human Behavior*, 46, 149–157.
<https://doi.org/10.1016/j.chb.2015.01.021>

- Barlett, C. P. (2015). Anonymously hurting others online: The effect of anonymity on cyberbullying frequency. *Psychology of Popular Media Culture*, 4(2), 70–79. <https://doi.org/10.1037/a0034335>
- Barlett, C. P., & Chamberlin, K. (2017). Examining cyberbullying across the lifespan. *Computers in Human Behavior*, 71, 444–449. <https://doi.org/10.1016/j.chb.2017.02.009>
- Barlett, C. P., Gentile, D. A., & Chew, C. (2016). Predicting cyberbullying from anonymity. *Psychology of Popular Media Culture*, 5(2), 171–180. <https://doi.org/10.1037/ppm0000055>
- Barlett, C. P., Kowalski, R. M., & Wilson, A. M. (2024). Meta-analyses of the predictors and outcomes of cyberbullying perpetration and victimization while controlling for traditional bullying perpetration and victimization. *Aggression and Violent Behavior*, 74, 101886. <https://doi.org/10.1016/j.avb.2023.101886>
- Belbasis, L., Bellou, V., & Ioannidis, J. P. A. (2022). Conducting umbrella reviews. *BMJ Medicine*, 1(1), e000071. <https://doi.org/10.1136/bmjmed-2021-000071>
- Berne, S., Frisé, A., Schultze-Krumbholz, A., Scheithauer, H., Naruskov, K., Luik, P., Katzer, C., Erentaite, R., & Zukauskienė, R. (2013). Cyberbullying assessment instruments: A systematic review. *Aggression and Violent Behavior*, 18(2), 320–334. <https://doi.org/10.1016/j.avb.2012.11.022>
- Best, P., Manktelow, R., & Taylor, B. (2014). Online communication, social media and adolescent wellbeing: A systematic narrative review. *Children and Youth Services Review*, 41, 27–36. <https://doi.org/10.1016/j.childyouth.2014.03.001>

- Brown, G., & Greenfield, P. M. (2021). Staying connected during stay-at-home: Communication with family and friends and its association with well-being. *Human Behavior and Emerging Technologies*, 3(1), 147–156. <https://doi.org/10.1002/hbe2.246>
- Caridade, S. M. M., & Braga, T. (2020). Youth cyber dating abuse: A meta-analysis of risk and protective factors. *Cyberpsychology: Journal of Psychosocial Research on Cyberspace*, 14(3). <https://doi.org/10.5817/CP2020-3-2>
- Chen, L., Ho, S. S., & Lwin, M. O. (2017). A meta-analysis of factors predicting cyberbullying perpetration and victimization: From the social cognitive and media effects approach. *New Media & Society*, 19(8), 1194–1213. <https://doi.org/10.1177/1461444816634037>
- Chen, Q., Chan, K. L., Guo, S., Chen, M., Lo, C. K., & Ip, P. (2023). Effectiveness of digital health interventions in reducing bullying and cyberbullying: A meta-analysis. *Trauma, Violence, & Abuse*, 24(3), 1986–2002. <https://doi.org/10.1177/15248380221082090>
- Chen, X., Wang, L., & Wang, Y. (2024). Experiences of bullying and victimization and adolescents' life satisfaction: A meta-analysis. *Aggression and Violent Behavior*, 76, 101930. <https://doi.org/10.1016/j.avb.2024.101930>
- Christina, S., Magson, N. R., Kakar, V., & Rapee, R. M. (2021). The bidirectional relationships between peer victimization and internalizing problems in school-aged children: An updated systematic review and meta-analysis. *Clinical Psychology Review*, 85, 101979. <https://doi.org/10.1016/j.cpr.2021.101979>

- Chun, J., Lee, J., Kim, J., & Lee, S. (2020). An international systematic review of cyberbullying measurements. *Computers in Human Behavior, 113*, 106485.
<https://doi.org/10.1016/j.chb.2020.106485>
- Cohen-Almagor, R. (2018). Social responsibility on the internet: Addressing the challenge of cyberbullying. *Aggression and Violent Behavior, 39*, 42–52.
<https://doi.org/10.1016/j.avb.2018.01.001>
- Dempsey, A. G., Sulkowski, M. L., Nichols, R., & Storch, E. A. (2009). Differences between peer victimization in cyber and physical settings and associated psychosocial adjustment in early adolescence. *Psychology in the Schools, 46*(10), 962–972.
<https://doi.org/10.1002/pits.20437>
- Doty, J. L., Girón, K., Mehari, K. R., Sharma, D., Smith, S. J., Su, Y.-W., Ma, X., Rijo, D., & Rousso, B. (2022). The dosage, context, and modality of interventions to prevent cyberbullying perpetration and victimization: A systematic review. *Prevention Science, 23*(4), 523–537. <https://doi.org/10.1007/s11121-021-01314-8>
- Eberle, V. (2023). *Sextortion risk and protective factors: A meta-analysis* [Doctoral Dissertation, Kansas State University]. K-State Electronic Theses, Dissertations, and Reports: 2004 -.
<https://hdl.handle.net/2097/43478>
- Elgar, F. J., Napoletano, A., Saul, G., Dirks, M. A., Craig, W., Poteat, V. P., Holt, M., & Koenig, B. W. (2014). Cyberbullying victimization and mental health in adolescents and the moderating role of family dinners. *JAMA Pediatrics, 168*(11), 1015.
<https://doi.org/10.1001/jamapediatrics.2014.1223>

- Englander, E., Donnerstein, E., Kowalski, R., Lin, C. A., & Parti, K. (2017). Defining cyberbullying. *Pediatrics*, *140*(Supplement_2), S148–S151.
<https://doi.org/10.1542/peds.2016-1758U>
- Erbıçer, E. S., Ceylan, V., Yalçın, M. H., Erbıçer, S., Akın, E., Koçtürk, N., & Doğan, T. (2023). Cyberbullying among children and youth in Türkiye: A systematic review and meta-analysis. *Journal of Pediatric Nursing*, *73*, 184–195.
<https://doi.org/10.1016/j.pedn.2023.09.003>
- Erdur-Baker, Ö. (2010). Cyberbullying and its correlation to traditional bullying, gender and frequent and risky usage of internet-mediated communication tools. *New Media & Society*, *12*(1), 109–125. <https://doi.org/10.1177/1461444809341260>
- Fisher, B. W., Gardella, J. H., & Teurbe-Tolon, A. R. (2016). Peer cybervictimization among adolescents and the associated internalizing and externalizing problems: A meta-analysis. *Journal of Youth and Adolescence*, *45*(9), 1727–1743. <https://doi.org/10.1007/s10964-016-0541-z>
- Foody, M., Samara, M., & O’Higgins Norman, J. (2017). Bullying and cyberbullying studies in the school-aged population on the island of Ireland: A meta-analysis. *British Journal of Educational Psychology*, *87*(4), 535–557. <https://doi.org/10.1111/bjep.12163>
- Gaffney, H., Farrington, D. P., Espelage, D. L., & Ttofi, M. M. (2019). Are cyberbullying intervention and prevention programs effective? A systematic and meta-analytical review. *Aggression and Violent Behavior*, *45*, 134–153. <https://doi.org/10.1016/j.avb.2018.07.002>

- García-Hermoso, A., Hormazabal-Aguayo, I., Oriol-Granado, X., Fernández-Vergara, O., & Del Pozo Cruz, B. (2020). Bullying victimization, physical inactivity and sedentary behavior among children and adolescents: A meta-analysis. *International Journal of Behavioral Nutrition and Physical Activity*, *17*(1), 114. <https://doi.org/10.1186/s12966-020-01016-4>
- Gardella, J. H., Fisher, B. W., & Teurbe-Tolon, A. R. (2017). A systematic review and meta-analysis of cyber-victimization and educational outcomes for adolescents. *Review of Educational Research*, *87*(2), 283–308. <https://doi.org/10.3102/0034654316689136>
- Gilbar, O., Charak, R., Trujillo, O., Cantu, J. I., Cavazos, V., & Lavi, I. (2023). Meta-analysis of cyber intimate partner violence perpetration and victimization: Different types and their associations with face-to-Face IPV among men and women. *Trauma, Violence, & Abuse*, *24*(3), 1948–1965. <https://doi.org/10.1177/15248380221082087>
- Gini, G., Card, N. A., & Pozzoli, T. (2018). A meta-analysis of the differential relations of traditional and cyber-victimization with internalizing problems. *Aggressive Behavior*, *44*(2), 185–198. <https://doi.org/10.1002/ab.21742>
- Guan, S.-S. A., & Subrahmanyam, K. (2009). Youth internet use: Risks and opportunities. *Current Opinion in Psychiatry*, *22*(4), 351–356. <https://doi.org/10.1097/YCO.0b013e32832bd7e0>
- Guo, S. (2016). A meta-analysis of the predictors of cyberbullying perpetration and victimization. *Psychology in the Schools*, *53*(4), 432–453. <https://doi.org/10.1002/pits.21914>

- Heerde, J. A., & Hemphill, S. A. (2019). Are bullying perpetration and victimization associated with adolescent deliberate self-harm? A meta-analysis. *Archives of Suicide Research*, 23(3), 353–381. <https://doi.org/10.1080/13811118.2018.1472690>
- Hinduja, S., & Patchin, J. W. (2008). Cyberbullying: An exploratory analysis of factors related to offending and victimization. *Deviant Behavior*, 29(2), 129–156. <https://doi.org/10.1080/01639620701457816>
- Holfeld, B., & Mishna, F. (2019). Internalizing symptoms and externalizing problems: Risk factors for or consequences of cyber victimization? *Journal of Youth and Adolescence*, 48(3), 567–580. <https://doi.org/10.1007/s10964-018-0974-7>
- Hu, T., Jin, F., & Deng, H. (2024). Association between gender nonconformity and victimization: A meta-analysis. *Current Psychology*, 43(1), 281–299. <https://doi.org/10.1007/s12144-023-04269-x>
- Hu, Y., Bai, Y., Pan, Y., & Li, S. (2021). Cyberbullying victimization and depression among adolescents: A meta-analysis. *Psychiatry Research*, 305, 114198. <https://doi.org/10.1016/j.psychres.2021.114198>
- Huang, N., Zhang, S., Mu, Y., Yu, Y., Riem, M. M. E., & Guo, J. (2024). Does the COVID-19 pandemic increase or decrease the global cyberbullying behaviors? A systematic review and meta-analysis. *Trauma, Violence, & Abuse*, 25(2), 1018–1035. <https://doi.org/10.1177/15248380231171185>

- Jang, H., Song, J., & Kim, R. (2014). Does the offline bully-victimization influence cyberbullying behavior among youths? Application of General Strain Theory. *Computers in Human Behavior, 31*, 85–93. <https://doi.org/10.1016/j.chb.2013.10.007>
- Joanna Briggs Institute. (2017). *Checklist for systematic reviews and research syntheses*. https://joannabriggs.org/ebp/critical_appraisal_tools
- John, A., Glendenning, A. C., Marchant, A., Montgomery, P., Stewart, A., Wood, S., Lloyd, K., & Hawton, K. (2018). Self-harm, suicidal behaviours, and cyberbullying in children and young people: Systematic review. *Journal of Medical Internet Research, 20*(4), e129. <https://doi.org/10.2196/jmir.9044>
- Kalia, D., & Aleem, S. (2017). Cyber victimization among adolescents: Examining the role of Routine Activity Theory. *Journal of Psychosocial Research, 12*(1), 223–232.
- Kamaruddin, I. K., Ma'rof, A. M., Mohd Nazan, A. I. N., & Ab Jalil, H. (2023). A systematic review and meta-analysis of interventions to decrease cyberbullying perpetration and victimization: An in-depth analysis within the Asia Pacific region. *Frontiers in Psychiatry, 14*, 1014258. <https://doi.org/10.3389/fpsy.2023.1014258>
- Killer, B., Bussey, K., Hawes, D. J., & Hunt, C. (2019). A meta-analysis of the relationship between moral disengagement and bullying roles in youth. *Aggressive Behavior, 45*(4), 450–462. <https://doi.org/10.1002/ab.21833>
- Koh, J., Tng, G. Y. Q., & Hartanto, A. (2022). Potential and pitfalls of mobile mental health apps in traditional treatment: An umbrella review. *Journal of Personalized Medicine, 12*(9), 1376. <https://doi.org/10.3390/jpm12091376>

- Kokkinos, C. M., Antoniadou, N., & Markos, A. (2014). Cyber-bullying: An investigation of the psychological profile of university student participants. *Journal of Applied Developmental Psychology, 35*(3), 204–214.
<https://doi.org/10.1016/j.appdev.2014.04.001>
- Kowalski, R. M., Giumetti, G. W., Schroeder, A. N., & Lattanner, M. R. (2014). Bullying in the digital age: A critical review and meta-analysis of cyberbullying research among youth. *Psychological Bulletin, 140*(4), 1073–1137. <https://doi.org/10.1037/a0035618>
- Kwan, G. C. E., & Skoric, M. M. (2013). Facebook bullying: An extension of battles in school. *Computers in Human Behavior, 29*(1), 16–25. <https://doi.org/10.1016/j.chb.2012.07.014>
- Lan, M., Law, N., & Pan, Q. (2022). Effectiveness of anti-cyberbullying educational programs: A socio-ecologically grounded systematic review and meta-analysis. *Computers in Human Behavior, 130*, 107200. <https://doi.org/10.1016/j.chb.2022.107200>
- Langos, C. (2012). Cyberbullying: The challenge to define. *Cyberpsychology, Behavior, and Social Networking, 15*(6), 285–289. <https://doi.org/10.1089/cyber.2011.0588>
- Li, C., Wang, P., Martin-Moratinos, M., Bella-Fernández, M., & Blasco-Fontecilla, H. (2022). Traditional bullying and cyberbullying in the digital age and its associated mental health problems in children and adolescents: A meta-analysis. *European Child & Adolescent Psychiatry. https://doi.org/10.1007/s00787-022-02128-x*
- Li, J., Huebner, E. S., & Tian, L. (2024). Linking childhood maltreatment to cyberbullying perpetration and victimization: A systematic review and multilevel meta-analysis. *Computers in Human Behavior, 156*, 108199. <https://doi.org/10.1016/j.chb.2024.108199>

- Livingstone, S., Stoilova, M., & Kelly, A. (2016). 14. Cyberbullying: Incidence, trends and consequences. In *Ending the torment: Tackling bullying from the schoolyard to cyberspace* (pp. 115–120). United Nations Office of the Special Representative of the Secretary-General on Violence against Children, New York, USA.
- López-Barranco, P. J., Jiménez-Ruiz, I., Pérez-Martínez, M. J., Ruiz-Penin, A., & Jiménez-Barbero, J. A. (2022). Systematic review and meta-analysis of the violence in dating relationships in adolescents and young adults. *Revista Iberoamericana de Psicología y Salud, 13*(2). <https://doi.org/10.23923/j.rips.2022.02.055>
- Lozano-Blasco, R., Barreiro-Collazo, A., Romero-Gonzalez, B., & Soto-Sanchez, A. (2023). The family context in cybervictimization: A systematic review and meta-analysis. *Trauma, Violence, & Abuse, 15*248380231207894. <https://doi.org/10.1177/15248380231207894>
- Lozano-Blasco, R., Cortés-Pascual, A., & Latorre-Martínez, M. P. (2020). Being a cybervictim and a cyberbully—The duality of cyberbullying: A meta-analysis. *Computers in Human Behavior, 111*, 106444–. <https://doi.org/10.1016/j.chb.2020.106444>
- Lozano-Blasco, R., Quilez-Robres, A., & Latorre-Coscolluela, C. (2023). Sex, age and cyber-victimization: A meta-analysis. *Computers in Human Behavior, 139*, 107491. <https://doi.org/10.1016/j.chb.2022.107491>
- Marciano, L., Schulz, P. J., & Camerini, A.-L. (2020). Cyberbullying perpetration and victimization in Youth: A meta-analysis of longitudinal Studies. *Journal of Computer-Mediated Communication, 25*(2), 163–181. <https://doi.org/10.1093/jcmc/zmz031>

- Menesini, E., Nocentini, A., & Calussi, P. (2011). The measurement of cyberbullying: dimensional structure and relative item severity and discrimination. *Cyberpsychology, Behavior, and Social Networking*, *14*(5), 267–274.
<https://doi.org/10.1089/cyber.2010.0002>
- Mills, L., Driver, C., McLoughlin, L. T., Anijärv, T. E., Mitchell, J., Lagopoulos, J., & Hermens, D. F. (2024). A systematic review and meta-analysis of electrophysiological studies of online social Exclusion: Evidence for the neurobiological impacts of cyberbullying. *Adolescent Research Review*, *9*(1), 135–163. <https://doi.org/10.1007/s40894-023-00212-0>
- Mishna, F., Cook, C., Saini, M., Wu, M.-J., & MacFadden, R. (2011). Interventions to prevent and reduce cyber abuse of youth: A systematic review. *Research on Social Work Practice*, *21*(1), 5–14. <https://doi.org/10.1177/1049731509351988>
- Modecki, K. L., Minchin, J., Harbaugh, A. G., Guerra, N. G., & Runions, K. C. (2014). Bullying prevalence across contexts: A meta-analysis measuring cyber and traditional bullying. *Journal of Adolescent Health*, *55*(5), 602–611.
<https://doi.org/10.1016/j.jadohealth.2014.06.007>
- Moher, D., Pham, Klassen, T. P., Schulz, K. F., Berlin, J. A., Jadad, A. R., & Liberati, A. (2000). What contributions do languages other than English make on the results of meta-analyses? *Journal of Clinical Epidemiology*, *53*(9), 964–972.
[https://doi.org/10.1016/S0895-4356\(00\)00188-8](https://doi.org/10.1016/S0895-4356(00)00188-8)
- Molero, M. M., Martos, Á., Barragán, A. B., Pérez-Fuentes, M. C., & Gázquez, J. J. (2022). Anxiety and depression from cybervictimization in adolescents: A meta-analysis and

- meta-regression Study. *The European Journal of Psychology Applied to Legal Context*, 14(1), 42–50. <https://doi.org/10.5093/ejpalc2022a5>
- Morrison, A., Polisena, J., Husereau, D., Moulton, K., Clark, M., Fiander, M., Mierzwinski-Urban, M., Clifford, T., Hutton, B., & Rabb, D. (2012). The effect of English-language restriction on systematic review-based meta-analyses: A systematic review of empirical studies. *International Journal of Technology Assessment in Health Care*, 28(2), 138–144. <https://doi.org/10.1017/S0266462312000086>
- Nesi, J., Burke, T. A., Bettis, A. H., Kudinova, A. Y., Thompson, E. C., MacPherson, H. A., Fox, K. A., Lawrence, H. R., Thomas, S. A., Wolff, J. C., Altemus, M. K., Soriano, S., & Liu, R. T. (2021). Social media use and self-injurious thoughts and behaviors: A systematic review and meta-analysis. *Clinical Psychology Review*, 87, 102038. <https://doi.org/10.1016/j.cpr.2021.102038>
- Ng, E. D., Chua, J. Y. X., & Shorey, S. (2022). The effectiveness of educational interventions on traditional bullying and cyberbullying among adolescents: A systematic review and meta-analysis. *Trauma, Violence, & Abuse*, 23(1), 132–151. <https://doi.org/10.1177/1524838020933867>
- Oblad, T. P. (2012). *Understanding cyberbullying in the net generation: A meta-analytic review* [Master Thesis, Texas Tech University Libraries]. Texas Tech University Libraries.
- O'Day, E. B., & Heimberg, R. G. (2021). Social media use, social anxiety, and loneliness: A systematic review. *Computers in Human Behavior Reports*, 3, 100070. <https://doi.org/10.1016/j.chbr.2021.100070>

- Olenik-Shemesh, D., Heiman, T., & Eden, S. (2012). Cyberbullying victimisation in adolescence: Relationships with loneliness and depressive mood. *Emotional and Behavioural Difficulties*, 17(3–4), 361–374.
<https://doi.org/10.1080/13632752.2012.704227>
- Olweus, D. (1995). *Bullying at School: What We Know and What We Can Do*. Blackwell Publishers/AIDC.
- O'Malley, R. L., & Holt, K. M. (2022). Cyber sextortion: An exploratory analysis of different perpetrators engaging in a similar crime. *Journal of Interpersonal Violence*, 37(1–2), 258–283. <https://doi.org/10.1177/0886260520909186>
- Pabian, S., & Vandebosch, H. (2016). An investigation of short-term longitudinal associations between social anxiety and victimization and perpetration of traditional bullying and cyberbullying. *Journal of Youth and Adolescence*, 45(2), 328–339.
<https://doi.org/10.1007/s10964-015-0259-3>
- Petrič, G. (2006). Conceptualizing and measuring the social uses of the internet: The case of personal web sites. *The Information Society*, 22(5), 291–301.
<https://doi.org/10.1080/01972240600904159>
- Petrosyn, A. (2023, March 10). Cybercrime encounter rate in selected countries. *Statista*.
<https://www.statista.com/statistics/194133/cybercrime-rate-in-selected-countries/>
- Polanin, J. R., Espelage, D. L., Grotz, J. K., Ingram, K., Michaelson, L., Spinney, E., Valido, A., Sheikh, A. E., Torgal, C., & Robinson, L. (2022). A systematic review and meta-

- analysis of interventions to decrease cyberbullying perpetration and victimization. *Prevention Science*, 23(3), 439–454. <https://doi.org/10.1007/s11121-021-01259-y>
- Pratt, T. C., Turanovic, J. J., Fox, K. A., & Wright, K. A. (2014). Self-control and victimization: A meta-analysis. *Criminology*, 52(1), 87–116. <https://doi.org/10.1111/1745-9125.12030>
- Resett, S., & Mesurado, B. (2021). Bullying and cyberbullying in adolescents: A meta-analysis on the effectiveness of interventions. In P. Á. Gargiulo & H. L. Mesones Arroyo (Eds.), *Psychiatry and Neuroscience Update* (pp. 445–458). Springer International Publishing. https://doi.org/10.1007/978-3-030-61721-9_32
- Sampasa-Kanyinga, H., Lalande, K., & Colman, I. (2020). Cyberbullying victimisation and internalising and externalising problems among adolescents: The moderating role of parent–child relationship and child’s sex. *Epidemiology and Psychiatric Sciences*, 29, e8. <https://doi.org/10.1017/S2045796018000653>
- Sarier, Y. (2022). Turkey’s analysis of cyberbullying and cyber victimization of students in Turkey in terms of demographic variables by meta-analysis method. *Kastamonu Eğitim Dergisi*, 30(2), 283–296. <https://doi.org/10.24106/kefdergi.754228>
- Seçkin Kapucu, M., Özcan, H., & Karakaya Özyer, K. (2021). The relationship between secondary school students’ digital literacy levels, social media usage purposes and cyberbullying threat level. *International Journal of Modern Education Studies*, 5(2). <https://doi.org/10.51383/ijonmes.2021.136>

- Slonje, R., & Smith, P. K. (2008). Cyberbullying: Another main type of bullying? *Scandinavian Journal of Psychology*, *49*(2), 147–154. <https://doi.org/10.1111/j.1467-9450.2007.00611.x>
- Smith, P. K. (2011). Bullying in schools: Thirty years of research. In C. P. Monks & I. Coyne (Eds.), *Bullying in Different Contexts* (1st ed., pp. 36–60). Cambridge University Press. <https://doi.org/10.1017/CBO9780511921018.003>
- Smith, P. K. (2019). Research on cyberbullying: Strengths and limitations. In H. Vandebosch & L. Green (Eds.), *Narratives in Research and Interventions on Cyberbullying among Young People* (pp. 9–27). Springer International Publishing. https://doi.org/10.1007/978-3-030-04960-7_2
- Subrahmanyam, K., & Greenfield, P. (2008). Online communication and adolescent relationships. *The Future of Children*, *18*(1), 119–146.
- Subrahmanyam, K., & Šmahel, D. (2011). Constructing identity online: Identity exploration and self-presentation. In K. Subrahmanyam & D. Smahel, *Digital Youth* (pp. 59–80). Springer New York. https://doi.org/10.1007/978-1-4419-6278-2_4
- Suler, J. (2005). The online disinhibition effect. *International Journal of Applied Psychoanalytic Studies*, *2*(2), 184–188. <https://doi.org/10.1002/aps.42>
- Sun, S., & Fan, X. (2018). Is there a gender difference in cyber-victimization?: A meta-analysis. *Journal of Media Psychology*, *30*(3), 125–138. <https://doi.org/10.1027/1864-1105/a000185>

- Tao, S., Reichert, F., Law, N., & Rao, N. (2022). Digital technology use and cyberbullying among primary school children: Digital literacy and parental mediation as moderators. *Cyberpsychology, Behavior, and Social Networking*, *25*(9), 571–579.
<https://doi.org/10.1089/cyber.2022.0012>
- Tran, H. G. N., Thai, T. T., Dang, N. T. T., Vo, D. K., & Duong, M. H. T. (2023). Cyber-victimization and its effect on depression in adolescents: A systematic review and meta-analysis. *Trauma, Violence, & Abuse*, *24*(2), 1124–1139.
<https://doi.org/10.1177/15248380211050597>
- Van Cleemput, K., Desmet, A., Vandebosch, H., Bastiaensens, S., Poels, K., & De Bourdeudhuij, I. (2014, February 3). *A systematic review of studies evaluating anti-cyberbullying programs*. Etmaal van de Communicatiewetenschap, Wageningen, Netherlands.
- Van Geel, M., Vedder, P., & Tanilon, J. (2014). Relationship between peer victimization, cyberbullying, and suicide in children and adolescents: A meta-analysis. *JAMA Pediatrics*, *168*(5), 435. <https://doi.org/10.1001/jamapediatrics.2013.4143>
- Vandebosch, H., & Van Cleemput, K. (2008). Defining cyberbullying: A qualitative Research into the perceptions of youngsters. *CyberPsychology & Behavior*, *11*(4), 499–503.
<https://doi.org/10.1089/cpb.2007.0042>
- Vogels, E. A. (2022, December 15). Teens & cyberbullying 2022. *Pew Research Centre*.
<https://www.pewresearch.org/internet/2022/12/15/teens-and-cyberbullying-2022/>

- Völlink, T., Dehue, F., McGukin, C., & Jacobs, N. C. L. (2016). An introduction in cyberbullying research. In T. Völlink, F. Dehue, & C. McGukin (Eds.), *Cyberbullying: From Theory to Intervention* (pp. 3–14). Routledge.
- Wade, A., & Beran, T. (2011). Cyberbullying: The New Era of Bullying. *Canadian Journal of School Psychology, 26*(1), 44–61. <https://doi.org/10.1177/0829573510396318>
- Walker, J., Craven, R. G., & Tokunga, R. S. (2013). Introduction. In S. Bauman, D. Cross, & J. Walker (Eds.), *Principles of Cyberbullying Research* (pp. 31–48). Routledge. <https://doi.org/10.4324/9780203084601-11>
- Walters, G. D. (2021). School-age bullying victimization and perpetration: A meta-analysis of prospective studies and research. *Trauma, Violence, & Abuse, 22*(5), 1129–1139. <https://doi.org/10.1177/1524838020906513>
- Wang, L., & Jiang, S. (2023). Effectiveness of parent-related interventions on cyberbullying among adolescents: A systematic review and meta-analysis. *Trauma, Violence, & Abuse, 24*(5), 3678–3696. <https://doi.org/10.1177/15248380221137065>
- Wang, M.-J., Yogeewaran, K., Andrews, N. P., Hawi, D. R., & Sibley, C. G. (2019). How common is cyberbullying among adults? Exploring gender, ethnic, and age differences in the prevalence of cyberbullying. *Cyberpsychology, Behavior, and Social Networking, 22*(11), 736–741. <https://doi.org/10.1089/cyber.2019.0146>
- Willard, N. E. (2007). *Cyberbullying and cyberthreats: Responding to the challenge of online social aggression, threats, and distress*. Champaign, IL: Research Press.

- Wirth, J. (2018). *The effectiveness of intervention programs for cyberbullying in schools: A systematic review and meta-analysis* [Master Thesis, University of Adelaide]. Adelaide Research & Scholarship. <https://hdl.handle.net/2440/129364>
- Wissink, I. B., Standaert, J. C. A., Stams, G. J. J. M., Asscher, J. J., & Assink, M. (2023). Risk factors for juvenile cybercrime: A meta-analytic review. *Aggression and Violent Behavior, 70*, 101836. <https://doi.org/10.1016/j.avb.2023.101836>
- Wong, L. Y. N. (2021). *Towards a comprehensive model of cyberbullying: Age differences, cultural differences and the complex role of morality* [Doctoral Dissertation, The Chinese University of Hong Kong]. ProQuest Dissertations Publishing.
- Yuchang, J., Junyi, L., Junxiu, A., Jing, W., & Mingcheng, H. (2019). The differential victimization associated with depression and anxiety in cross-cultural perspective: A meta-analysis. *Trauma, Violence, & Abuse, 20*(4), 560–573. <https://doi.org/10.1177/1524838017726426>
- Zhang, Y., & Chen, J.-K. (2023). Emotional intelligence and school bullying victimization in children and youth students: A meta-analysis. *International Journal of Environmental Research and Public Health, 20*(6), 4746. <https://doi.org/10.3390/ijerph20064746>
- Zweig, J. M., Lachman, P., Yahner, J., & Dank, M. (2014). Correlates of cyber dating abuse among teens. *Journal of Youth and Adolescence, 43*(8), 1306–1321. <https://doi.org/10.1007/s10964-013-0047-x>
- Zych, I., Baldry, A. C., Farrington, D. P., & Llorent, V. J. (2019). Are children involved in cyberbullying low on empathy? A systematic review and meta-analysis of research on

empathy versus different cyberbullying roles. *Aggression and Violent Behavior*, 45, 83–97. <https://doi.org/10.1016/j.avb.2018.03.004>

Zych, I., Viejo, C., Vila, E., & Farrington, D. P. (2021). School bullying and dating violence in adolescents: A systematic review and meta-analysis. *Trauma, Violence, & Abuse*, 22(2), 397–412. <https://doi.org/10.1177/1524838019854460>

Decision Letter, first revision:

24th June 2024

Dear Dr. Kasturiratna,

Thank you for your patience as we've prepared the guidelines for final submission of your Nature Human Behaviour manuscript, "A Systematic Review of Meta-Analyses on the Risk Factors, Protective Factors, Consequences, and Interventions of Cyberbullying Victimization" (NATHUMBEHAV-23103327A). Please carefully follow the step-by-step instructions provided in the attached file, and add a response in each row of the table to indicate the changes that you have made. Please also check and comment on any additional marked-up edits we have proposed within the text. Ensuring that each point is addressed will help to ensure that your revised manuscript can be swiftly handed over to our production team.

We would hope to receive your revised paper, with all of the requested files and forms within two-three weeks. Please get in contact with us if you anticipate delays.

Nature Human Behaviour offers a Transparent Peer Review option for new original research manuscripts submitted after December 1st, 2019. As part of this initiative, we encourage our authors to

support increased transparency into the peer review process by agreeing to have the reviewer comments, author rebuttal letters, and editorial decision letters published as a Supplementary item. When you submit your final files please clearly state in your cover letter whether or not you would like to participate in this initiative. Please note that failure to state your preference will result in delays in accepting your manuscript for publication.

In recognition of the time and expertise our reviewers provide to Nature Human Behaviour's editorial process, we would like to formally acknowledge their contribution to the external peer review of your manuscript entitled "A Systematic Review of Meta-Analyses on the Risk Factors, Protective Factors, Consequences, and Interventions of Cyberbullying Victimization". For those reviewers who give their assent, we will be publishing their names alongside the published article.

Cover suggestions

We welcome submissions of artwork for consideration for our cover. For more information, please see our guide for cover artwork.

ORCID

Non-corresponding authors do not have to link their ORCIDs but are encouraged to do so. Please note that it will not be possible to add/modify ORCIDs at proof. Thus, please let your co-authors know that if they wish to have their ORCID added to the paper they must follow the procedure described in the following link prior to acceptance: <https://www.springernature.com/gp/researchers/orcid/orcid-for-nature-research>

Nature Human Behaviour has now transitioned to a unified Rights Collection system which will allow our Author Services team to quickly and easily collect the rights and permissions required to publish your work. Approximately 10 days after your paper is formally accepted, you will receive an email in providing you with a link to complete the grant of rights. If your paper is eligible for Open Access, our Author Services team will also be in touch regarding any additional information that may be required to arrange payment for your article.

Please note that *Nature Human Behaviour* is a Transformative Journal (TJ). Authors may publish their research with us through the traditional subscription access route or make their paper immediately open access through payment of an article-processing charge (APC). Authors will not be required to make a final decision about access to their article until it has been accepted. Find out more about Transformative Journals

[REDACTED]

Best regards,

[REDACTED]

On behalf of

[REDACTED]

Reviewer #1:

Remarks to the Author:

Thank you for your in-depth and comprehensive responsiveness to the comments. While the manuscript is much improved, it is now 81 pages, with about 7000 words of text. Some of what has been added could be more concise, and I'd encourage you to trim wordiness throughout the manuscript as much as possible. I defer to the editors with respect to length, but I do think you face a trade-off between comprehensiveness and readability due to length.

In addition, I have a few small suggestions:

It may strengthen your discussion of measurement to underscore the importance of behavioral measurement (e.g., Mehari & Farrell, 2014) in the face of imprecise definitions.

Routine Measurement Theory was added to the contextual section of the introduction—in your view does it only affect contextual variables? Or does this theory apply across variables? If it applies across the study, consider a separate section for the theory, or add a phrase/sentence about how it applies broadly in the current study section.

Reviewer #2:

Remarks to the Author:

The authors have been highly responsive to all the feedback provided by each reviewers and have made extensive changes to their manuscript; as a result, it has greatly improved the contribution this excellent work makes. No further comments at this time.

Author Rebuttal, first revision:

Ms. Ref. No.: NATHUMBEHAV-23103327A

Title: Umbrella Review of Meta-Analyses on the Risk Factors, Protective Factors, Consequences, and Interventions of Cyberbullying Victimization

We thank the Reviewers for their thorough and incisive reviews, which have significantly strengthened the revised paper. We strove to carefully address both Reviewers' concerns and suggestions and hope that this revision is much stronger. Our point-by-point responses are listed and detailed below, along with the Reviewers' comments.

Thank you once again for your consideration of our paper.

Reviewer #1

Thank you for your in-depth and comprehensive responsiveness to the comments.

- 1. While the manuscript is much improved, it is now 81 pages, with about 7000 words of text. Some of what has been added could be more concise, and I'd encourage you to trim wordiness throughout the manuscript as much as possible. I defer to the editors with respect to length, but I do think you face a trade-off between comprehensiveness and readability due to length.*

Response:

We thank the Reviewer for the positive feedback and constructive comments, which we have incorporated in our revision to enhance the quality of our manuscript. We have trimmed the manuscript such that it is as concise as possible while maintaining the integrity of the content. Additionally, we have carefully edited the text to meet the journal's word limit requirements.

- In addition, I have a few small suggestions: It may strengthen your discussion of measurement to underscore the importance of behavioral measurement (e.g., Mehari & Farrell, 2014) in the face of imprecise definitions.*

Response:

We thank the Reviewer for this important suggestion, and have incorporated a section on the importance of behavioural measurements within the discussion (p. 20). Specifically, we have discussed how most meta-analyses included in the current review mainly defined cyberbullying by focusing on the different media platforms through which cyberbullying occurs, rather than on the different acts of cyberbullying (Erbiçer et al., 2023; Kowalski et al., 2014; Modecki et al., 2014; Molero et al., 2022). However, research suggests that individuals do not distinguish cyberbullying based on the medium used but rather on the nature of the bullying acts themselves (Mehari et al., 2014). Therefore, future work should aim to refine definitions that emphasise behaviours involved in cyberbullying and incorporate behavioural measurements within cyberbullying scales to more accurately capture the phenomenon.

- Routine Measurement Theory was added to the contextual section of the introduction—in your view does it only affect contextual variables? Or does this theory apply across variables? If it applies across the study, consider a separate section for the theory, or add a phrase/sentence about how it applies broadly in the current study section.*

Response:

We thank the Reviewer for the constructive suggestion. We have included a discussion of the Routine Activity Theory in the introduction, specifically in relation to unregulated environments that provide

vulnerable targets for cyberbullying (p. 7). Specifically, we mention that unregulated environments provide vulnerable targets and allow the unrestrained perpetration of cyberbullying in the absence of parental guardians or teachers, consistent with the Routine Activity Theory—deviant behaviours like cyberbullying occur in the presence of motivated offenders, suitable targets, and an absence of capable guardians (Aizenkot, 2022; Akgül, 2023; Kalia & Aleem, 2017). Routine Activity Theory suggests that the lack of effective supervision increases the opportunity for cyberbullying, emphasising the importance of considering environmental factors as a predictor of cyberbullying victimisation.

Reviewer #2

The authors have been highly responsive to all the feedback provided by each reviewers and have made extensive changes to their manuscript; as a result, it has greatly improved the contribution this excellent work makes. No further comments at this time.

Response:

We thank the Reviewer for the positive feedback, and greatly appreciate all the constructive comments given throughout the review process which greatly strengthened our manuscript.

References

- Aizenkot, D. (2022). The Predictability of Routine Activity Theory for Cyberbullying Victimization Among Children and Youth: Risk and Protective Factors. *Journal of Interpersonal Violence*, 37(13–14), NP11857–NP11882. <https://doi.org/10.1177/0886260521997433>
- Akgül, G. (2023). Routine Activities Theory in cyber victimization and cyberbullying experiences of Turkish adolescents. *International Journal of School & Educational Psychology*, 11(2), 135–144. <https://doi.org/10.1080/21683603.2021.1980475>
- Erbıçer, E. S., Ceylan, V., Yalçın, M. H., Erbıçer, S., Akın, E., Koçtürk, N., & Doğan, T. (2023). Cyberbullying among children and youth in Türkiye: A systematic review and meta-analysis. *Journal of Pediatric Nursing*, 73, 184–195. <https://doi.org/10.1016/j.pedn.2023.09.003>
- Kalia, D., & Aleem, S. (2017). Cyber Victimization Among Adolescents: Examining the Role of Routine Activity Theory. *Journal of Psychosocial Research*, 12(1), 223-.
- Kowalski, R. M., Giumetti, G. W., Schroeder, A. N., & Lattanner, M. R. (2014). Bullying in the Digital Age: A Critical Review and Meta-Analysis of Cyberbullying Research Among Youth. *Psychological Bulletin*, 140(4), 1073–1137. <https://doi.org/10.1037/a0035618>
- Mehari, K. R., Farrell, A. D., & Le, A.-T. H. (2014). Cyberbullying Among Adolescents: Measures in Search of a Construct. *Psychology of Violence*, 4(4), 399–415. <https://doi.org/10.1037/a0037521>
- Modecki, K. L., Minchin, J., Harbaugh, A. G., Guerra, N. G., & Runions, K. C. (2014). Bullying Prevalence Across Contexts: A Meta-analysis Measuring Cyber and Traditional

Bullying. *Journal of Adolescent Health*, 55(5), 602–611.

<https://doi.org/10.1016/j.jadohealth.2014.06.007>

Molero, M. M., Martos, Á., Barragán, A. B., Pérez-Fuentes, M. C., & Gázquez, J. J. (2022). Anxiety and Depression from Cybervictimization in Adolescents: A Metaanalysis and Meta-regression Study. *The European Journal of Psychology Applied to Legal Context*, 14(1), 42–50.

<https://doi.org/10.5093/ejpalc2022a5>

Final Decision Letter:

Dear Ms Kasturiratna,

We are pleased to inform you that your Article "A Systematic Review of Meta-Analyses on the Risk Factors, Protective Factors, Consequences, and Interventions of Cyberbullying Victimisation", has now been accepted for publication in *Nature Human Behaviour*.

Please note that *Nature Human Behaviour* is a Transformative Journal (TJ). Authors may publish their research with us through the traditional subscription access route or make their paper immediately open access through payment of an article-processing charge (APC). Authors will not be required to make a final decision about access to their article until it has been accepted. Find out more about Transformative Journals

Once your manuscript is typeset and you have completed the appropriate grant of rights, you will receive a link to your electronic proof via email with a request to make any corrections within 48 hours. If, when you receive your proof, you cannot meet this deadline, please inform us at

rjsproduction@springernature.com immediately. Once your paper has been scheduled for online publication, the Nature press office will be in touch to confirm the details.

With best regards,

[REDACTED]